# Clostridia from preterm infants metabolize human milk oligosaccharides to suppress pathobionts and modulate intestinal function in organoids

Jonathan A. Chapman [1,14], Andrea C. Masi[1,14], Lauren C. Beck[1], Hannah Watson[1], Gregory R. Young[1], Hilary P. Browne [2], Yan Shao [2], Raymond Kiu [3,4,5], Andrew Nelson[6], Jennifer A. Doyle[1], Pawel Palmowski[7], Márton Lengyel[8], James P. R. Connolly [7], Christopher A. Lamb [1,9], Andrew Porter[7], Trevor D. Lawley [2], Lindsay J. Hall [3,4,5,10], Nicholas D. Embleton [11,12], John D. Perry [13], Janet E. Berrington [1,11] & Christopher J. Stewart [1] ✉

Infant gut microbiome development is strongly impacted by breastmilk and human milk oligosaccharides (HMOs), which can protect preterm infants against pathologies including necrotizing enterocolitis. HMO metabolism in bifidobacteria is well characterized and linked to health outcomes, but the scope of HMO-utilizing species remains unclear. Here, using a combination of genomics, proteomics and metabolomics, we show that *Clostridium* species isolated from preterm infants (born at <32 weeks gestation), in particular *Clostridium perfringens* lacking the toxin perfringolysin O (PfoA), metabolized HMOs. *Clostridium* species produced beneficial metabolites including short-chain fatty acids and tryptophan catabolites at higher quantities than *Bifidobacterium* species in vitro. Cell-free supernatant from *C. perfringens* was non-toxic to colonic cell lines, promoted the growth of commensal bifidobacteria and inhibited growth of pathobionts isolated from the preterm infant gut in vitro. It also suppressed inflammation in preterm-derived intestinal organoids. These findings expand our understanding of HMO-metabolizing microbes and suggest that *pfoA⁻ C. perfringens* strains could contribute to healthy infant gut development.

Early-life gut microbiome development plays a critical role in shaping short- and long-term health. Preterm infants born with <32 weeks of gestation undergo altered development of their gut microbiome that is partly linked to pathologies such as necrotizing enterocolitis (NEC), an inflammatory-mediated bowel disease with a high risk of mortality and morbidity[1]. Receipt of human milk is an important driver of infant gut microbiome composition[2] and protects against NEC, most probably through provision of bioactive factors, such as human milk

oligosaccharides (HMOs)[3–7]. HMOs are complex unconjugated sugars indigestible to humans that act as prebiotics for some gut bacteria, most notably *Bifidobacterium* spp. that are associated with breast-fed babies' microbiome and health[8].

HMO-utilizing bacteria such as *Bifidobacterium* spp. improve intestinal barrier function and positively influence immune system development, lowering systemic inflammation[9] and protecting against immune-mediated diseases such as atopy and asthma[10]. This is partly

mediated by bacterial metabolites including short-chain fatty acids (SCFAs)[11] and tryptophan catabolites[12,13] that interact with host cells. *Bifidobacterium* spp. also shape the wider microbiota by producing metabolic breakdown products from HMOs, allowing 'cross-feeding' by other beneficial species, promoting their growth and suppressing growth of pathogens[14,15]. This knowledge has contributed to the rise in probiotic use in preterm infants over the past decade and inclusion of synthetic HMOs in term formula, including 2′-fucosyllactose (2′-FL), lacto-*N*-tetraose (LNT), lacto-*N*-neotetraose (LNnT) and 6′-sialyllactose (6′SL)[16]. Recent studies have demonstrated other genera can also digest HMOs through varied pathways, as observed for *Bacteroides*[17], *Akkermansia*,[18] *Roseburia–Eubacterium* group[19], *Ruminococcus*[20] and *Collinsella*[20]. Notably, these genera do not typically colonize preterm infants[21] and the full diversity of bacteria that metabolize HMOs within this population is unknown.

The current study sought to comprehensively describe previously unrecognized HMO-utilizing bacteria that colonize preterm infants and decipher their potential function within the gut at transcriptomic, proteomic and metabolomic levels. We further assessed their roles in modulating abundant species from the preterm gut microbiome, as well as their impact on intestinal barrier function, as proposed mechanisms of neonatal health.

## Results

### *Clostridium* spp. and *Bifidobacterium* spp. isolated from preterm infants metabolize HMOs

We screened the abilities of 29 bacterial isolates, mostly from preterm infant stool (*n* > 15 infants), to grow on 6 different HMOs, and glucose and lactose (Fig. 1a). These species were obtained by untargeted cultivation. Post hoc we re-analysed metagenome data from ref. 21, revealing that these 29 isolates represent a median of 80% (interquartile range 61%–91%) of all relative microbial abundance observed in 123 preterm infants. Only *Bifidobacterium* and *Clostridium* species were able to use HMOs (Fig. 1a). Except for *B. animalis*, all *Bifidobacterium* (*n* = 7 isolates) grew on at least one of the HMOs, with LNT and LNnT being most frequently used. Except for *C. butyricum*, all *Clostridium* species, namely *C. perfringens, C. tertium, C. baratii* and *C. paraputrificum* (*n* = 11 isolates), were able to grow on one or more of the HMOs tested.

Of particular note was that four *C. perfringens* isolates clustered on the basis of the sugar growth profile with the probiotic-derived *B. infantis* (LB1; Labinic, Biofloratech), demonstrating that these *C. perfringens* share some metabolic functionality with this clinically used strain. Having discovered that *C. perfringens* could use HMOs, we tested a wider collection of *C. perfringens* strains (obtained from ref. 22), which showed that all could use HMOs (Supplementary Data Fig. 1). Unlike *Bifidobacterium*, *C. perfringens* generally could not grow on LNT. The HMO disialyllacto-*N*-tetraose (DSLNT) has previously been associated with protection from NEC in observational studies[5,7] and *C. perfringens* strain AM1 reached the highest optical density on this HMO (Fig. 1a).

Using genus-level publicly available data from the Mechanisms Affecting Gut of Preterm Infants (MAGPIE) study[23], we found that *Clostridium* was prevalent in preterm infants, but at significantly lower rates in those diagnosed with NEC compared with time-matched healthy controls (*P* < 0.001; Extended Data Fig. 1a). In comparison, prevalence of *Bifidobacterium*, the only other preterm infant HMO users, was similar between NEC and controls (Extended Data Fig. 1b). In terms of relative abundance, *Bifidobacterium* (median: 0.056, IQR: 0.73) was slightly higher than *Clostridium* (median: 0.052, IQR: 2.24) but neither were significantly different between NEC and controls (Extended Data Fig. 1c,d). At the strain level, *C. perfringens* lacking the gene encoding the toxin perfringolysin O (*pfoA*) are linked to the neonatal-derived health-associated hypovirulent lineage V[22], while carriage of *pfoA* is associated with increased risk of NEC and paediatric inflammatory bowel disease[24]. Further strain-level analysis using publicly available metagenome-assembled genomes[25] showed *pfoA*⁻ *C. perfringens* strains were relatively common in preterm infants (detected in 32/158 infants), with a median abundance of 0.6% (IQR = 0.01–2.06). Greater proportional prevalence (Extended Data Fig. 1e; *P* = 0.14) and relative abundance (control: median 0.64%, IQR = 0.01–2.05%; NEC: median 0.26%, IQR = 0.12–21.0%; *P* = 0.07) of *pfoA*⁻ strains were observed in control than in NEC infants, but these differences did not reach the significance threshold. Temporal colonization by *pfoA*⁻ strains was significantly more stable in controls (median = 100%, IQR = 100–100%) than in NEC cases (median = 53%, IQR = 43–77%) (Extended Data Fig. 1f; *P* = 0.03). Thus, in subsequent work we focused on AM1, a *C. perfringens pfoA*⁻ isolate notable for its ability to grow on health-associated DSLNT (this specific AM1 isolate will be referred to as *CP-pfoA*⁻ henceforth) (Fig. 1a–c). This isolate was also found to be susceptible to commonly prescribed antibiotics in neonatal intensive care (Supplementary Table 1).

Whole-genome sequencing confirmed that the *Clostridium* strains did not contain the same HMO utilization gene cluster observed in *Bifidobacterium* strains. To identify genes potentially involved in HMO metabolism, RNA-seq was performed during exponential growth of *CP-pfoA*⁻ using DSLNT, LNnT, 6′SL and lactose. Transcriptome data showed clustering based on sugar growth profiles (Extended Data Fig. 2a). Comparing each HMO to lactose, the highest number of differentially expressed genes (DEGs) were observed with DSLNT, followed by LNnT and 6′SL (Supplementary Table 2). Among the top 20 upregulated DEGs with DSLNT were those encoding a predicted glycoside hydrolase (GH) 101 CAZyme (endo-α-*N*-acetylgalactosaminidase; locus 01633) which showed the highest log fold change (7.5), and enzymes involved in sialic acid metabolism (*nanM*, two *nanA* genes) (Fig. 1d). Specific to DSLNT, the most upregulated genes included one encoding a GH112 protein (1,3-beta-galactosyl-*N*-acetylhexosamine phosphorylase, locus 02923) and three genes involved in diacetylchitobiose transport (*ngcG, ngcF, ngcE*; loci 02918, 02919, 02920). On 6′SL, the top 20 upregulated DEGs involved sialic acid (*nanM*, two *nanA*, *nanE*) and fucose (*fucI, fucU, fucA, fucO, fucP*) metabolism (Extended Data Fig. 2b). Finally, in LNnT, most top upregulated genes encoded hypothetical proteins, but also genes involved in arginine metabolism and transport (*argG, argH, artQ, artP*) (Extended Data Fig. 2c). A total of 11 different classes

---

**Fig. 1 | *Clostridium* spp. and *Bifidobacterium* spp. have the capacity to metabolize HMOs. a**, Growth of 29 bacterial isolates on 6 HMOs and lactose. The values reported represent the maximum OD$_{600}$ reached normalized to glucose. *Clostridium* and *Bifidobacterium* are overrepresented owing to their ability to use HMOs and subsequent testing of species and strain variability in HMO utilization. **b**, Alignment of 11 de novo genomes with toxin and colonization profiles and genome sizes. Lineage assignment is according to ref. 22. **c**, A phylogenetic tree of 313 strains where the 11 de novo genomes in this study were compared with 302 strains assigned to 8 lineages, as in ref. 22. Lineage V is a hypovirulent clade. Three isolate genomes in this study were assigned to this lineage. All three were found to be lacking toxin gene *pfoA*, which is typical of lineage V strains. **d**, Volcano plot of RNA-seq data 'on supernatant' for AM1 grown on DSLNT vs lactose. A gene was considered differentially expressed when absolute log$_2$(fold change) > 2

and *P*$_{adj}$ < 0.05. Statistical significance was calculated using a two-tailed Wald test, followed by adjustment for multiple comparisons using the Benjamini–Hochberg method. **e,f**, Proteomics data on supernatant (**e**) and pellet (**f**) for AM1 grown on DSLNT vs lactose. Proteins were deemed significant when associated to an absolute log$_2$(fold change) > 1 and *P*$_{adj}$ < 0.05. Statistical significance was calculated using a two-tailed moderated *t*-test, followed by adjustment for multiple comparisons using the Benjamini–Hochberg method. **d–f**, A positive log$_2$(fold change) indicates upregulation on DSLNT relative to lactose, while a negative fold change indicates downregulation on DSLNT relative to lactose. LNT, lacto-*N*-tetraose; LNnT, lacto-*n*-neotetraose; LNFP I, lacto-*N*-fucopentaose I; 6′-SL, 6′-sialyllactose; 2′-FL, 2′-fucosyllactose; DSLNT, disialyllacto-*N*-tetraose; GH, glycoside hydrolase.

of GHs were upregulated across all HMOs tested, 3 of which have been shown to act on HMOs; GH29, GH85 and GH112 (ref. 26). Others, such as the GH84 *nagJ* have been shown to act on mucin *O*-glycans and may also target HMOs owing to structural similarity[27].

Proteomics on supernatant (that is, secreted) and cell pellet (that is, intracellular or cell-associated) peptides also clustered by sugar

utilized (Extended Data Fig. 2d,e). Similar to RNA-seq, growth on DSLNT resulted in upregulation of multiple GH family and sialic acid degrading (NanA, NanM, NanE, NanH) proteins (Fig. 1e,f). The protein encoded in locus 01633 (GH101, top upregulated gene in transcriptomics) was the fourth most significant protein in DSLNT supernatant, but not in the pellet, suggesting that this is an uncharacterized enzyme acting

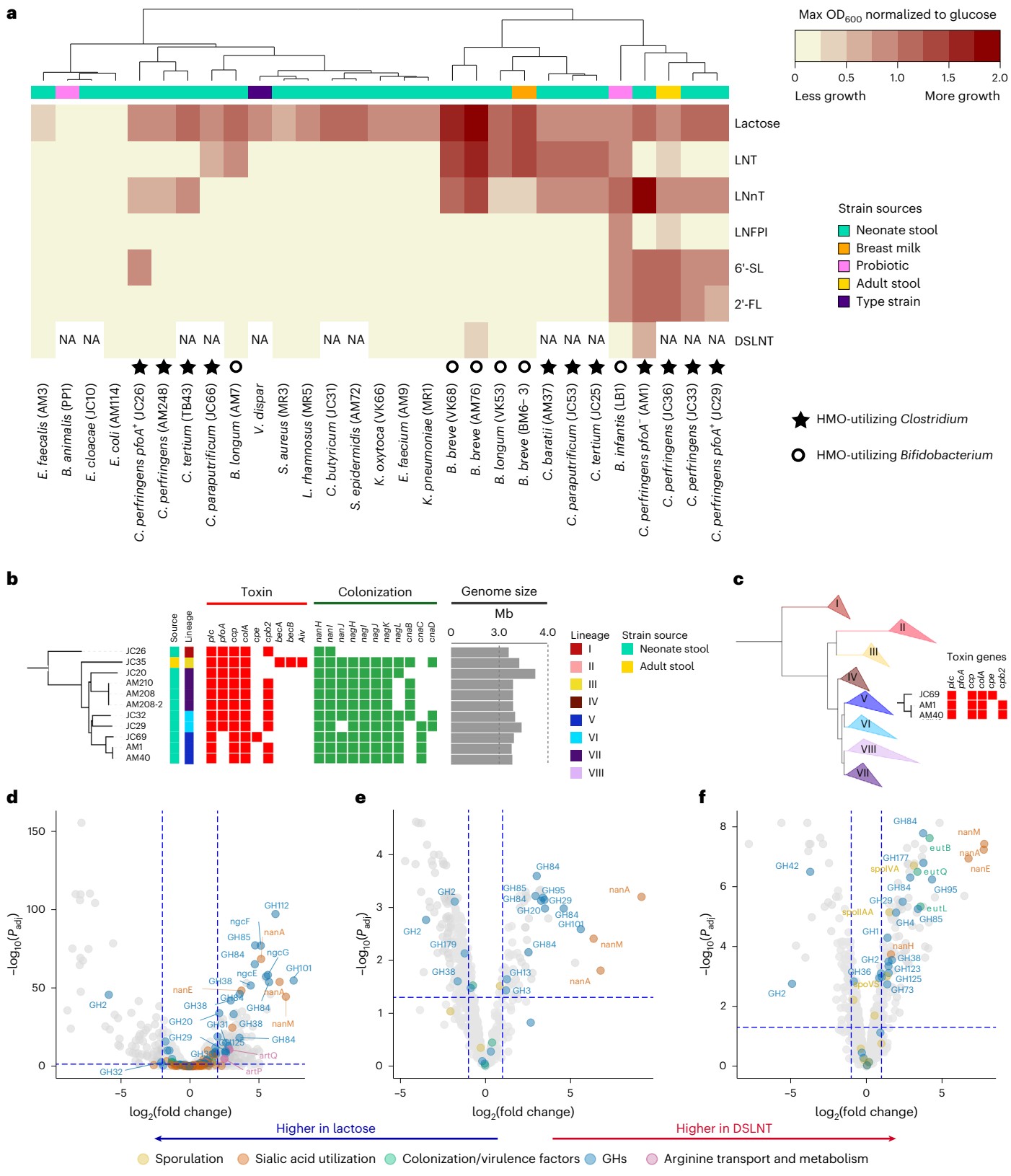

extracellularly, probably in conjunction with two NanA proteins and NanM which represented the three most upregulated proteins. HMO quantification of spent media showed that lactose, LNnT and 6'SL were completely degraded, but DSLNT metabolism generated undigested by-products, in particular LNT (Supplementary Table 3). Specifically, accumulation of LNB suggests the presence of an enzyme able to break the β1-3 bond between LNB and lactose before complete desialylation. The presence of LNT further indicates that the sialidases could act on both DSLNT and the sialylated LNB, and the full digestion of DSLNT was possible only when the β1-3 bond was cleaved first.

### *Clostridium* spp. produced more diverse and abundant beneficial metabolites than *Bifidobacterium* spp

We next compared metabolites produced by HMO-utilizing *Clostridium* (*C. perfringens*, *C. tertium*, *C. baratii* and *C. paraputrificum*) and *Bifidobacterium* (*B. infantis*, *B. breve* and *B. longum*) strains. We hypothesized that *Clostridium* species metabolizing HMOs may, similar to *Bifidobacterium* species, produce beneficial metabolites such as SCFAs that play critical roles in gut health including providing energy, regulating the immune system, maintaining gut barrier integrity and modulating microbiome composition[11,28]. SCFA profiling was therefore performed on culture supernatants collected following the HMO growth assay shown in Fig. 1a, allowing assessment of production during growth on the individual HMOs, glucose or lactose. Compared with *Bifidobacterium*, supernatants from *C. perfringens* and *C. baratii* contained significantly greater growth-adjusted concentrations of butyrate (all $P < 0.05$) and propionate (all $P < 0.05$) (Fig. 2a). *C. perfringens* and *C. baratii* also produced greater total quantities of SCFAs than *Bifidobacterium* (Fig. 2a,b). Consistent findings were seen in media supplemented with either glucose or lactose (Supplementary Data Fig. 2).

To investigate a broader range of metabolite production, we next generated cell-free supernatants (CFSs) from each strain of interest and performed untargeted metabolomics. Two CFSs were produced per strain, using either a strain-specific mixture of the HMOs that we found they can utilize, or glucose. Unsupervised ordinations of the metabolomic profiles following growth on HMOs showed clustering by species and more broadly by genus (Fig. 2c). This was consistent with glucose (Extended Data Fig. 3a), and glucose and HMO metabolomic profiles were comparable within each strain (Supplementary Table 4). Thus, we focused subsequent analysis on CFSs derived from growth with HMOs.

Compared to media controls, CP-*pfoA⁻* CFS showed significantly increased polyamine formation, tryptophan biosynthesis and catabolism, butyrate/isobutyrate production, and generation of other neuromodulatory, immunomodulatory or antimicrobial metabolites, with depletion and modification of amino acids (Fig. 2d). Many of these metabolites were differentially abundant in other CFSs, with high similarities between the 3 *C. perfringens* and *C. tertium* CFSs (Extended Data Fig. 3b–e). However, similarity across all 5 preterm

infant-derived *Clostridium* strains was low, with only 13 upregulated metabolites shared by all (Extended Data Fig. 3b). *B. infantis* (LB1) and *B. breve* (AM76) CFSs were distinct in containing multiple gamma-glutamyl amino acids (Extended Data Fig. 3f,g). *C. baratii* (AM37), *C. paraputrificum* (JC53) and *B. longum* (AM7) showed a low number of these metabolites of interest, namely 6, 3 and 1, respectively (Extended Data Fig. 3h–j).

*C. perfringens* and *C. tertium* produced a broad range of indole-containing tryptophan catabolites (associated with promoting intestinal barrier function and inhibiting inflammation)[12,29,30] along with tryptophan itself, at significantly higher levels than *Bifidobacterium* and other *Clostridium* spp. (Fig. 2e). In addition, *C. perfringens* and *C. tertium* CFSs contained higher levels of polyamines[31] (associated with increased tight junction expression and inflammation reduction, although accumulation to higher concentrations has been shown to induce cytotoxicity)[32–34] and their precursors (Extended Data Fig. 4a). Only ornithine and citrulline were significantly raised in *Bifidobacterium* CFSs, specifically in *B. infantis* (LB1). Only *C. perfringens* CFSs contained significantly raised levels of histamine (produced by probiotic strains, can alter intestinal motility[35], and suppress cytokine secretion and wider intestinal inflammation)[36] (Extended Data Fig. 4b). Neuromodulators 3-hydroxybutyrate and *N*-acetylaspartate were also significantly higher in *C. perfringens* CFSs compared with those in all other strains (Extended Data Fig. 4b). Finally, both phenyllactate (broad spectrum antimicrobial)[37] and lactate (major metabolite of the infant gut, acidifies the gut lumen and an intermediate for SCFAs)[38] were significantly higher in all *Bifidobacterium* and *Clostridium* CFSs compared with media only, except in *C. baratii* (Extended Data Fig. 4b).

### *Clostridium* spp. CFSs suppressed pathobiont growth and promoted *Bifidobacterium* spp. growth

We next assessed whether the *Clostridium* and *Bifidobacterium* CFSs could suppress the growth of 4 of the most abundant pathobionts present in the preterm gut microbiome (*Escherichia coli*, *Klebsiella pneumoniae*, *Klebsiella oxytoca* and *Enterobacter cloacae*)[21], all of which were isolated from preterm infant stool (Fig. 3). Each CFS showed inhibitory activity against at least 3 pathobionts, with the majority suppressing growth of all 4 (Fig. 3a,b). CFS from the probiotic *B. infantis* had the strongest inhibitory activity, with growth of all pathobionts reduced to <2% of their no-treatment controls. *C. tertium* and 3 *C. perfringens* CFSs showed a similar pattern of high inhibition and clustered with *B. infantis* on the basis of CFS inhibitory capacity (Fig. 3a). These 4 *Clostridium* CFSs reduced each pathobiont's growth to ≤55% of their no-treatment controls (all $P < 0.001$). *K. pneumoniae* was most susceptible to the 4 *Clostridium* and *B. infantis* CFSs, with growth reduced by each to <15% of the no-treatment control (all $P < 0.001$).

All CFSs were found to be weakly acidic, with mean pH ranging from 6.64 to 4.36 (Supplementary Data Fig. 3a). In addition, all strains

**Fig. 2 | *Clostridium* spp. produced wider varieties and higher quantities of beneficial metabolites compared with *Bifidobacterium* spp. a**, Individual SCFA production by species grown on individual HMOs ($n = 38$). SCFA profiling was performed on culture supernatants collected following the HMO growth assay shown in Fig. 1a. Per strain, the HMOs used were those on which growth was seen (as indicated in Fig. 1a). SCFA concentrations in µg ml⁻¹ were divided by the maximum $OD_{600}$ recorded for each strain to provide growth-adjusted concentrations. These adjusted concentrations for each strain were then combined into a single boxplot per species. Statistical comparisons were performed using ANOVA, followed by adjustment for multiple comparisons using two-tailed Tukey's HSD method. Conditions with the same letters are not significantly different. **b**, Total unadjusted concentration of SCFAs in culture supernatants of *Bifidobacterium* and *Clostridium* spp. grown on individual HMOs. The raw SCFA concentrations for each strain were averaged per species. **c**, Principal component analysis (PCA) of untargeted metabolomics data for CFSs generated from bacterial cultures growing on cocktails of HMOs. Dashed

arrows indicate the top 5 metabolites by loadings magnitude. Stars indicate centroids for groupings. **d**, Metabolites detected in CFSs of *C. perfringens pfoA⁻* (AM1) growing on a cocktail of the HMOs 6'-SL, LNnT and 2'-FL, compared with blank ZMB1 medium. A positive log2(fold change) indicates production of metabolites by the strain, while a negative fold change indicates metabolite depletion. Metabolites of interest are highlighted and numbered. Statistical significance was calculated using two-tailed moderated *t*-test, followed by adjustment for multiple comparisons using the Benjamini–Hochberg method. **e**, Levels of tryptophan catabolites for CFSs generated from *Bifidobacterium* and *Clostridium* spp. during growth on strain-specific cocktails of HMOs that we found they can utilize (see Fig. 1a for strain HMOs) ($n = 30$). Statistical comparisons were performed using ANOVA, followed by adjustment for multiple comparisons using two-tailed Tukey's HSD method. Conditions with the same letters are not significantly different. For all boxplots: centre line, median; box limits, upper and lower quartiles; whiskers, 1.5× interquartile range. BHBA, beta-hydroxybutyric acid.

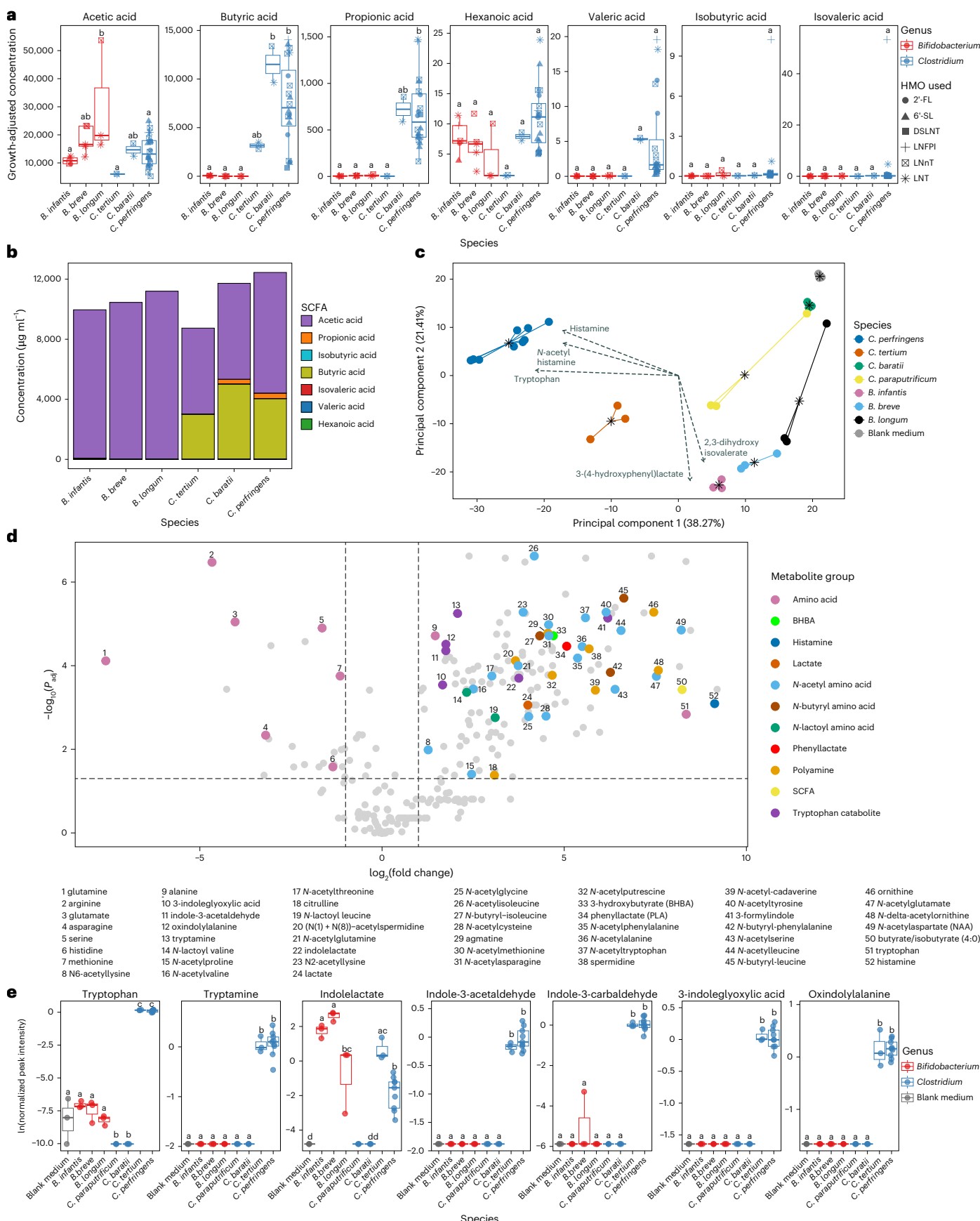

used to generate CFSs are prolific producers of acetate and, in the case of *Clostridium*, also butyrate (Fig. 2c). We therefore speculated that pH or SCFAs may be mediating the inhibitory effects seen. Pathobiont growth was tested with an acetate:butyrate mix, blank ZMB1 medium and *CP-pfoA⁻* CFS, all adjusted to multiple pHs. Acidic blank medium inhibited growth, but not to the same extent as CFSs, while acidic SCFAs matched or exceeded the inhibitory activity of CFSs (Fig. 3c). Neutralized SCFAs also inhibited 3 pathobionts, but to a lesser extent, while neutralized *CP-pfoA⁻* CFS lost its activity. The most pH sensitive of the pathobionts, *E. coli* (Fig. 3c), was then used to test the effect of adjusting pH on all CFSs. Only 3 CFSs, including *CP-pfoA⁻*, lost their activity when neutralized, while the remainder had reduced activity (Supplementary Data Fig. 3b). The loss of inhibition following *CP-pfoA⁻* CFS neutralization suggests pH dependence, probably mediated by pH-sensitive metabolites, such as SCFAs. The similar reduction in inhibitory effects observed with neutralized SCFA mix and *CP-pfoA⁻* CFS further supports this role. However, acidic blank medium was less inhibitory than *CP-pfoA⁻* CFS, hence pH is not the sole factor and additional pH-sensitive antimicrobial factors probably also contribute to the inhibitory effect.

Within the neonatal gut, it is critical that beneficial bacteria are not inhibited during suppression of pathobiont growth. The CFS of *CP-pfoA⁻* was selected to determine its impact on *Bifidobacterium* species growth owing to it being in the neonatal-derived hypovirulent lineage V[22], lacking the *pfoA* toxin gene, having the ability to use health-associated DSLNT[5–7], and its production of several beneficial SCFAs and metabolites. *CP-pfoA⁻* CFS derived from growth on glucose was found to significantly enhance the growth of naturally occurring *B. breve* and *B. longum* across all time points (all $P < 0.05$; Fig. 3d,e). However, a significant impact on the growth of the probiotic (that is, not naturally occurring) *B. infantis* was found after 48 h ($P < 0.01$; Fig. 3d,e). Notably, because the *CP-pfoA⁻* CFS used in these experiments was derived following growth on glucose, the promotion of *B. breve* and *B. longum* is due to microbial metabolites and not cross-feeding of HMO degradation by-products.

### *Clostridium* CFSs dampened inflammation in an intestinal organoid model

Diet–microbe–host interaction in the preterm gut is critical in understanding health and disease and developing effective therapies[39]. We examined the potential toxicity of the CFSs collected following growth on HMOs on the human gut using Caco-2 cells before exposing preterm infant-derived intestinal organoids (PIOs). When added to a concentration of 25% v/v, the majority of *Clostridium* spp. and *Bifidobacterium* spp.-derived CFSs did not reduce Caco-2 viability below 50% (Extended Data Fig. 5). This concentration was therefore selected for subsequent experiments. While there was strain-to-strain variability, *Clostridium* CFSs were less toxic than *Bifidobacterium* CFSs at 25% and 50% v/v. Notably, *CP-pfoA⁻* CFS at 10% and 25% v/v maintained full Caco-2 viability (Extended Data Fig. 5).

The preterm gut epithelium was modelled using PIO monolayers, within an anaerobic co-culture system (Fig. 4a)[40]. PIOs were treated with selected CFSs in isolation and in the presence of inflammatory stimuli (lipopolysaccharide (LPS) and flagellin). *CP-pfoA⁻* CFS was tested alongside CFS from hypervirulent lineage I *C. perfringens* JC26 (*CP-pfoA⁺*), *C. tertium* (related HMO-using *Clostridium* spp.) and *B. infantis* (commercially available probiotic strain).

PIOs lack immune cells but secrete cytokines that can be used as a proxy for immunomodulation. Unsupervised ordination of basolateral cytokine data showed that overall cytokine profiles of PIOs treated with LPS and flagellin ('stimuli only') were significantly different from those treated with CFS only regardless of the specific strain (all principal component 1 (PC1) $P < 0.05$; Fig. 4b). In the presence of stimuli, only treatment with 25% v/v CFS resulted in cytokine profiles that were significantly different from those with 'stimuli only' (PC1 $P = 0.014$) and comparable to those with 'CFS only' controls (PC1 $P = 0.23$; Fig. 4b). This trend was not observed in the apical cytokine data (Fig. 4c).

All CFSs significantly inhibited stimuli-induced basolateral secretion of IL-8 (all $P < 0.05$; Fig. 4d). Only *CP-pfoA⁻* also significantly inhibited IL-8 apically ($P < 0.001$), while *C. tertium* induced significantly greater apical secretion ($P < 0.001$). Changes in secretions of a further 4 inflammatory cytokines were also detected (Extended Data Fig. 6a–d). Only *CP-pfoA⁻* and *CP-pfoA⁺* inhibited basolateral secretion of all 4, with only *CP-pfoA⁻* also significantly inhibiting apical secretion of both CCL7 ($P < 0.001$; Extended Data Fig. 6b) and CCL2 ($P < 0.001$; Extended Data Fig. 6d). *C. tertium* ($P < 0.001$) and *CP-pfoA⁺* ($P = 0.02$) induced significant increases in apical TNF. We confirmed that the inhibitory effects observed were not caused by the ZMB1 medium used to produce the CFSs, except in the case of apical CCL2 ($P = 0.035$; Extended Data Fig. 7). Nevertheless, the fold change was lower than with *CP-pfoA⁺*, *B. infantis* and *CP-pfoA⁻*, indicating that components of ZMB1 media were not solely responsible. Except for basolateral IL-8, where inhibition was reduced at neutral pH, inhibition patterns were comparable between acidic and neutral pH *CP-pfoA⁻* CFSs, confirming that the dampening of pro-inflammatory cytokines was not due to the acidic pH of the CFSs (Extended Data Fig. 8).

We also tested whether CFSs at 25% v/v alone could induce inflammatory cytokine secretion compared to PIO medium only. Both *CP-pfoA⁻* and *B. infantis* CFSs did not induce any significant IL-8 secretion, but *CP-pfoA⁺* ($P = 0.03$) and *C. tertium* ($P < 0.001$) both triggered apical IL-8 production (Fig. 4e). For the other 4 cytokines measured, *C. tertium* induced apical CXCL5 ($P = 0.026$), TNF ($P < 0.001$) and CCL7 ($P < 0.001$), *CP-pfoA⁺* induced apical TNF ($P = 0.017$), *B. infantis* induced apical CCL7 ($P = 0.04$), while *CP-pfoA⁻* did not induce secretion of any cytokine apically or basolaterally (Extended Data Fig. 9a–d). Thus, overall only *CP-pfoA⁻* CFS had no pro-inflammatory impact of its own.

### The presence of *pfoA* determines the impact of *C. perfringens* on the intestinal epithelium

Aside from CFS, we next assessed the impact of live *CP-pfoA⁻* combined with inflammatory stimuli on PIOs. Apical secretion of CCL2, CCL7 and CXCL5 and basolateral secretion of IL-8 were all significantly reduced in the presence of live *CP-pfoA⁻* (all $P ≤ 0.005$; Fig. 5a), indicating similar activity to the CFS. *CP-pfoA⁺* and CP-*pfoA⁻* share 4 toxin and 2 colonization genes, with *CP-pfoA⁻* encoding a further 7 colonization factors (Fig. 1c). Proteomics confirmed that *CP-pfoA⁺* CFS contained PfoA, while it was absent from *CP-pfoA⁻* (Fig. 5b). Toxicity assays in Caco-2 cells showed *CP-pfoA⁺* CFS reduced cell viability to 49%, compared

**Fig. 3 | *Clostridium* spp. CFSs suppressed pathogen growth without impacting naturally occurring *Bifidobacterium* spp. growth. a**, Growth of pathobionts in ZMB1 supplemented with glucose with CFSs from 6 *Clostridium* and 3 *Bifidobacterium* isolates. Values represent area under the curve (AUC) for growth in media supplemented with CFS as a percentage of the control AUC. Statistical comparisons were performed using ANOVA, followed by Dunnett's test to adjust for multiple comparisons, whereby the growth of each strain following addition of pH7 ZMB1 medium was used as the control. **b**, Growth curves of pathobionts in ZMB1 supplemented with glucose and treated with AM1 CFS. **c**, Impact of acidic pH and/or the presence of the SCFAs acetate and butyrate on pathobiont growth.

Values represent AUC for growth as a percentage of the control AUC. Statistical comparisons were performed using ANOVA, followed by Dunnett's test to adjust for multiple comparisons, whereby the growth of each strain following addition of pH7 ZMB1 medium was used as the control. **d**, Growth of *Bifidobacterium* isolates in ZMB1 supplemented with glucose and AM1 CFS. Values represent AUC for growth in media supplemented with CFS as a percentage of the control AUC. Statistical comparisons were performed using ANOVA, followed by Dunnett's test to adjust for multiple comparisons, whereby the growth of each strain following addition of pH7 ZMB1 medium was used as the control. **e**, Growth curves of *Bifidobacterium* spp. in ZMB1 supplemented with glucose and treated with AM1 CFS.

with 99% for *CP-pfoA⁻* ($P < 0.001$; Fig. 5c). In PIO monolayers under aerobic conditions, *CP-pfoA⁺* CFS increased non-mitochondrial oxygen consumption ($P < 0.001$), reduced ATP production ($P = 0.054$) and increased proton leak ($P < 0.001$) (Fig. 5d and Supplementary Data Fig. 4). *CP-pfoA⁻* CFS also increased proton leak to a similar degree

($P < 0.001$), but simultaneously enhanced basal respiration ($P < 0.001$), maximal respiration ($P < 0.001$), ATP production ($P = 0.012$) and spare respiratory capacity ($P = 0.002$). Overall, this analysis revealed that *CP-pfoA⁺* CFS and *CP-pfoA⁻* CFS reduced and increased mitochondrial bioenergetic function, respectively.

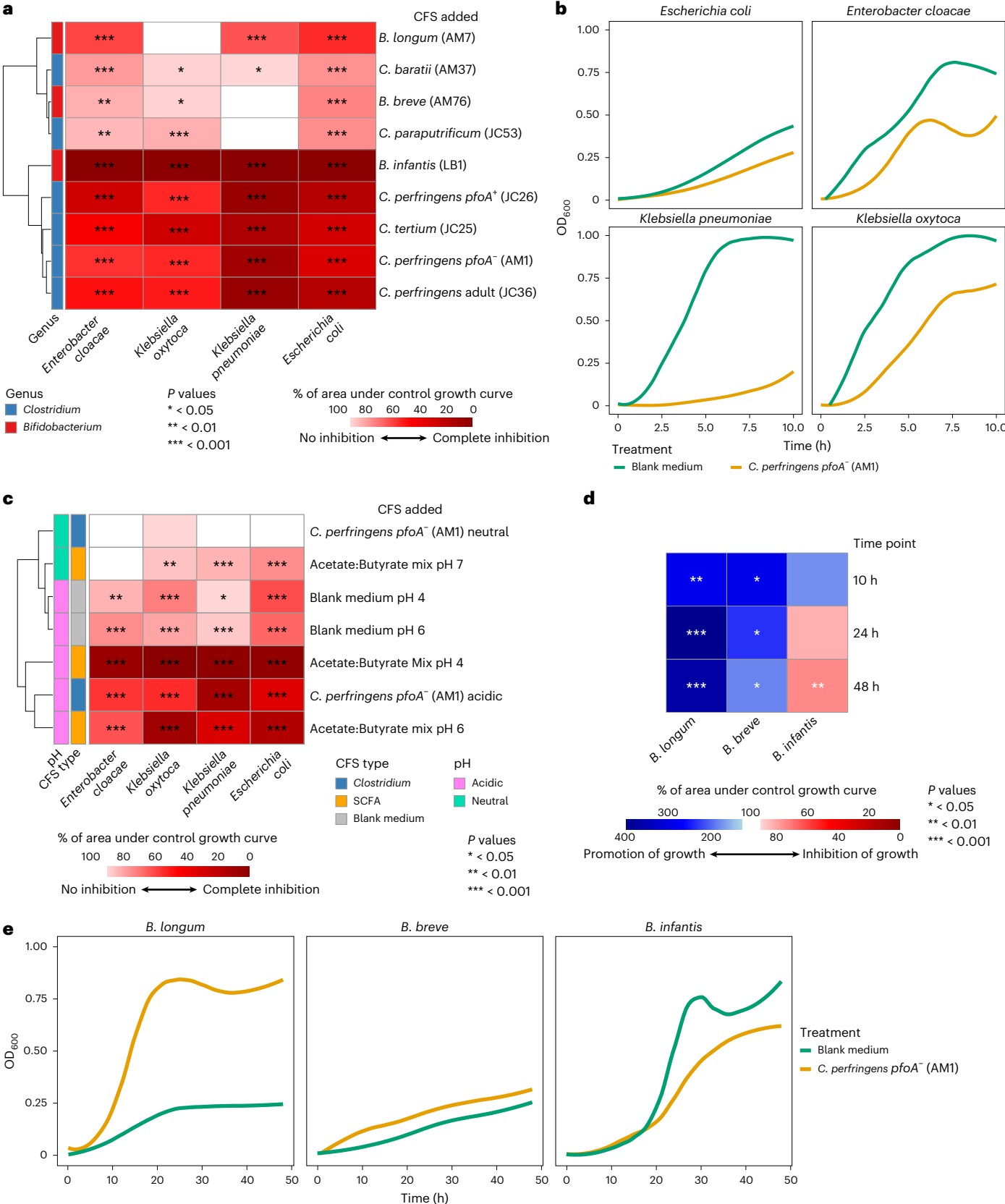

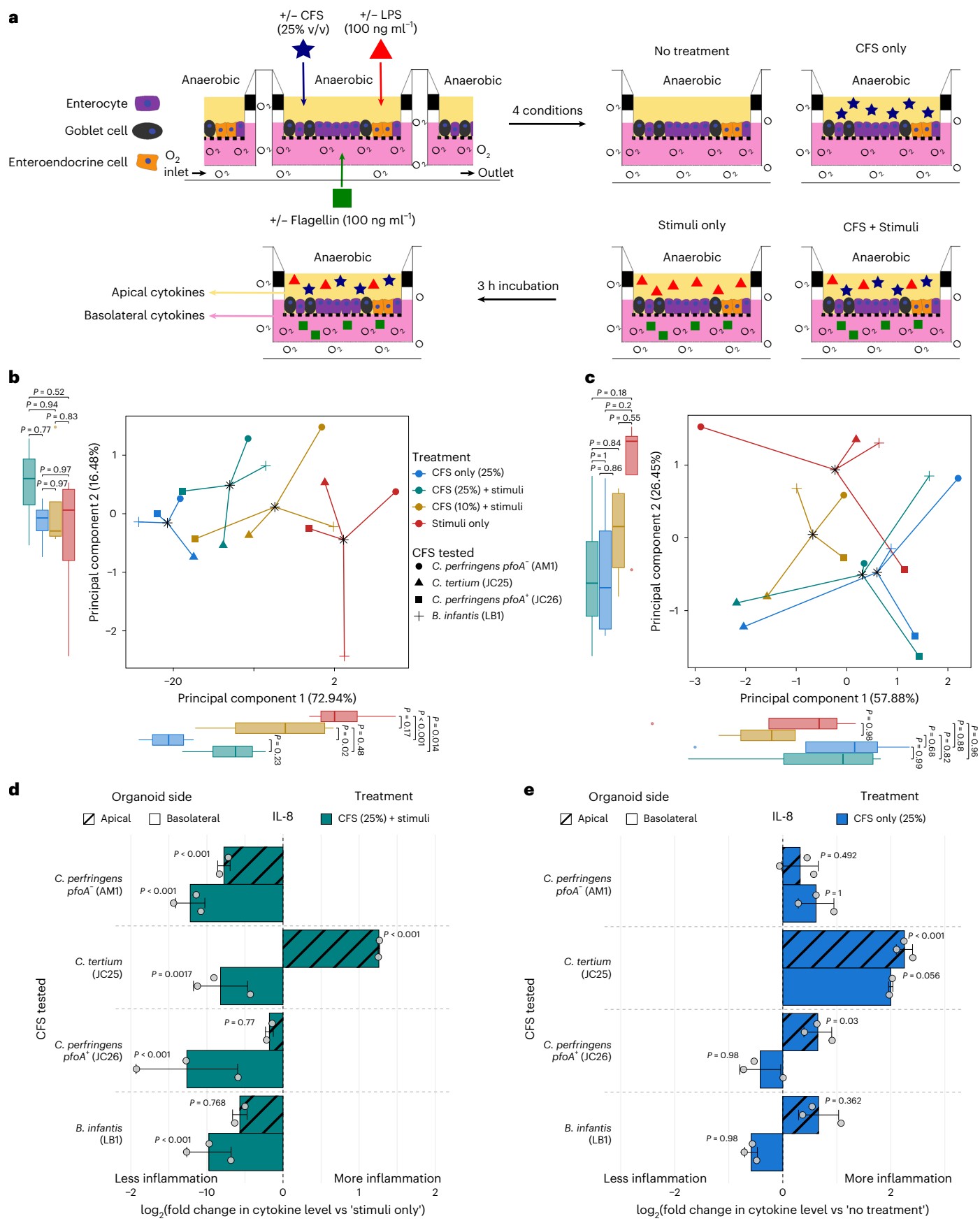

**Fig. 4 | *Clostridium* spp. and *Bifidobacterium* spp. CFSs dampened inflammation in a preterm intestinal-derived organoid co-culture model.** **a**, Schematic of organoid monolayer inflammation assays. **b**,**c**, Principal component analysis (PCA) of basolateral (**b**) and apical (**c**) cytokine (IL-8, TNF, CCL2, CCL7 and CXCL5) profiles ($n = 60$, 20 conditions in triplicate). The data were converted to fold changes compared to the negative control and $\log_2$ transformed. There was no apical CXCL5 data for the experiment with the AM1 CFS so this cytokine was removed from apical analysis. For all boxplots: centre line, median; box limits, upper and lower quartiles; whiskers, 1.5× interquartile range. Statistical comparisons were performed using ANOVA, followed by adjustment for multiple comparisons using two-tailed Tukey's HSD method. **d**, $\log_2$ fold change of apical and basolateral IL-8 secretion during 'CFS + stimuli' treatment compared to 'stimuli only'. *P* values represent the differences between the unprocessed detected cytokine levels. Data are presented as mean ± s.d. Statistical comparisons were performed using ANOVA, followed by Dunnett's test to adjust for multiple comparisons, whereby cytokine secretion from 'stimuli only' was used as the control. **e**, $\log_2$(fold change) of apical and basolateral IL-8 secretion during 'CFS only' treatment compared to 'no treatment'. *P* values represent the differences between the unprocessed detected cytokine levels. Data are presented as mean ± s.d. Statistical comparisons were performed using ANOVA, followed by Dunnett's test to adjust for multiple comparisons, whereby cytokine secretion from 'no treatment' was used as the control. In **d** and **e**, $n = 4$–6 to calculate fold changes per CFS tested.

We next sought to compare microbe–microbe and microbe–host interaction with live co-culture of *CP-pfoA*⁺ and CP-*pfoA*⁻ on PIOs. We observed that *CP-pfoA*⁺ had a faster growth rate than *CP-pfoA*⁻ (Fig. 5e). Cytokine secretion from PIOs treated with live *CP-pfoA*⁻ and *CP-pfoA*⁺ in the anaerobic co-culture system was then assessed against untreated controls. Live *C. perfringens* significantly reduced apical CCL7, CXCL5 and CCL2 secretion to below basal levels, regardless of *pfoA* status (all $P \le 0.001$; Fig. 5f). Live *CP-pfoA*⁺ halved basolateral IL-8 ($P = 0.005$) and more than quadrupled apical MIF, IL-1RA and IL-18 (all $P < 0.001$). *CP-pfoA*⁻ also increased MIF ($P = 0.037$), but to lower levels than *CP-pfoA*⁺, and had no significant impact on any other cytokine compared to basal media.

Given the contrasting results between potentially beneficial *CP-pfoA*⁻ and pathogenic CP-*pfoA*⁺, we next investigated whether pre-colonization of PIOs with *CP-pfoA*⁻ could protect against *CP-pfoA*⁺ damage. This pre-colonization with *CP-pfoA*⁻ significantly reduced the levels of MIF, IL-1RA and IL-18 that are elevated by *CP-pfoA*⁺ (all $P \le 0.002$; Extended Data Fig. 10a–c). Finally, live *CP-pfoA*⁺ also significantly compromised barrier integrity ($P < 0.001$), which was prevented by pre-colonization with *CP-pfoA*⁻ ($P = 0.003$; Fig. 5g). In isolation, *CP-pfoA*⁻ increased barrier integrity, with trans-epithelial electrical resistance (TEER) values similar to those of untreated PIOs ($P = 0.906$).

## Discussion

Using preterm stool bacterial isolates, we discovered that numerous *Clostridium* species from the preterm infant gut microbiome can metabolize HMOs. Since the publication of the current study as a preprint[41], other work focused on 2′-FL has reported HMO metabolism by *C. perfringens*[42,43]. *C. perfringens* showed strain-to-strain variation in HMO use, independent of *pfoA* status, but only a *pfoA*⁻ strain could use DSLNT, a health-associated HMO[5–7]. Compared with *Bifidobacterium* spp., *Clostridium* spp. produced a broader range of potentially beneficial

metabolites including but not limited to tryptophan catabolites and SCFAs, especially butyrate which attenuates experimental NEC in rats[44]. A limitation of the untargeted metabolomics data shown here, however, is that only relative intensities were obtained for each metabolite, which prevents direct comparison with published quantitative datasets. *CP-pfoA*⁻ CFS showed inhibitory effects on common infant pathobionts, generally promoted growth of beneficial microbes, enhanced mitochondrial bioenergetic function in preterm intestine-derived organoids, and suppressed the inflammatory response in an organoid co-culture model. When using live microbes to study competitive exclusion, *CP-pfoA*⁻ protected against CP-*pfoA*⁺-mediated damage to epithelium barrier integrity and suppressed the pro-inflammatory activity of the pathogenic strain.

*C. perfringens* are especially prevalent in newborns[22,45], which may explain their ability to utilize HMOs. Given that evolution has favoured HMOs to be highly abundant in human milk, early-life colonizers that use these prebiotics have been proposed as likely having therapeutic potential, as seen with probiotics containing HMO-using *Bifidobacterium* spp.[46–48]. Indeed, in the current work we show higher prevalence of *Clostridium* in healthy preterm infants from a recent UK-wide study, which is consistent with observations at the class level from another US-based study[49]. In contrast, some studies have linked the relative abundance of *Clostridium* to NEC[50,51]. Notably, these studies were relatively small (that is, 8–11 NEC cases) and relied on amplicon sequencing that is limited to the genus level. Recent mechanistic strain-level work employing murine models showed that *pfoA*⁺ *C. perfringens* caused significantly greater cellular damage than *pfoA*⁻ *C. perfringens*[21]. Strains that lacked *pfoA* comprised their own phylogenetic hypovirulent or 'commensal-like' lineage V and were considered to not encode the necessary virulence traits required to cause NEC[22]. Similar findings have been reported in other inflammatory conditions including paediatric inflammatory bowel disease[24]. Adding to previous

**Fig. 5 | *Clostridium perfringens* showed strain-specific impacts on PIO monolayers during live co-culture depending on *pfoA* carriage. a**, $\log_2$(fold change) in apical and basolateral cytokine secretion during combined treatment with live *C. perfringens pfoA*⁻ (AM1) for 3 h, with inflammatory stimuli added at 1 h, compared to 'stimuli only' ($n = 4$–6 to calculate fold changes). *P* values represent the differences between the raw cytokine levels. Data are presented as mean ± s.d. Statistical comparisons were performed using ANOVA, followed by Dunnett's test to adjust for multiple comparisons, whereby cytokine secretion from 'stimuli only' was used as the control. **b**, Proteomic detection of PFOA in the *C. perfringens pfoA*⁻ (AM1) and *C. perfringens pfoA*⁺ (JC26) CFSs used in co-culture experiments, with GAPDH provided for reference. **c**, MTS cell viability assay data for Caco-2 cells treated with 25% v/v CFSs from *C. perfringens pfoA*⁻ (AM1) and *C. perfringens pfoA*⁺ (JC26) strains ($n = 3$ per group). Values are shown as % of the viability measured for Caco-2 cells incubated in DMEM. Data are presented as mean ± s.d. Statistical comparison was performed using unpaired two-tailed *t*-test. **d**, Summary of changes to mitochondrial energetic function induced by CFSs, quantified as changes to the OCR of cells. Detailed quantification of changes to each parameter and associated statistical analyses are shown

in Supplementary Data Fig. 4. **e**, Growths of *C. perfringens pfoA*⁻ (AM1) and *C. perfringens pfoA*⁺ (JC26) growing in ZMB1 medium supplemented with glucose ($n = 3$ per timepoint per strain). Data are presented as mean ± s.d. Per timepoint, statistical comparisons were performed using unpaired two-tailed *t*-test. **f**, Changes in apical and basolateral cytokine secretion during 3 h treatment with either live *C. perfringens pfoA*⁻ (AM1), live *C. perfringens pfoA*⁺ (JC26) or live *C. perfringens pfoA*⁻ (AM1) for 3 h + *C. perfringens pfoA*⁺ (JC26) added at 1 h ($n = 4$–6 to calculate fold changes, per treatment). Data converted to fold change compared to 'no treatment' and $\log_2$ transformed for plotting. *P* values represent the differences between the raw cytokine levels. Data are presented as mean ± s.d. Statistical comparisons were performed using ANOVA, followed by Dunnett's test to adjust for multiple comparisons, whereby cytokine secretion from 'no treatment' was used as the control. **g**, Change in barrier integrity as measured by TEER during organoid incubation with live *C. perfringens* strains ($n = 2$ per group). Conditions with the same letters are not significantly different. Data are presented as mean ± s.d. Statistical comparisons were performed using ANOVA, followed by adjustment for multiple comparisons using two-tailed Tukey's HSD method. n.d., not detected.

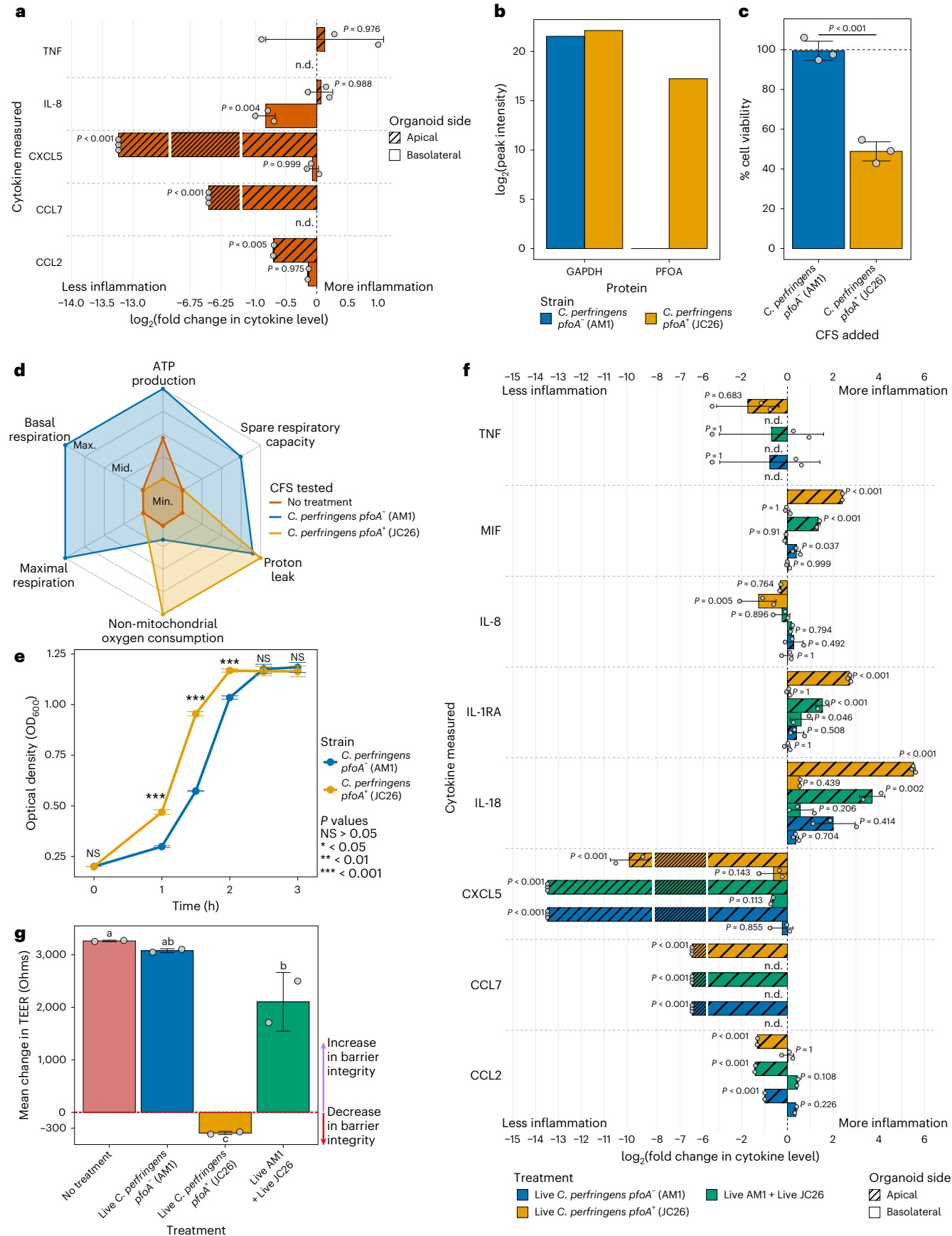

in vivo work, our human-centric and preterm-relevant experiments further demonstrated that both live $CP\text{-}pfoA^-$ and its CFS appeared to have no pro-inflammatory impact of their own within our PIO model, despite $CP\text{-}pfoA^-$ still encoding other known $C.\ perfringens$ toxins. Taken together, this underscores the need for strain-level microbiome data and highlights the beneficial potential of $CP\text{-}pfoA^-$.

While no consistent agent has been associated with NEC in preterm infants, *Klebsiella* has been implicated in several studies, including the largest current analysis of NEC infant stool that employed metagenomics[25,52]. *C. perfringens, C. tertium* and *B. infantis* CFSs showed inhibition of *K. oxytoca* and *K. pneumoniae*, as well as other pathobionts. In the case of $CP\text{-}pfoA^-$, this was dependent on acidic pH and likely mediated by SCFAs, although other mechanisms, such as production of bacteriocins, may also be involved. $CP\text{-}pfoA^-$ further promoted the growth of naturally occurring infant *B. breve* and *B. longum*, but not commercial probiotic-derived *B. infantis* of unknown origin. In industrialized nations, *B. breve* and *B. longum* are the predominant *Bifidobacterium* in infants[2,21,53] and were notable for their strong growth on LNT. Cross-feeding with *C. perfringens*, which metabolized DSLNT into LNT, could therefore be an important contributor to promoting *B. breve* and *B. longum* colonization in the infant gut.

Probiotics appear to reduce NEC in preterm infants, although precise mechanisms remain unclear[54]. Competitive exclusion, whereby two species cannot coexist if they have identical niches, may be one mechanism. We found that previous colonization of PIOs with $CP\text{-}pfoA^-$ can protect the intestinal epithelium from $CP\text{-}pfoA^+$-mediated damage. Furthermore, CFS from $CP\text{-}pfoA^-$ had positive roles in modulating the gut microbiome (inhibiting pathobionts and promoting *Bifidobacterium*), improving mitochondrial function (spare respiratory capacity and respiration rate) and suppressing inflammation, warranting further exploration for novel microbial therapies based on $pfoA^-$ *C. perfringens*. On the other hand, we corroborate and extend previous work by showing that human intestinal cells treated with PfoA-containing CFS from $CP\text{-}pfoA^+$ had a ~50% reduction in viability and showed signs of mitochondrial damage[22,55]. Such evidence cautions against supplementing HMOs to infants colonized with $pfoA^+$ *C. perfringens*, which is especially important as several commercial formula products now include selected synthetic HMOs in term formula.

In summary, we report that a range of *Clostridium* species use HMOs in the preterm gut, producing a broad range of SCFAs, tryptophan catabolites and other potentially beneficial immunomodulatory metabolites that positively influence host physiology. Thus, $CP\text{-}pfoA^-$ may play an important and previously unrecognized role in gut microbiome-mediated immune education during early life. However, where infants are potentially colonized with pathogenic $pfoA^+$ *C. perfringens*, these results caution against widespread HMO supplementation.

## Methods

### Ethics and sample collection

Preterm infants (born at <32 weeks gestation) were born or transferred to a single tertiary-level Neonatal Intensive Care Unit (NICU) in Newcastle upon Tyne, United Kingdom, and participated in the Supporting Enhanced Research in Vulnerable Infants (SERVIS) study (REC10/H0908/39) after written informed parental consent. The study protocol was approved by Newcastle Hospitals NHS Foundation Trust (NUTH), NRES Committee North East and N Tyneside 2. Parents were approached in the first week of life when the study was explained by a member of the research team. Parents were given the option to opt in or out of each specific aspect on a single consent form. Parents of infants who were initially extremely unwell were only approached when they were considered stable by the bedside nurse and medical team. Approaches were made by experienced neonatal staff familiar with the studies being described, sample collection and parental communication. Written signed consent was obtained after the parents have had time to consider the information. Stool samples were regularly collected from nappies/diapers of preterm infants into sterile collection pots by nursing staff. Breast milk samples were collected from residuals from an infant's feeding systems. Samples were initially stored at −20 °C before being transferred to −80 °C for long-term storage. Intestinal tissue samples used to generate organoid cell lines were salvaged following surgical resection. Participants were not compensated for donation.

### Bacterial isolation and identification

Stool samples were thawed on ice and initially diluted roughly 1:10 w/v in sterile anaerobic phosphate buffered saline (PBS). Tenfold serial dilutions were then performed using sterile anaerobic PBS and various dilutions (typically $10^{-2}$ and $10^{-4}$) were cultured by adding 100 µl of inoculum onto an agar plate before spreading to cover the surface of the agar plate and incubating for up to 96 h. Numerous different agar media were used including brain heart infusion (BHI), transoligosaccharide propionate (TOS), *Bifidus* selective medium (BSM), De Man, Rogosa and Sharpe (MRS), fastidious anaerobe agar (FAA) and yeast extract-peptone-dextrose (YPD). Unique appearing colonies based on morphology, colour and size were subcultured twice. Full-length 16S rRNA gene sequencing (27F 5′-AGAGTTTGATCCTGGCTCAG3′; 1492R 5′-GGTTACCTTGTTACGACTT-3′) and matrix-assisted laser desorption ionization−time of flight mass spectrometry (Bruker MALDI−TOF MS) of single fresh colonies were used to initially identify isolates to genus or species level. The *B. infantis* strain LB1 was isolated and identified from a sample of the probiotic product Labinic (Biofloratech) using the same methodology described above. Isolates were added to glycerol for long-term storage at −80 °C. The isolates studied herein were derived from at least 15 infants. We cannot be certain of the number above this due to use of some samples where the tube label was compromised. This was decided on the basis that reference to the individual infants or their clinical data would not be needed, which is the case, therefore this does not impact the results in any way.

### HMOs

The HMOs 2′FL, DSLNT, LNT, LNnT, LNFPI and 6′SL were manufactured to analytical grade and kindly donated by dsm-firmenich through the HMO Donation Program.

### Whole-genome sequencing and genomic analysis

Glycerol stocks of 11 *C. perfringens* strains were streaked out on BHI agar plates and incubated in an anaerobic chamber at 37 °C overnight. Single colonies were then picked from these plates, transferred to 5 ml BHI broth and incubated, with shaking at 110 r.p.m., in the anaerobic chamber at 37 °C for 48 h. Cultures were then centrifuged at 5,000 *g* for 10 min at 4 °C and the supernatants removed. Genomic DNA was extracted using the MasterPure Complete DNA and RNA Purification kit (Lucigen). Cell pellets were resuspended in 500 µl PBS, centrifuged at 5,000 *g* for 5 min and the supernatant removed. Pellets were then resuspended in 300 µl proteinase K master mix, made up following manufacturer protocol using stocks from the kit, and incubated in a heat block at 65 °C for 15 min, with 10 s of vortexing every 5 min. The incubation then continued for a further 45 min. Tubes were cooled to room temperature and 2 µl RNase A from the kit was added to each. Samples were incubated at 37 °C for 1 h and then cooled on ice for 5 min. Kit MPC protein precipitation reagent (150 µl) was added to each tube, followed by vortexing for 10 s and centrifuging at 5,000 *g* (4 °C, 10 min). Supernatants were transferred to fresh tubes and 500 µl 2-propanol added to each, with mixing done by inverting 40 times. Samples were then centrifuged at 5,000 *g* (4 °C, 10 min) and the supernatants removed. DNA pellets were then washed by adding and removing 1 ml 70% ethanol twice. Tubes were then left open in a laminar flow hood to allow any residual ethanol to evaporate and the DNA pellets to dry. Nuclease-free water (40 µl) was then added, with each then briefly vortexed and left to resuspend at 4 °C overnight. Extracted DNA was then transferred to −20 °C for storage.

All 11 *C. perfringens* isolates were sequenced on the NovaSeq 6000 system (2 × 151 bp) at the Wellcome Sanger Institute. Genomes were assembled using SPAdes[56] and draft assembly genomes were quality checked using checkm (v.1.1.3)[57] and GUNC (v.1.0.5)[58], ensuring ≥90% completeness and ≤5% contamination before taxonomic assignment (species-level assignment) via gtdb-tk (v.2.3.2)[59]. Draft genome assemblies were then annotated using prokka (v.1.14)[60], with GFF-annotated files being used for input for core gene alignment construction via panaroo (v.1.2.8)[61]. The core gene alignment generated was used to construct a phylogenetic tree using IQ-TREE (v.2.0.5)[62] together with 673 public genomes published previously for lineage assignment purpose[22]. The tree was visualized using iTOL (v.6.0)[63]. Toxin genes and colonization factors of *C. perfringens* isolates were screened computationally using ABRicate v.1.0.1 (https://github.com/tseemann/abricatev) via TOX-Iper sequence database (https://github.com/raymondkiu/TOXIper). Genome sizes were calculated using sequence-stats v.1.0 (https://github.com/raymondkiu/sequence-stats). A mash-distance sequence tree comprising solely 11 *C. perfringens* isolate genomes was generated via Mashtree (v.1.2.0)[64] with default parameters. The distance tree was mid-point rooted and visualized in iTOL (v.6.0)[63].

### Isolate growth curves

All isolates were grown overnight in BHI, except for bifidobacteria which were grown in MRS supplemented with L-cysteine HCl (0.05% w/v). The chemically defined medium Zhang Mills Block 1 (ZMB1) was selected for use for this work, as it can support the high-density growth of a wide variety of organisms (for example, *Clostridium*, *Bifidobacterium*, *Lactobacillus*, *Klebsiella*, *Bacteroides* and so on) and is well suited to downstream analytical applications, such as metabolomics[65]. ZMB1 without glucose was prepared, and various sugars were tested in 1% w/v concentration: base ZMB1 medium, glucose, lactose, 2′FL, DSLNT, LNT, LNnT, LNFP I, 6′SL. All isolates could not be tested on DSLNT due to limited supply of this HMO. The overnight growth for each isolate was centrifuged at 5,000 *g* for 5 min at 4 °C, and the pellet was resuspended in the same amount of anaerobic PBS. Of the growth resuspended in PBS, 20 μl was spiked in 180 μl of media, and the growth was measured in a Cerillo Stratus plate reader for 150 h. All isolates were tested in triplicate and wells with media only were included in each plate to check for contamination.

### Analysis of *Clostridium* and *Bifidobacterium* prevalences and abundances in an existing preterm infant microbiota dataset

The previously published MAGPIE Study recruited preterm infants from NICUs across the United Kingdom and generated microbiota data using 16S rRNA gene sequencing (V4 region)[23]. A total of 34 samples were available before NEC diagnosis along with 306 healthy controls (no NEC diagnosis) from infants cared for at NICUs reporting no regular probiotic use during the MAGPIE study. Details of microbial DNA extraction, library preparation and sequence data processing are available in the original study manuscript. Feature counts of bacterial operational taxonomic units in the original MAGPIE data were merged at the genus level to determine the prevalence and relative abundance of *Clostridium* and *Bifidobacterium* in preterm infants. One sample per participant was selected across predefined weekly timepoints based on day of life (DOL): 0–7, 8–14, 15–21 and 22+. Where multiple samples were available for a single patient at any given timepoint, those from the earlier DOL were selected. We used only samples before NEC diagnosis to avoid bias from NEC treatment (for example, antibiotics) and, given that the median day of NEC onset is day 20, we grouped all samples from week 4 onwards. Samples from NICUs reporting regular probiotic use were excluded to avoid bias arising from probiotic supplementation with bifidobacterial spp. A two-sided *z*-test was used to test for differences in the proportion of samples where *Clostridium* or *Bifidobacterium* was present between infants diagnosed with NEC and controls using the 'prop.test' function in R[66]. Generalized linear models

were used to test for differences in relative abundance of *Clostridium* and *Bifidobacterium* between NEC and controls using the 'glmmTMB' function in R. NICU site and DOL were included in models as fixed effects. Infant identifiers were included as random effects. Model estimates of relative abundance were compared between NEC and control infant groups with the 'anova' function in R. Prevalence delta for both genera was calculated at each timepoint as the difference between the proportion of control and NEC samples containing each genus.

### Analysis of *pfoA⁻ C. perfringens* prevalence in existing preterm infant microbiota dataset

Reference [25] previously assembled microbial genomes (MAGs) from metagenomic sequencing of 1,163 stools from 160 preterm infants (NEC = 32, control = 126). Details of microbial DNA extraction, library preparation and MAG generation are available in the original study manuscript. We downloaded MAGs and associated taxonomic annotations, revealing 109 unique *C. perfringens* MAGs, present in 50% of infants. A custom blastN database containing two variants of the *pfoA* gene[67,68] was generated and used to screen each *C. perfringens* MAG for the presence of the gene[69]. BlastN (v.2.16.0) parameters included a minimum word size of 100 nucleotides, a max *E*-value of 0.1 and constrained each query sequence to a single reference sequence. To ensure that *pfoA* presence or absence was not confounded by metagenomic sequencing effort, we explored correlations between CheckM[57] assembly completeness scores and counts of reads mapping to each assembly using simple linear regression with the 'lm' function in R. We compared *pfoA⁻ C. perfringens* prevalence (binary presence/absence, per infant) in NEC infants (*n* = 32) within stools collected before diagnosis of NEC, to prevalence in age-matched samples (from first month of life) from control infants (*n* = 126) using a two-sided *z*-test. Infants in which *pfoA⁻ C. perfringens* was observed in a single sample were classified as having *pfoA⁻ C. perfringens*. Next, using only samples from infants classified as having *pfoA⁻ C. perfringens*, we compared average proportional abundance (metagenomic microbial reads mapped to assembly, per sample) and persistence (proportion of a single infant's samples in which *pfoA⁻ C. perfringens* was present, per infant) of *pfoA⁻ C. perfringens* between control (*n* = 29) and NEC (*n* = 3) infants.

### Antibiotic resistance testing

Antimicrobial susceptibility testing was performed using the agar dilution method following the Clinical and Laboratory Standards Institute guidelines. *Brucella* agar (Oxoid) supplemented with 5% laked sheep blood, hemin and vitamin K was used for the tests. The antimicrobial agents tested included vancomycin (range 0.125–4 mg l⁻¹), ampicillin (0.016–4 mg l⁻¹), metronidazole (0.064–8 mg ⁻¹), meropenem (0.004–0.5 mg l⁻¹) and penicillin (0.008–1 mg l⁻¹). All antibiotics were purchased from Discovery Fine Chemicals. All isolates were grown on preferred agar for 48 h at 37 °C in an anaerobic chamber before testing. For each isolate, 4–5 representative colonies were picked and resuspended to 0.5 McFarland bacterial suspension, and 1 μl was inoculated using a multipoint inoculator. Plates were incubated in an anaerobic chamber at 37 °C for 48 h. Minimum inhibitory concentrations (MICs) were identified as the lowest concentration of antimicrobial agent leading to visible inhibition of growth compared to control plate without antibiotics. Control strains with defined MIC concentrations were represented by *C. perfringens* NCTC 8237, *B. fragilis* NCTC 9343, *S. aureus* NCTC 12973 and *E. coli* NCTC 12241. Resistance to antibiotics was defined on the basis of the EUCAST clinical breakpoints 2024 for clostridia using the *C. perfringens* breakpoints, while for bifidobacteria the breakpoint values used were taken from the data for 'anaerobe, Gram-positive bacteria' reported in EUCAST clinical breakpoints 2021.

### RNA-sequencing of *C. perfringens* AM1 grown on specific HMOs

*C. perfringens* AM1 was grown in BHI to mid-exponential phase. Bacteria were centrifuged at 5,000 *g* for 5 min at 4 °C and then resuspended to

0.1 starting optical density (OD) in ZMB1 supplemented with either lactose or the HMOs they could grow on at a 0.5% concentration (w/v) (LNnT, DSLNT, 6'SL). RNA was extracted from 2 ml of culture when it reached mid-exponential phase using the RNeasy mini kit (Qiagen) following manufacturer instructions. Bacteria were pelleted by centrifuging at 8,000 g for 1 min, supernatant was saved for further analysis, and the cells were resuspended in 700 μl of RLT solution and 500 μl of 100% ethanol. The subsequent steps were performed following protocol instructions and the RNA was eluted in 30 μl in RNase-free water. Total RNA was quantified using the Qubit RNA High Sensitivity kit (Invitrogen) and the RNA integrity number (RIN) was determined using the RNA 6000 Nano kit on the BioAnalyzer 2100 (Agilent). Total RNA was diluted to 100 ng and ribosomal RNA depletion was carried out using the Ribo Zero Plus kit (Illumina) following manufacturer instructions. Library preparation was performed on the depleted RNA using the NEBNext Ultra II Directional RNA Library Prep kit for Illumina (New England Biolabs) and sequenced on the NovaSeq 6000 SP 100 cycle kit (Illumina). High-quality reads were aligned to the AM1 strain genome with Bowtie2 (v.2.4.5)[70] using the 'very-sensitive' option. Aligned reads were then processed using HTSeq (v.2.0.8)[71] to generate gene-level count data. 'DESeq2' (v.1.44.0)[72] in R (v.4.4.0)[66] was used for counts normalization and differential gene expression comparison between conditions. A gene was considered differentially expressed when absolute $\log_2$(fold change) > 2 and $P_{adj}$ < 0.05.

### Proteomics of *C. perfringens* AM1 grown on specific HMOs

Proteomics was performed on cell pellets and supernatants from *C. perfringens* AM1 grown to mid-exponential phase. Of the bacterial culture, 500 μl was centrifuged at 8,000 g for 1 min, the supernatant was saved, and the pellet was washed twice in 1 ml of cold PBS. The second PBS wash was removed, and the pellet and supernatant were stored at −80 °C until they were analysed. Secretome samples were first precipitated using the methanol/chloroform method. To 250 μl of growth media, 528 μl of methanol and 66 μl of chloroform were added, then vortexed and mixed with 698 μl of water. Samples were vortexed again and centrifuged for 15 min at 4,000 g at 4 °C. The top phase of the supernatant was removed. Cold methanol (1 ml) was added, followed by 30 min centrifugation at 16,000 g at 4 °C. The supernatant was discarded and the pellet air dried and dissolved in 30 μl of S-trap lysis buffer (5% SDS, 50 mM TEAB, pH 8.5). The cell pellets were sonicated in 100 μl of S-trap lysis buffer. Protein concentrations were measured using Micro BCA Protein Assay. An equivalent of 15 μg of total protein was used for digestion. Proteins were reduced with dithiothreitol at the final concentration of 20 mM (65 °C, 30 min). Cysteines were alkylated by incubation with iodoacetamide (40 mM final concentration, 30 min, room temperature in dark) and then acidified by adding 27.5% phosphoric acid to a final concentration of 2.5% (v/v). The samples were then loaded onto spin columns in six volumes of binding buffer (90% methanol, 100 mM TEAB, pH 8) and centrifuged at 4,000 g for 30 s. The columns were then washed with binding buffer (three times) and the flow through was discarded. Proteins were digested with trypsin (Worthington) in 50 mM TEAB pH 8.5, at a ratio of 10:1 protein to trypsin overnight at 37 °C. Peptides were eluted with three washes of: first 50 μl 50 mM TEAB, second 50 μl 0.1% formic acid and third 50 μl 50% acetonitrile with 0.1% formic acid. The solution was frozen, then dried in a centrifugal concentrator and reconstituted in 15 μl of 0.1% formic acid and 2% acetonitrile. Of each peptide sample, 1 μl was loaded per LC−MS run. Peptides were separated using an UltiMate 3000 RSLCnano high-performance liquid chromatography (HPLC). Samples were first loaded/desalted onto Acclaim PepMap100 C18 LC column (5 mm -0.3 mm i.d., 5 μm, 100 Å, Thermo Fisher) at a flow rate of 10 μl min⁻¹ maintained at 45 °C and then separated on a 75 μm × 75 cm C18 column (Thermo EasySpray -C18 2 μm) with integrated emitter using a 60 min nonlinear gradient from 92.5% A (0.1% formic acid (FA) in 3% dimethylsulfoxide) and 7.5% B (0.1% FA in 80% acetonitrile, 3%

dimethylsulfoxide), to 40% B, at a flow rate of 150 nl min⁻¹. The eluent was directed to a Thermo Q-Exactive HF mass spectrometer through the EasySpray source at a temperature of 300 °C and spray voltage of 1,500 V. The total LC−MS run time was 120 min. Orbitrap full scan resolution was 120,000, ACG Target $5 \times 10^6$, maximum injection time 100 ms, scan range 375−1,300 m/z. DIA MS/MS data were acquired with 15 m/z windows covering 328.5−1,251 m/z, at 30,000 resolution, maximum injection time of 100 ms, with ACG target set to $3 \times 10^6$ and normalized collision energy level of 27.

The acquired data were analysed in DIA-NN (v.1.8)[73] against the *C. perfringens* proteome sequence database (Uniprot UP000000818, version from 16 July 2024) combined with common Repository of Adventitious Proteins (cRAP), fragment m/z: 300−1,800, enzyme: trypsin, allowed missed-cleavages: 2, peptide length: 6−30, precursor m/z 300−1,250, precursor charge: 2−4, fixed modifications: carbamidomethylation(C), variable modifications: Oxidation(M), Acetylation(*N*-term). The normalized data were then analysed using 'Limma' (v.3.60.4)[74] in R (v.4.4.0)[66]. Supernatant and pellet samples were analysed separately. Proteins were deemed significant when associated with an absolute $\log_2$(fold change) > 1 and $P_{adj}$ < 0.05.

### HMO and lactose quantification in bacterial culture supernatants

HMOs and lactose were measured following a previously published method[75]. This method uses labelling by reductive amination, with 4-aminobenzoic acid ethyl ester (benzocaine) as the labelling reagent and picoline borane as the reducing agent, then applies HPLC separation with UV detection.

### Fucose, LNB and sialic acid quantification in bacterial culture supernatants

Fucose, LNB and free sialic acid were measured using HPLC−MS. Samples were diluted tenfold in 1:1 acetonitrile:water, and 50 μl internal standard solution (4 g l⁻¹ sucrose) was added. A 7-point calibration curve of all three compounds was used. Ultra-high-performance liquid chromatography (UHPLC; Thermo Fisher Ultimate 3000) was used with a binary pump coupled to a Bruker microTOF Q-TOF mass spectrometer. A Thermo Fisher Accucore 150-Amide-HILIC analytical column was used (dimensions 150 × 3.0 mm, 2.6 μm particle size). A binary gradient of acetonitrile (eluent A) and 50 mM ammonium formate adjusted to pH 3 (eluent B) was used at a flow rate of 0.8 ml min⁻¹. The initial composition was 75% A and 25% B, changed to 70% A and 30% B over 5 min, then to 60% A and 40% B in 0.2 min. This was held for 2.8 min to elute any potentially present larger oligosaccharides, then changed to 75% A and 25% B in 0.1 min and equilibrated for 3 min before the next injection. The MS was used in negative mode, and extracted ion chromatograms at m/z 341 (internal standard, [M-H]⁻), m/z 209 (fucose, [M + HCOOH-H]⁻), m/z 428 (LNB, [M + HCOOH-H]⁻) and m/z 308 (sialic acid, [M-H]⁻) were used for integration, selected on the basis of the mass spectra recorded in the calibration solutions. A quadratic calibration curve with internal standard was used.

### SCFA profiling of *Clostridium* and *Bifidobacterium* culture supernatants and data analysis

For isolates found to grow on HMOs following the above growth curve protocol, supernatants from wells where bacterial growth was observed were collected, centrifuged at 5,000 g for 5 min at 4 °C to remove any bacteria, and the supernatants collected. Quantitative measurement of SCFAs (acetic acid, propionic acid, butyric acid, isobutyric acid, valeric acid, isovaleric acid and hexanoic acid) in the collected samples was then performed by Creative Proteomics, using a gas chromatography−mass spectrometry (GC−MS) method. Samples were diluted in water containing labelled internal standards for each chain length (C2−C6). The free short-chain fatty acids were derivatized using methyl chloroformate in 1-propanol, yielding propyl esters before

subsequent liquid–liquid extraction into hexane and analysis on an SLB-5ms (30 × 0.25 mm × 1.0 μm) column and detection using GC–EI–MS in SIM mode. The analytes were quantified using 8-point calibration curves. For analysis of the data, the raw SCFA concentrations for each strain were divided by the maximum $OD_{600}$ measured during culture of the strain as part of the growth curve protocol described above, giving growth-adjusted concentrations. These values were then averaged for each species. Boxplots of these data were then created in R (v.4.4.0)[66] using the ggplot2 package. Between-species comparisons for mean SCFA growth-adjusted concentrations were performed using analysis of variance (ANOVA) followed by Tukey's honestly significant different (HSD) test in R[66], with $P < 0.05$ set as the threshold for statistical significance.

### Preparation of bacterial CFSs

A loopful of a glycerol stock was streaked out across an agar plate of the isolate's preferred medium and incubated anaerobically at 37 °C overnight. A single colony was then picked from this plate, transferred to 5 ml broth based on the isolate's preferred medium and incubated overnight. Two 5 ml aliquots of ZMB1 were prepared. Glucose was added to 10 mg ml$^{-1}$ to one aliquot, while the HMOs known to sustain growth of the isolate (Fig. 1a) were added to the second aliquot in equal masses, to a total concentration of 10 mg ml$^{-1}$. Due to insufficient material for DSLNT, this HMO could not be included in the medium. The $OD_{600}$ of the overnight growth of the isolate was measured and the volume required to dilute to OD 0.05 in 5 ml of ZMB1 was calculated. Two aliquots of the calculated volume of the culture were then centrifuged at 5,000 $g$ for 5 min at 4 °C, and the pellet was resuspended in 500 μl of anaerobic PBS. Centrifugation was repeated and the pellets then resuspended in 250 μl of ZMB1 + glucose and ZMB1 + HMOs, respectively. The culture aliquots were then added to the remaining 4.75 ml of ZMB1 + glucose and ZMB1 + HMOs. The two cultures were then incubated anaerobically at 37 °C, with shaking at 130 r.p.m., for 2 days. At the end of the incubation period, the two cultures were centrifuged at 5,000 $g$ for 10 min at 4 °C, and the supernatants transferred to new tubes in the anaerobic chamber. These supernatants were then filter sterilized with Merck Millex-GP Sterile 0.22 μm syringe filters, producing CFSs. CFSs were prepared in triplicate for each strain.

### Untargeted metabolomics of CFSs

Metabolomics was performed on each CFS replicate per strain by Metabolon, as described below.

**Sample preparation for untargeted metabolomics.** Samples were prepared using the automated MicroLab STAR system from Hamilton Company. Several recovery standards were added before the first step in the extraction process for quality control (QC) purposes. To remove protein, dissociate small molecules bound to protein or trapped in the precipitated protein matrix, and recover chemically diverse metabolites, proteins were precipitated with methanol under vigorous shaking for 2 min (Glen Mills GenoGrinder 2000) followed by centrifugation. The resulting extract was divided into multiple fractions: two for analysis by two separate reverse phase (RP)/UPLC–MS/MS methods with positive ion mode electrospray ionization (ESI), one for analysis by RP/UPLC–MS/MS with negative ion mode ESI, one for analysis by HILIC/UPLC–MS/MS with negative ion mode ESI, while the remaining fractions were reserved for backup. Samples were placed briefly on a TurboVap (Zymark) to remove the organic solvent. The sample extracts were stored overnight under nitrogen before preparation for analysis.

**Quality assurance/QC.** Several types of control were analysed in concert with the experimental samples: a pooled matrix sample generated by taking a small volume of each experimental sample (or alternatively, use of a pool of well-characterized human plasma) served as a technical replicate throughout the data set; extracted water samples served

as process blanks; and a cocktail of QC standards that were carefully chosen not to interfere with the measurement of endogenous compounds was spiked into every analysed sample, allowed instrument performance monitoring and aided chromatographic alignment. Instrument variability was determined by calculating the median relative standard deviation (RSD) for the standards that were added to each sample before injection into the mass spectrometers. Overall process variability was determined by calculating the median RSD for all endogenous metabolites (that is, non-instrument standards) present in 100% of the pooled matrix samples. Experimental samples were randomized across the platform run with QC samples spaced evenly among the injections.

**Data generation.** All methods utilized a Waters ACQUITY ultra-performance liquid chromatography (UPLC) and a Thermo Scientific Q-Exactive high resolution/accurate mass spectrometer interfaced with a heated electrospray ionization (HESI-II) source and Orbitrap mass analyser operated at 35,000 mass resolution (PMID: 32445384). The dried sample extracts were then reconstituted in solvents compatible to each of the four methods. Each reconstitution solvent contained a series of standards at fixed concentrations to ensure injection and chromatographic consistency. One aliquot was analysed using acidic positive ion conditions, chromatographically optimized for more hydrophilic compounds (PosEarly). In this method, the extract was gradient eluted from a C18 column (Waters UPLC BEH C18 2.1×100 mm, 1.7 μm), using water and methanol, containing 0.05% perfluoropentanoic acid (PFPA) and 0.1% FA. Another aliquot was also analysed using acidic positive ion conditions; however, it was chromatographically optimized for more hydrophobic compounds (PosLate). In this method, the extract was gradient eluted from the same aforementioned C18 column using methanol, acetonitrile, water, 0.05% PFPA and 0.01% FA, and was operated at an overall higher organic content. Another aliquot was analysed using basic negative ion optimized conditions using a separate dedicated C18 column (Neg). The basic extracts were gradient eluted from the column using methanol and water, but with 6.5 mM ammonium bicarbonate at pH 8. The fourth aliquot was analysed via negative ionization following elution from a HILIC column (Waters UPLC BEH Amide 2.1 × 150 mm, 1.7 μm) using a gradient consisting of water and acetonitrile with 10 mM ammonium formate, pH 10.8 (HILIC). The MS analysis alternated between MS and data-dependent MSn scans using dynamic exclusion. The scan range varied slightly between methods but covered 70–1,000 $m/z$. Raw data files were archived and extracted as described below.

**Bioinformatics.** Raw data were extracted, peak identified and QC processed using a combination of Metabolon-developed software services (applications). Each of these services perform a specific task independently, and they communicate/coordinate with each other using industry-standard protocols. Compounds were identified by comparison to library entries of purified standards or recurrent unknown entities. Metabolon maintains a library, based on authenticated standards, that contains the retention time/index (RI), mass to charge ratio ($m/z$) and fragmentation data on all molecules present in the library. Furthermore, biochemical identifications are based on three criteria: retention index within a narrow RI window of the proposed identification, accurate mass match to the library ±10 ppm, and the MS/MS forward and reverse scores between the experimental data and authentic standards. The MS/MS scores are based on a comparison of the ions present in the experimental spectrum to the ions present in the library spectrum. While there may be similarities between molecules based on one of these factors, the use of all three data points is utilized to distinguish and differentiate biochemicals. More than 5,400 commercially available purified or in-house synthesized standard compounds have been acquired and analysed on all platforms for determination of their analytical characteristics. An additional 7,000

mass spectral entries have been created for structurally unnamed biochemicals, which have been identified by virtue of their recurrent nature (both chromatographic and mass spectral). These compounds have the potential to be identified by future acquisition of a matching purified standard or by classical structural analysis. Metabolon continuously adds biologically relevant compounds to its chemical library to further enhance its level of Tier 1 metabolite identifications.

**Compound quality control.** A variety of curation procedures were carried out to ensure that a high-quality data set was made available for statistical analysis and data interpretation. The QC and curation processes were designed to ensure accurate and consistent identification of true chemical entities, and to remove or correct those representing system artefacts, mis-assignments, mis-integration and background noise. Metabolon data analysts use proprietary visualization and interpretation software to confirm the consistency of peak identification and integration among the various samples.

**Metabolite quantification and data normalization.** Peaks were quantified using AUC. For studies spanning multiple days, a data normalization step was performed to correct variation resulting from instrument inter-day tuning differences. Essentially, each compound was corrected in run–day blocks by registering the medians to equal one (1.00) and normalizing each data point proportionately (termed the 'block correction'). For studies that did not require more than 1 day of analysis, no normalization was necessary, other than for purposes of data visualization. In certain instances, biochemical data might have been normalized to an additional factor (for example, cell counts, total protein as determined by Bradford assay, osmolality and so on) to account for differences in metabolite levels due to differences in the amount of material present in each sample.

## Metabolomics data analysis

Following receipt of data from Metabolon, imputation was performed for missing values in the median normalized data. Imputed values were calculated per metabolite by identifying the lowest value measured for each metabolite and dividing it by 4. All data were then transformed using natural logarithm. All subsequent analyses were then performed on these transformed data. Per strain, the metabolomes of glucose and HMO-derived CFSs were compared using permutational multivariate analysis of variance (PERMANOVA) with the vegan (v.2.6.8)[76] R package, with method set to Euclidean distance. Differential abundance analysis comparing metabolite levels in CFSs to those in ZMB1 medium were performed using Limma (v.3.56.2)[74], with thresholds set at $\log_2$(fold change) ± 1 and $P_{adj} < 0.05$. $P$ values were adjusted using the Benjamini–Hochberg method. Statistical comparisons of metabolite levels between species were performed using ANOVA, followed by Tukey's HSD test. $P < 0.05$ was set as the threshold for statistical significance. A Venn diagram of metabolites shared across *Clostridium* species was generated using the VennDiagram package (v.1.7.3) (https://cran.r-project.org/web/packages/VennDiagram/index.html).

## CFS activity assay against pathobionts and bifidobacteria

A glycerol stock of the isolate being tested was streaked out on an agar plate of the isolate's preferred medium and incubated anaerobically at 37 °C overnight. Single colonies were then picked from the plate, transferred to 5 ml broth based on the isolate's preferred medium and incubated anaerobically at 37 °C overnight. On the same day, a fresh batch of ZMB1 medium was prepared and split into three aliquots. One aliquot was supplemented with glucose to a concentration of 10 mg ml⁻¹, the second with glucose to a concentration of 20 mg ml⁻¹ and the third left as the base medium. All media aliquots were then left in the anaerobic chamber overnight to remove any oxygen. The next day, the $OD_{600}$ of the bacterial culture was measured and the volume required to dilute it to an $OD_{600}$ of 0.2 in 3 ml ZMB1 + glucose

(10 mg ml⁻¹) was calculated. The required volume of the culture was then centrifuged at 5,000 *g* for 5 min at 4 °C and the supernatant removed. The pellet was then resuspended in 1 ml anaerobic PBS and centrifuged again as before, with the supernatants removed. The pellet was then resuspended in 3 ml ZMB1 + glucose (10 mg ml⁻¹). The culture was then incubated anaerobically at 37 °C, with shaking at 110 r.p.m., until it reached mid-exponential growth phase. At this point, the $OD_{600}$ was measured and the volume required to dilute to an $OD_{600}$ of 0.1 in 10 ml ZMB1 + glucose (20 mg ml⁻¹) was calculated. The required volume of the cultures was then centrifuged at 5,000 *g* for 5 min at 4 °C and the supernatant removed. The pellet was resuspended in 10 ml ZMB1 + glucose (20 mg ml⁻¹). Of this isolate culture, 100 µl was then added to the wells of a 96-well plate. Glucose-based CFSs (see 'Preparation of bacterial CFSs') were used for these assays and were thawed under anaerobic conditions. Of each CFS being tested, 100 µl was added to individual wells. If required for the experiment, CFS pH was adjusted from acidic to neutral using concentrated NaOH, with pH measured using a Thermo Scientific Orion ROSS Ultra pH electrode. In addition, an SCFA mixture containing acetate and butyrate diluted in ZMB1 was prepared, with concentrations of each based on the average concentrations detected in all *Clostridium* samples during SCFA profiling by Creative Proteomics (acetate 122.3 mM, butyrate 47.6 mM). This SCFA mix was split into 3 aliquots and each adjusted to pH 4, 6 and 7 using either concentrated HCl or concentrated NaOH as required. Of the SCFA mix at each pH, 100 µl was added to individual wells on the assay plate. Three aliquots of ZMB1 base medium were also adjusted to pH 4, 6 and 7, and 100 µl of each added to individual wells. All conditions were set up in triplicate on the plate. Following this set up, the minimum concentration of glucose in each well was 10 mg ml⁻¹. The plate was then shaken at 100 r.p.m. for 1 min. The lid was then removed and the plate sealed with a Diversified Biotech Breathe-Easy plate seal. The sealed plate was then placed in a Cerillo Stratus plate reader and the assay run for up to 3 days, with readings taken at 600 nm every 3 min. For each test, AUC for time vs $OD_{600}$ was calculated using a trapezoidal model with the trapz() function in the caTools package (v.1.18.2)[77]. Statistical comparisons were performed using ANOVA followed by Dunnett's test, comparing each test to the AUC for the growth of each strain with pH 7 ZMB1 medium added. The threshold for significance was set at $P < 0.05$. For plotting using the pheatmap (v.1.0.12)[78] package, the AUC for the growth of each strain with pH 7 ZMB1 medium added was set as a 100% growth reference for all other conditions and the percentages of this control AUC calculated accordingly. Selected growth curves were generated by plotting mean $OD_{600}$ calculated from 3 replicates.

## MTS assay to measure viability of Caco-2 cells incubated with CFS

Caco-2 cells were cultured in Advanced DMEM medium supplemented with FBS to 10% and GlutaMAX. For the assay, the cells were seeded at a density of 5,000 cells per well in a volume of 200 µl per well across 96-well plates and incubated at 37 °C and 5% $CO_2$ for 24 h. The old medium was then removed and the following conditions set up in 100 µl: (1) DMEM-only control, (2) base ZMB1 medium at 10%, 25%, 50% and 75% dilutions in DMEM, (3) 11× HMO CFSs in separate wells at 10%, 25%, 50% and 75% dilutions in DMEM and (4) 12× glucose CFSs in separate wells at 10%, 25%, 50% and 75% dilutions in DMEM. All conditions were set up in triplicate. The assays were then incubated at 37 °C and 5% $CO_2$ for 24 h. Medium was then removed from each well and replaced with 100 µl fresh DMEM. MTS reagent was reduced by metabolically active cells into a formazan salt that is soluble in tissue culture medium, giving a detectable colour change which can be quantified by measuring absorbance at 490 nm using a plate reader. The measured absorbance at 490 nm corresponds to the quantity of formazan produced, which in turn corresponds to the number of living cells in the culture. Thus, the measured absorbance is directly proportional to the number of viable cells and allows the viability of a culture to be compared with

that of others. Fresh MTS reagent was prepared by adding phenazine methosulfate to a concentration of 5% to MTS tetrazolium salt solution. Of the reagent, 20 µl was then added to each well of the assay plate, which was then incubated at 37 °C and 5% $CO_2$ for 4 h. MTS reagent was also added to triplicate wells containing only DMEM to act as a blank. The absorbance at 490 nm of each well was then measured. Following blank correction, absorbances were converted to % viabilities, with absorbances for DMEM-only controls being used as the denominators in % calculations. 'DMEM only' controls were therefore set as 100% viability, and % viabilities for CFS and ZMB1 conditions were calculated relative to those controls. Calculated viabilities for each condition were then averaged. Statistical comparisons were performed at the genus level, with ANOVA and Tukey HSD test used to compare average % viabilities for *Clostridium* and *Bifidobacterium* CFSs and blank ZMB1 medium across each concentration tested. $P < 0.05$ was set as the threshold for statistical significance.

## Organoid media production

Organoid media were made as previously described[79].

Complete media growth factor negative (CMGF⁻): 500 ml Advanced DMEM/F12, 5 ml 100× GlutaMAX, 5 ml 1 M HEPES.

Complete media growth factor positive (CMGF⁺) in a volume of 500 ml: 78 ml CMGF⁻, 250 ml Wnt3A-conditioned media produced from ATCC CRL-2647 cells (ATCC), 100 ml R-spondin-conditioned media produced from R-spondin1-expressing 293T cells (Merck), 50 ml Noggin-conditioned media produced from 293-Noggin cells[80], 10 ml B27 (50×), 5 ml N2 (100×), 5 ml nicotinamide (10 mM), 1 ml *N*-acetylcysteine (1 mM), 500 µl gastrin (10 nM), 500 µl A83 (500 nM), 166 µl SB202190 (10 µM), 50 µl EGF (50 ng ml⁻¹).

High Wnt medium in a volume of 500 ml: 250 ml CMGF+, 250 ml Wnt3A-conditioned media.

Differentiation (DIF) medium in a volume of 500 ml: 458 ml CMGF⁻, 25 ml Noggin-conditioned media, 10 ml B27 (50×), 5 ml N2 (100×), 1 ml *N*-acetylcysteine (500 mM), 500 µl gastrin (100 µM), 500 µl A83 (500 µM), 50 µl EGF (500 µg ml⁻¹).

## Establishment and culture of PIOs

Intestinal epithelial organoid lines were established from intestinal crypts isolated from preterm neonate tissue as described in ref. 79. Tissue was minced, washed with chelating solution, antifungals and antibiotics, and crypt cells extracted by gentle shaking in EDTA. Extracted cells were then suspended within phenol-red-free and growth factor-reduced Matrigel basement membrane matrix (Corning), which was then pipetted as small dots into the wells of a 24-well plate at a maximum of 3 dots per well. The Matrigel was set by incubating at 37 °C for 30 min. Following polymerization of the Matrigel, 500 µl High Wnt growth medium was added to each well and the organoids left to grow from the crypt cells, with incubation at 37 °C and 5% $CO_2$. Media were changed for fresh every Monday, Wednesday and Friday. Organoids were passaged by removing spent medium and adding 300 µl 0.05% trypsin-EDTA (Gibco) per well. The trypsin was then pipetted up and down to break up the Matrigel and suspend the organoids. Organoids were incubated in trypsin at 37 °C and 5% $CO_2$ for 5 min, with 350 µl FBS (Merck) then added to stop trypsinization. Following centrifugation at 363 *g* for 5 min, supernatant was removed and the organoids resuspended in ice-cold Matrigel, the volume of which depended on the number of wells being seeded. The Matrigel was then pipetted as small dots into the wells of a 24-well plate at a maximum of 3 dots per well and set by incubating at 37 °C for 30 min. Following polymerization of the Matrigel, 500 µl High Wnt growth medium was added to each well and the organoids left to grow, with incubation at 37 °C and 5% $CO_2$. All organoid experiments were conducted with cell line NCL 27, within passages 10–15. This line was established from a male patient born at 24 weeks gestation, using ileum tissue salvaged from surgery performed on DOL 10 due to the development of necrotizing enterocolitis.

## Organoid monolayer co-culture with bacterial CFSs and inflammatory stimuli

Three-dimensional (3D) organoids were processed into 2D monolayers in 6.5 mm Transwells (Corning) following the protocol described in ref. 40. 3D organoid cultures were processed into single-cell suspensions by washing with EDTA, trypsinization (0.05% trypsin-EDTA) and passage through a 40 µm nylon cell strainer (Corning). Cells were then resuspended in CMGF⁺ growth medium ($1.8 × 10^6$ cells per ml) and seeded onto Transwells pre-coated with diluted (1:40 in cold PBS) Matrigel. Growth medium was replaced with DIF medium after 2–3 days, once TEERs were near 300 Ω, indicating monolayer confluence. Monolayers were differentiated for 4 days before experiments. Monolayers were then placed into an anaerobic co-culture system[40], which was in turn placed into an anaerobic chamber. Oxygen was flowed through the base of the co-culture unit, maintaining aerobic conditions on the basolateral sides of the monolayers while the apical sides remained under the anaerobic conditions of the chamber. Thus, the physiological oxygen gradient of the gut epithelium was recreated in this model. The inflammatory stimuli used were LPS from *Escherichia coli* 0111:B4 (InvivoGen) and flagellin from *Salmonella typhimurium* (InvivoGen), following a previously optimized method for organoid inflammation induction[81]. CFSs were prepared as described above. The following conditions were set up in triplicate across the monolayers: (1) no stimuli, no CFS ('control'), (2) +apical CFS to 25% v/v ('CFS 25%'), (3) +apical LPS (100 ng ml⁻¹) + basolateral flagellin (100 ng ml⁻¹) ('stimuli'), (4) +apical CFS to 25% v/v + apical LPS (100 ng ml⁻¹) + basolateral flagellin (100 ng ml⁻¹) ('CFS 25% and stimuli'). The experiment was run for 3 h. The TEERs of the monolayers were measured before the start and at the end of the experiment. Endpoint TEERs were subtracted from startpoint TEERs to obtain the change in TEER during the experiment. Outliers were identified by calculating robust/modified $z$-scores for each data point. This method uses the median to calculate median absolute deviation. Any data point with a robust $z$-score >+3.5 or <−3.5 was removed from the dataset. The mean change in TEER for each condition was then calculated. Statistical comparisons of changes in TEER between conditions were performed using ANOVA and Tukey HSD test. Conditions indicated with matching letters are not significantly different. Furthermore, after the incubation, apical and basolateral supernatants were collected for cytokine assays.

## Cytokine assays

Secreted interleukin-8 (IL-8) was measured using a DuoSet ELISA kit following manufacturer protocol. All other cytokines were measured using custom U-Plex assays (Meso Scale Discovery) following manufacturer protocol. Outliers were identified by calculating robust/modified $z$-scores for each data point. This method uses the median to calculate median absolute deviation. Any data point with a robust $z$-score >+3.5 or <−3.5 was removed from the dataset. ANOVA followed by a Dunnett's test to compare each condition to the untreated control were used to perform statistical analysis of cytokine secretion data. For comparisons of all against all, the ANOVA was followed by Tukey's HSD test. $P < 0.05$ was set as the threshold for statistical significance.

## Organoid monolayer co-culture with live *C. perfringens* and inflammatory stimuli

Three-dimensional organoids were processed into 2D monolayers in 6.5 mm Transwells following the protocol described in ref. 40. Monolayers were then placed into the anaerobic co-culture system, which was in turn placed into an anaerobic chamber. Oxygen was flowed through the base of the co-culture unit, maintaining aerobic conditions on the basolateral sides of the monolayers while the apical sides remained under the anaerobic conditions of the chamber. Thus, the physiological oxygen gradient of the gut epithelium was recreated in this model. The inflammatory stimuli used were LPS from *E. coli* 0111:B4 (InvivoGen) and flagellin from *S. typhimurium* (InvivoGen), following a

previously optimized method for organoid inflammation induction[81]. ZMB1 medium was prepared a day ahead, mixed 1:1 with organoid DIF medium and this mixture left in the anaerobic chamber overnight. *C. perfringens* AM1 and JC26 were grown overnight in BHI broth. The next day, the $OD_{600}$ of each culture was measured and volumes required to dilute each to an $OD_{600}$ of 0.2 in 2 ml ZMB1:DIF were calculated. The required volumes of the cultures were then centrifuged at 5,000 $g$ for 5 min at 4 °C and the supernatants removed. The pellets were then resuspended in 1 ml anaerobic PBS and centrifuged again as before, with the supernatants removed. All pellets were then resuspended in 2 ml ZMB1:DIF. The cultures were then incubated anaerobically at 37 °C, with shaking at 110 r.p.m. until they each reached mid-exponential growth phase. The $OD_{600}$ of each was then measured and volumes required to dilute them to an $OD_{600}$ of 0.2 in 0.2 ml ZMB1 were calculated. The required volumes of the cultures were then centrifuged at 5,000 $g$ for 5 min at 4 °C and the supernatants removed. The pellets were then resuspended in 0.2 ml ZMB1:DIF. These cultures were then added to the apical side of the monolayers as required. A second culture of JC26 was prepared in the same way but timed to be added to the appropriate monolayers at the end of the first hour of the experiment. The following conditions were set up in triplicate across the monolayers: (1) no stimuli, no bacteria, (2) +AM1, (3) +JC26, (4) +AM1 for 1 h, then add JC26, (5) no treatment for 1 h then add apical LPS (100 ng ml$^{-1}$) and basolateral flagellin (100 ng ml$^{-1}$) and (6) +hAM1 for 1 h, then add apical LPS (100 ng ml$^{-1}$) and basolateral flagellin (100 ng ml$^{-1}$). The ZMB1:DIF mixture was used as the apical medium across all conditions.

The experiment was run for 3 h. The TEERs of the monolayers were measured with an epithelial Ohm meter before the start and at the end of the experiment. Endpoint TEERs were subtracted from startpoint TEERs to obtain the changes in TEER during the experiment. The mean change in TEER for each condition was then calculated. Outliers were identified by calculating robust/modified $z$-scores for each data point. This method uses the median to calculate the median absolute deviation. Any data point with a robust $z$-score >+3.5 or <−3.5 was removed from the dataset. Statistical comparisons of changes in TEER between conditions were performed using ANOVA and Tukey HSD test. $P < 0.05$ was set as the threshold for statistical significance. Conditions indicated with matching letters are not significantly different. Furthermore, after the incubation, apical and basolateral supernatants were collected for cytokine assays.

## Proteomics on *C. perfringens* AM1 and JC26 CFSs

CFSs were generated as described above. Proteins were purified as described in the section 'Proteomics of *C. perfringens* AM1 grown on specific HMOs'. An equivalent of 5 μg of total protein was then used for digestion. Digested protein was reduced, alkylated, digested and washed as described above, except that they were finally reconstituted in 20 μl of 0.1% formic acid and 2% acetonitrile. Of each sample, ~0.5 μg was loaded onto Evotips according to manufacturer instructions and peptides separated on an 8 cm × 100 μm Evosep Endurance C18 column (Evosep, EV1094) using an Evosep One system (Evosep) with a predefined samples per day protocol 60. The 21 min gradient ran from 0–35% solvent B (solvent A: 0.1% formic acid in water, solvent B: 0.1% formic acid in acetonitrile) at 100 nl min$^{-1}$. Through a 20 μm captive spray emitter (Bruker) at 50 °C, the analytes were directed to a timsToF HT mass spectrometer (Bruker Daltonics). The instrument operated in DIA-PASEF mode, acquiring mass and ion mobility ranges of 300–1,400 $m/z$ and 0.6–1.4 1/K0, with a total of 16 variable IM-$m/z$ windows with two quadrupole positions per window, designed using py_diAID from a pooled sample subjected to DDA-PASEF[82]. TIMS ramp and accumulation times were 100 ms, total cycle time was ~1.8 s. Collision energy was applied in a linear fashion, where ion mobility = 0.6–1.6 1/K0 and collision energy = 20–59 eV. The acquired data were analysed as described in 'Proteomics of *C. perfringens* AM1 grown on specific HMOs'.

## Seahorse mitochondrial stress test

The impacts of CFSs on organoid mitochondrial bioenergetic function were tested using the Seahorse XF Cell Mito Stress Test (Agilent). This is a live cell assay that detects changes in parameters of mitochondrial function through directly measuring the oxygen consumption rate (OCR) of cells in response to the addition of modulators of respiration. The modulators used were oligomycin (2 μM), carbonyl cyanide-4 (trifluoromethoxy) phenylhydrazone (FCCP) (4 μM), rotenone (0.5 μM) and antimycin A (0.5 μM). Stocks of the modulators were diluted in organoid CMGF$^-$ medium for use in this experiment. The Seahorse XFe96 sensor cartridge (Agilent) containing these modulators was prepared following manufacturer instructions. Organoid monolayers were generated from NCL 27 at passage 12, as described above, except that they were seeded into a Seahorse XFe96/XF Pro Cell Culture microplate (Agilent) at a concentration of 0.3 × 10$^6$ cells per ml, with 200 μl per well. Before seeding, the Seahorse microplate was coated with 32 μl per well of Matrigel diluted 40:1 in PBS. The PBS was removed before monolayer seeding. Monolayers were differentiated for 4 days before the start of experiment. On the day of the experiment, the DIF medium was removed from the plate and the cells washed twice in 200 μl pre-warmed CMGF$^-$ medium. After washing, 135 μl CMGF$^-$ medium was added to each well. Each CFS being tested or ZMB1 (base medium) was then added to individual wells to final concentrations of 25% v/v and giving a total volume of 180 μl, following manufacturer protocol. A combination of LPS (100 ng ml$^{-1}$) and flagellin (100 ng ml$^{-1}$) was also added to a set of wells to create an 'inflammatory stimuli' condition. A set of wells containing only CMGF$^-$ were also set up as negative controls. All conditions were set up in triplicate. The plate was then incubated for 1 h at 37 °C in a non-CO$_2$ incubator, following manufacturer protocol. The plate was loaded into a Seahorse XF96 Analyser (Agilent) and the XF Mito Stress Test protocol run. Following instrument calibration, the assay was run for 73 min. Raw data were then processed and exported from the associated software Wave (v.2.6.3, Agilent). Outliers were identified by calculating robust/modified $z$-scores for each data point. This method uses the median to calculate the median absolute deviation. Any data point with a robust $z$-score >+3.5 or <−3.5 was removed from the dataset. Statistical analyses of raw OCR values for each parameter measured were performed using ANOVA followed by Dunnett's test to compare each condition to the negative control. $P < 0.05$ was set as the threshold for statistical significance.

## *C. perfringens* AM1 and JC26 growth curves in ZMB1:DIF medium

Glycerol stocks of AM1 and JC26 were streaked out on BHI agar plates and incubated in an anaerobic chamber at 37 °C overnight. Single colonies were then picked from these plates, transferred to 3 ml BHI broth and incubated in the anaerobic chamber at 37 °C overnight. A 50:50 mixture of ZMB1 and DIF media was also prepared and incubated in the anaerobic chamber overnight. The next day, the $OD_{600}$ of each culture was measured and volumes required to dilute each to an $OD_{600}$ of 0.2 in 5 ml ZMB1:DIF were calculated. The required volumes of the cultures were then centrifuged at 5,000 $g$ for 5 min at 4 °C and the supernatants removed. The pellets were resuspended in 1 ml anaerobic PBS and centrifuged again as before, with the supernatants removed. All pellets were then resuspended in 5 ml ZMB1:DIF. Three 5 ml cultures each of AM1 and JC26 were set up in this manner, constituting 3 biological replicates per strain. The cultures were then incubated anaerobically at 37 °C, with shaking at 110 r.p.m. The $OD_{600}$ of all cultures were measured, first at 60 min, and then every 30 min up to 180 min. Statistical comparisons between the two strains at each timepoint were made using unpaired $t$-test, with a threshold of $P < 0.05$ set for significance.

## Statistical analysis

All statistical analyses described above were performed in R (v.4.4.0)[66]. For outlier detection where described, medians and median absolute

deviations were calculated using the 'median()' and 'mad()' functions in R. ANOVA was performed using the 'aov()' function in R. Tukey's HSD tests were performed using the 'TukeyHSD()' function in R and letters representing statistical significance generated using the 'HSD.test()' function in the agricolae (v.1.3-7) package (https://cran.r-project.org/web/packages/agricolae/index.html). Dunnett's tests were performed using the 'glht()' function in the multcomp (v.1.4-29) package (https://cran.r-project.org/web/packages/multcomp/index.html). *t*-tests were performed using the 't.test()' function in R.

### Reporting summary

Further information on research design is available in the Nature Portfolio Reporting Summary linked to this article.

## Data availability

The RNA-seq data have been deposited in the Sequencing Read Archive (SRA) under study accession number PRJNA1214204. The proteomics datasets are deposited in MassIVE under submission ID MSV000096907. Sequencing reads for de novo genomes have been deposited in the European Nucleotide Archive (ENA) under accession number ERP187615. Source data are provided with this paper.

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

## Acknowledgements

This work was funded by a Sir Henry Dale Fellowship jointly funded by the Wellcome Trust and the Royal Society (221745/Z/20/Z, to C.J.S.), a Newcastle University Academic Career Track (NUACT) Fellowship (to C.J.S.), the 2021 Lister Institute Prize Fellow Award (to C.J.S.) and a Newcastle University UKRI Impact Accelerator Award (IAA-CiC; MR/X50290X/1 to C.J.S.). G.R.Y. and C.A.L. acknowledge support from the NIHR Newcastle Biomedical Research Centre. H.P.B., Y.S. and T.D.L. acknowledge funding support from the Wellcome Trust (220540/Z/20/A). L.J.H. acknowledges support from Wellcome Investigator Award (220540/Z/20/A). Human milk oligosaccharides were kindly donated by dsm-firmenich through the HMO Donation Program. We thank D. Molnár-Gábor (dsm-firmenich, Denmark) for running high-performance liquid chromatography of the cell-free supernatants; S. Kroll, K. Sim and A. G. Shaw (Imperial College London) for providing some of the *Clostridium perfringens* isolates presented in Supplementary Fig. 1; and J. Palmer for support with the Meso Scale Discovery assays.

## Author contributions

A.C.M., J.A.C. and C.J.S. conceived and designed the experiments. A.C.M., J.A.C. and J.A.D. tested bacterial isolate growth on HMOs. G.R.Y. analysed *Clostridium* prevalence in the MAGPIE dataset. R.K. and L.J.H. performed phylogenetic and genomic analyses on *Clostridium perfringens* isolates and provided additional bacterial isolates for testing on HMOs. A.C.M. performed the experiments, sample preparation and data analyses to characterize *Clostridium* HMO utilization genes and proteins. J.P.R.C. and A.N. supported the RNA-seq, and P.P. and A.P. performed the proteomics. M.L. performed the high-performance liquid chromatography of the cell-free supernatants. A.C.M. and J.A.C. generated samples for SCFA and metabolomics analysis and J.A.C. analysed the data. J.A.C. performed pathobiont and *Bifidobacterium* growth challenge assays using CFSs. P.P. and A.P. performed proteomics on CFSs and J.A.C. analysed the data. J.A.C. performed the MTS assay. A.C.M. and J.A.C. cultured PIOs. J.A.C. performed organoid co-cultures with CFSs and live bacteria including ELISAs and MSD cytokine assays. J.A.C. performed AM1 and JC26 growth curves. J.A.C., H.W. and C.A.L. designed and performed organoid Seahorse assays and J.A.C. analysed the data. J.D.P. performed MALDI–TOF to identify bacterial isolates at the genus or species level. H.P.B., Y.S. and T.D.L. performed whole-genome sequencing of bacterial isolates. L.C.B. performed genome assembly and annotation. J.E.B. and N.D.E. oversaw patient sample collection. C.J.S. conceived and supervised the study. J.A.C., A.C.M. and C.J.S. prepared figures and wrote the paper. All authors approved the final version of the manuscript.

## Competing interests

J.A.C., A.C.M., N.D.E., J.E.B. and C.J.S. are co-inventors on a patent relating to this work (GB2304447.2A). N.D.E. and J.E.B. declare research funding paid to their institution from Prolacta Biosciences, NeoKare UK and Danone Early Life Nutrition for grants between 2016–2022. N.D.E. declares lecture honoraria from Nestle Nutrition Institute and Abbot Nutrition and declares providing consultancy advice to legal firms involved in class action for infants developing NEC; all honoraria and consultancy fees were donated to charity. C.J.S. declares lecture honoraria from Nestlé Nutrition Institute. T.D.L. is the co-founder and CSO of Microbiotica. The other authors declare no competing interests.

## Additional information

**Extended data** is available for this paper at https://doi.org/10.1038/s41564-026-02297-4.

**Correspondence and requests for materials** should be addressed to Christopher J. Stewart.

¹Translational and Clinical Research Institute, Newcastle University, Newcastle upon Tyne, UK. ²Host–Microbiota Interactions Laboratory, Wellcome Sanger Institute, Hinxton, UK. ³Department of Microbes, Infection and Microbiomes, School of Infection, Inflammation and Immunology, College of Medicine and Health, University of Birmingham, Birmingham, UK. ⁴Institute of Microbiology and Infection, University of Birmingham, Birmingham, UK. ⁵Food, Microbiome and Health, Quadram Institute Bioscience, Norwich, UK. ⁶Faculty of Health and Life Sciences, Northumbria University, Newcastle upon Tyne, UK. ⁷Biosciences Institute, Newcastle University, Newcastle upon Tyne, UK. ⁸dsm-firmenich, Hørsholm, Denmark. ⁹Department of Gastroenterology, Newcastle upon Tyne Hospitals NHS Foundation Trust, Newcastle upon Tyne, UK. ¹⁰Norwich Medical School, University of East Anglia, Norwich, UK. ¹¹Newcastle Neonatal Service, Newcastle upon Tyne Hospitals NHS Foundation Trust, Newcastle upon Tyne, UK. ¹²Population Health Sciences Institute, Newcastle University, Newcastle upon Tyne, UK. ¹³Microbiology Department, Freeman Hospital, Newcastle upon Tyne Hospitals NHS Foundation Trust, Newcastle upon Tyne, UK. ¹⁴These authors contributed equally: Jonathan A. Chapman, Andrea C. Masi. ✉e-mail: christopher.stewart@newcastle.ac.uk

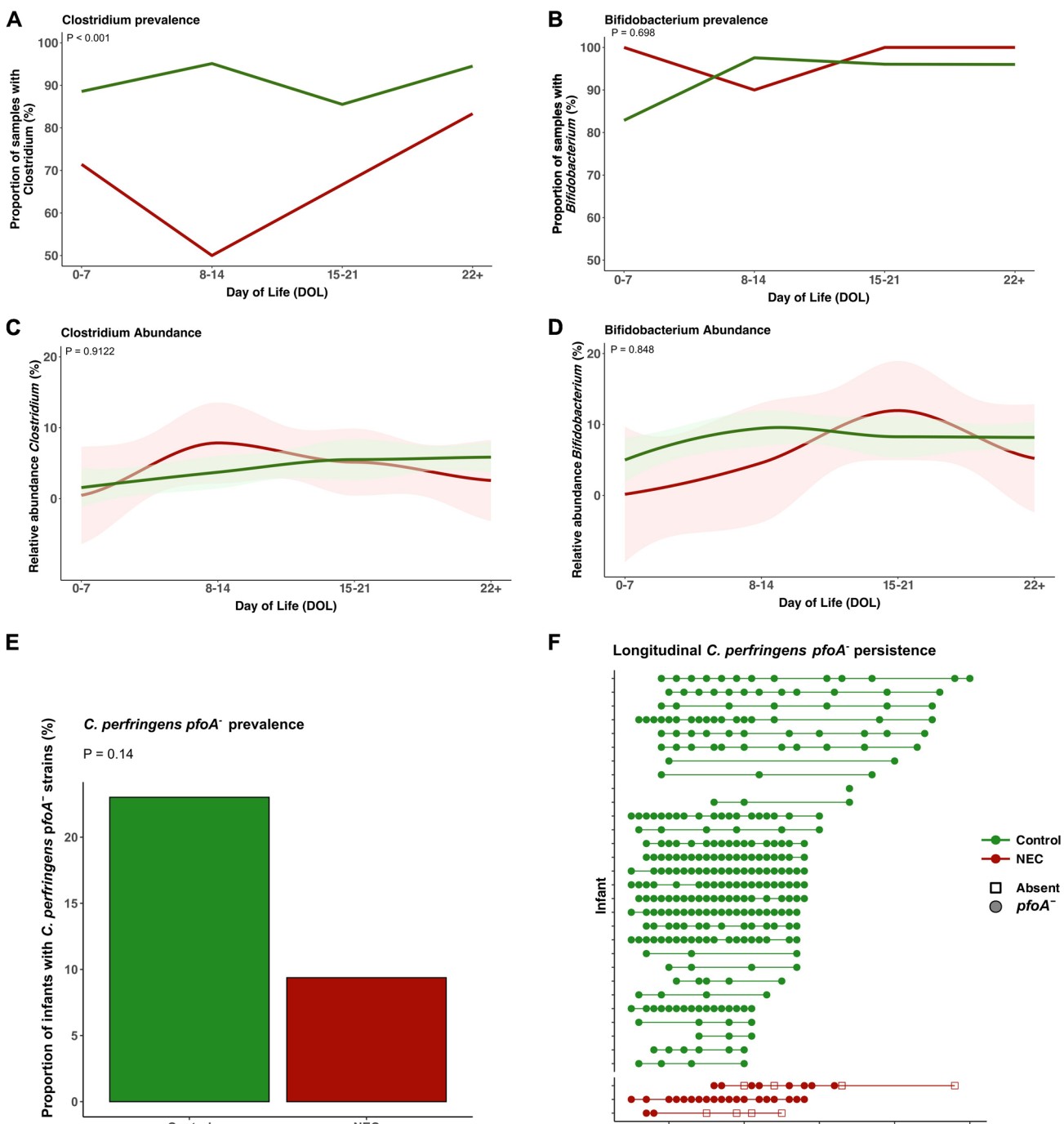

**Extended Data Fig. 1 | *Clostridium* colonisation patterns during early life, from two publicly available datasets. (A-D)** Prevalences and relative abundances of *Clostridium* and *Bifidobacterium* in NEC (n = 14) and control (n = 112) infants, who did not receive probiotics. Samples were binned based on day of life (DOL) as follows: 0-7 (CTRL: 70 samples, NEC: 7 samples); 8-14 (CTRL: 82 samples, NEC: 10 samples); 15-21 (CTRL: 76 samples, NEC: 9 samples); 22+ (CTRL: 78 samples, NEC: 6 samples). Infant groups were matched for gestational age (control: median = 28 weeks, IQR = 26-29; NEC: median = 27 weeks, IQR = 26-28; p = 0.10). Data from the previously published Mechanisms Affecting Gut of Preterm Infants (MAGPIE) Study[23]. **(A-B)** Statistical significance was calculated using a two-tailed z-test. **(C-D)** Statistical significance was calculated using a generalised linear model, followed by an ANOVA. **(A)** Prevalence of *Clostridium* **(B)** Prevalence of *Bifidobacterium* **(C)** Relative abundance of *Clostridium* **(D)** Relative abundance of *Bifidobacterium*. Analysis performed on publicly available data from the MAGPIE study[23]. **(E-F)** Prevalence and colonisation persistence of *C. perfringens pfoA⁻* in NEC (n = 32) and control (n = 126) infants. Samples from NEC infants were pre-diagnosis of NEC and control samples were matched to the same DOL range. Analysis performed on bacterial metagenome assembled genomes (MAGs) previously published by Olm *et al.*[25]. **(E)** Prevalence *C. perfringens pfoA⁻* in NEC versus control infants. Statistical significance was calculated using a two-tailed z-test. **(F)** Persistence of *C. perfringens pfoA⁻* colonisation in NEC versus control infants. Each circle or square represents a stool sample.

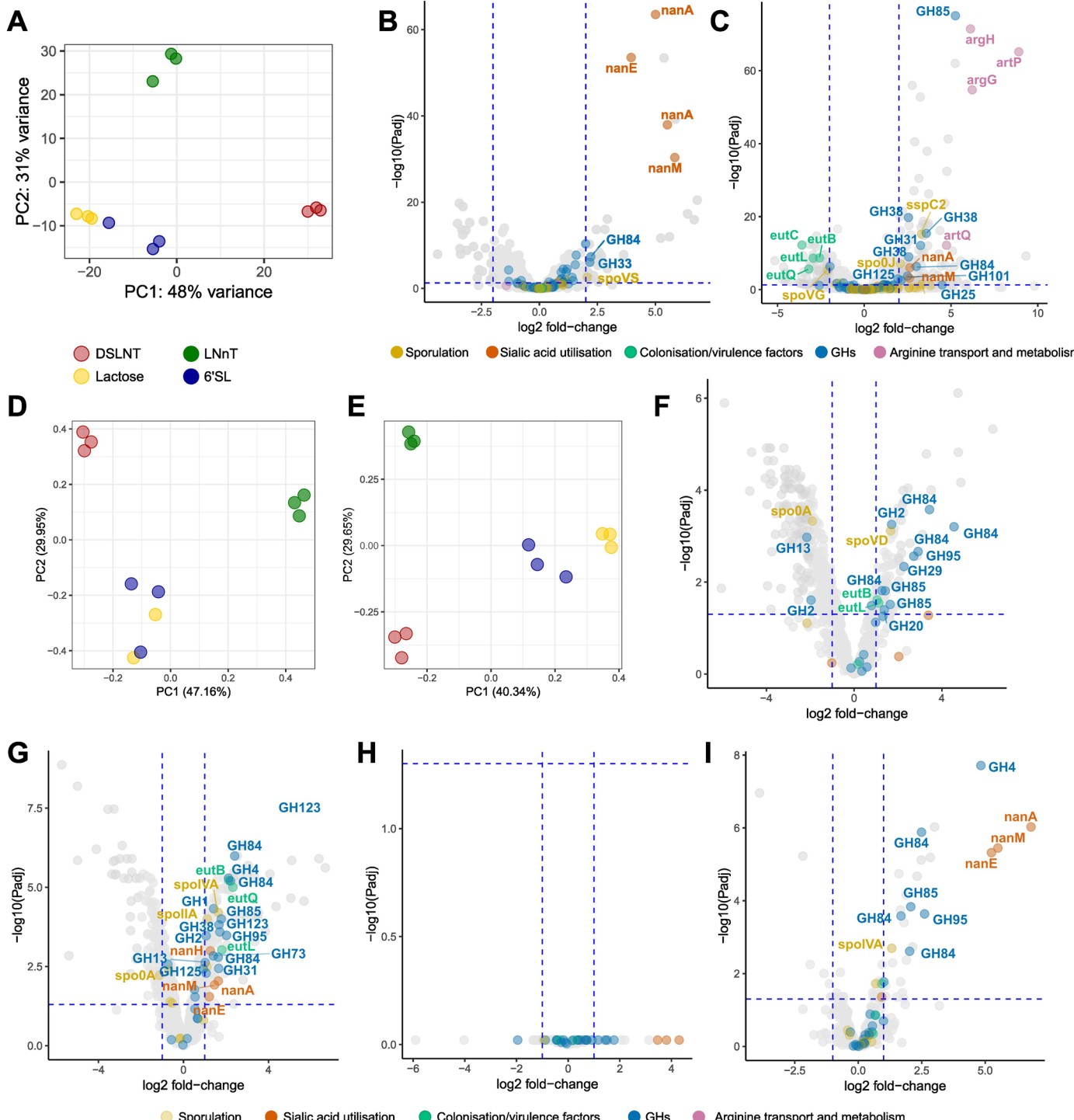

**Extended Data Fig. 2 | RNA-seq and proteomics analysis of AM1 grown on DSLNT, LNnT, 6'SL and lactose.** (**A**) Principal Components Analysis (PCA) of RNA-seq data from AM1 grown to exponential phase on DSLNT, LNnT, 6'SL and lactose. (**B-C**) Volcano plot of RNA-seq data for AM1 grown to exponential phase on 6'SL (**B**) and LNnT (**C**) compared to lactose. A positive log2 fold-change indicates upregulation on LNnT or 6'SL relative to lactose, while a negative fold change indicates downregulation on LNnt or 6'SL relative to lactose. Statistical significance was calculated using a two-tailed Wald test, followed by adjustment for multiple comparisons using the Benjamini-Hochberg method. (**D-E**) PCA of proteomics data on supernatant (**D**) and cell pellet (**E**) from AM1 grown to exponential phase on DSLNT, LNnT, 6'SL and lactose. (**F-G**) Volcano plot of proteomics data from supernatant (**F**) and pellet (**G**) of AM1 grown to exponential phase on LNnT compared to lactose. A positive log2 fold-change

indicates upregulation on LNnT relative to lactose, while a negative fold change indicates downregulation on LNnt relative to lactose. Statistical significance was calculated using a two-tailed moderated t-test, followed by adjustment for multiple comparisons using the Benjamini-Hochberg method. (**H-I**) Volcano plot of proteomics data from supernatant (**F**) and pellet (**G**) of AM1 grown to exponential phase on 6'SL compared to lactose. A positive log2 fold-change indicates upregulation on 6'SL relative to lactose, while a negative fold change indicates downregulation on 6'SL relative to lactose. Statistical significance was calculated using a two-tailed moderated t-test, followed by adjustment for multiple comparisons using the Benjamini-Hochberg method. LNnT, lacto-n-neotetraose; 6'-SL, 6'-sialyllactose; DSLNT, disialyllacto-N-tetraose; GH, glycoside hydrolase.

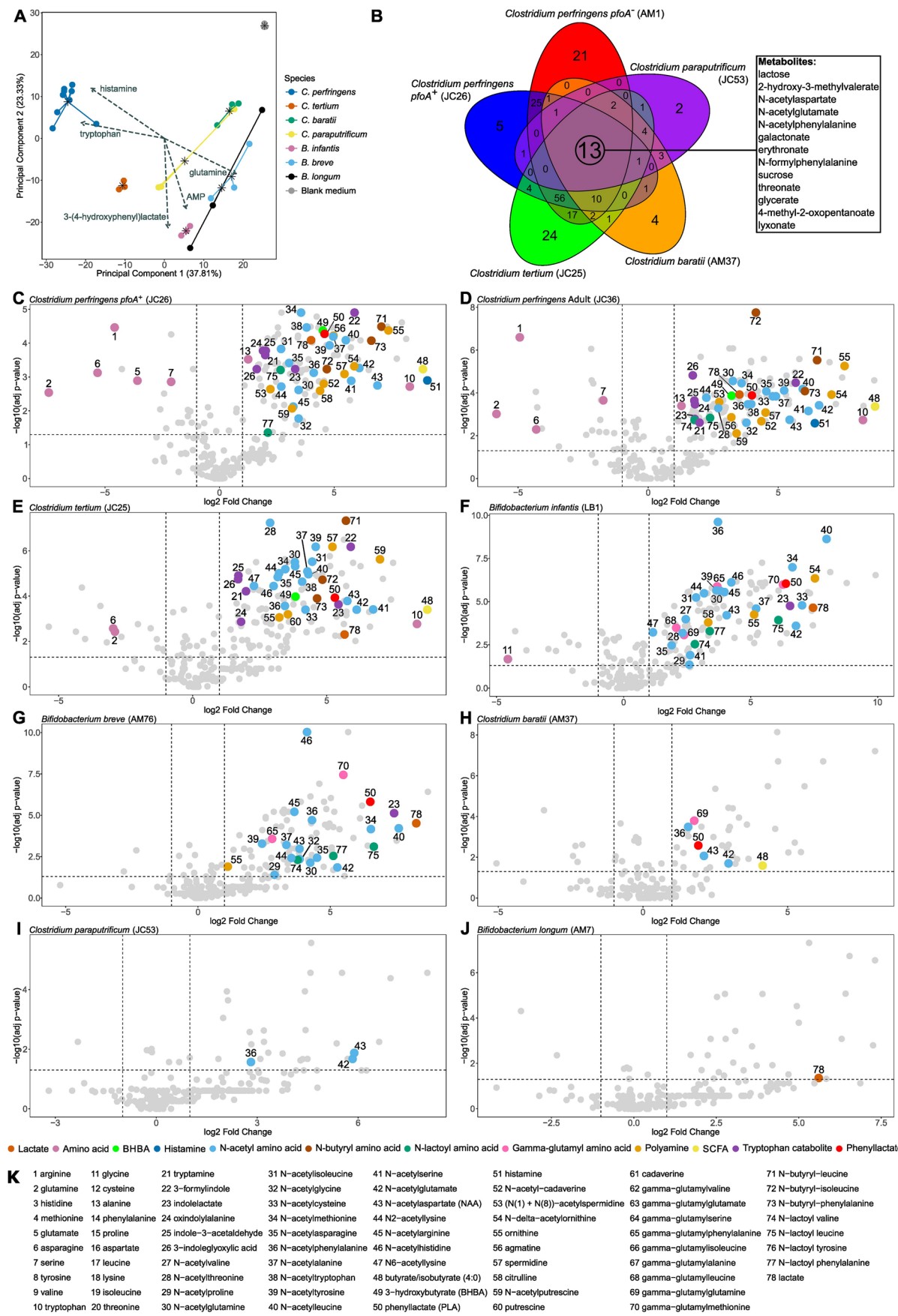

**Extended Data Fig. 3 | See next page for caption.**

**Extended Data Fig. 3 | Untargeted metabolomics analysis of cell free supernatants of isolates grown on strain-specific mixtures of human milk oligosaccharides (HMOs).** (**A**) PCA of untargeted metabolomics data for cell free supernatant (CFS) generated from bacterial cultures growing on glucose. Arrows indicate the top five metabolites by loadings magnitude. (**B**) Numbers of upregulated metabolites shared between the preterm infant-derived *Clostridium* strains, during their growth on mixtures of HMOs. (**C-K**) Metabolites detected in CFSs of isolates grown on strain-specific mixtures of the HMOs we found they can utilise (Fig. 1a) compared with blank ZMB1 medium. A positive log2 fold-change indicates production of metabolites by the strain, while a negative fold change indicates metabolite depletion. Metabolites of interest are highlighted and numbered. Statistical significance was calculated using a two-tailed moderated t-test, followed by adjustment for multiple comparisons using the Benjamini-Hochberg method. (**C**) Metabolites for CFS of *C. perfringens pfoA*+ JC26 growing on cocktail of the HMOs 6'SL and LNnT, compared with blank ZMB1 medium. (**D**) Metabolites for CFS of *C. perfringens* Adult JC36 growing on cocktail of the

HMOs 6'SL, LNnT, 2FL, LNT and LNFPI, compared with blank ZMB1 medium. (**E**) Metabolites for CFS of *C. tertium* JC25 growing on cocktail of the HMOs LNT and LNnT, compared with blank ZMB1 medium. (**F**) Metabolites for CFS of *B. infantis* LB1 growing on cocktail of the HMOs 6'SL, LNnT, 2'FL, LNT and LNFPI, compared with blank ZMB1 medium. (**G**) Metabolites for CFS of *B. breve* AM76 growing on cocktail of the HMOs LNT and LNnT, compared with blank ZMB1 medium. (**H**) Metabolites for CFS of *C. baratii* AM37 growing on cocktail of the HMOs LNT and LNnT, compared with blank ZMB1 medium. (**I**) Metabolites for CFS of *C. paraputrificum* JC53 growing on cocktail of the HMOs LNT and LNnT, compared with blank ZMB1 medium. (**J**) Metabolites for CFS of *B. longum* AM7 growing on the HMO LNT, compared with blank ZMB1 medium. (**K**) Key for labelled metabolites of interest. HMOs, human milk oligosaccharides; LNT, lacto-N-tetraose; LNnT, lacto-N-neotetraose; LNFP I, lacto-N-fucopentaose, 6'-SL, 6'-sialyllactose; 2'FL, 2' fucosyllacotse; BHBA, beta-hydroxybutyric acid; SCFA, short chain fatty acid.

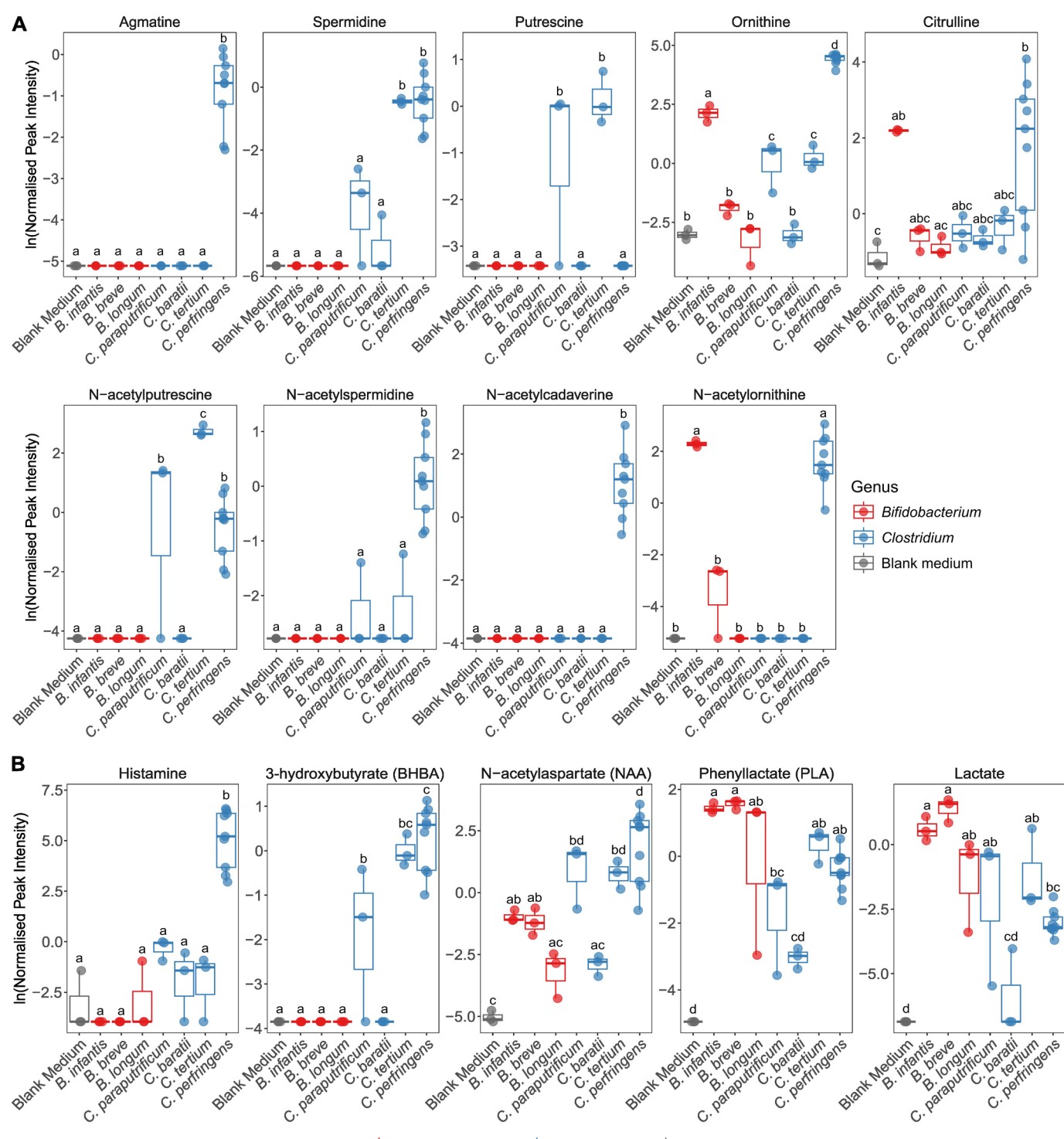

**Extended Data Fig. 4 | Levels of metabolites of interest produced by *Bifidobacterium* and *Clostridium* spp. grown on human milk oligosaccharides (HMOs) (n = 30).** Isolates were grown on strain-specific mixtures of the HMOs we found they can utilise (Fig. 1a). Statistical comparisons were performed using an ANOVA, followed by adjustment for multiple comparisons using two-tailed Tukey's HSD method. (**A**) Levels of polyamines produced by *Bifidobacterium* and *Clostridium* spp. during growth on HMOs. Conditions with the same letters are not significantly different. (**B**) Levels of neuromodulatory, immunomodulatory or antimicrobial metabolites produced by *Bifidobacterium* and *Clostridium* spp. during growth on HMOs. Conditions with the same letters are not significantly different. For all boxplots: centre line, median; box limits, upper and lower quartiles; whiskers, 1.5x interquartile range.

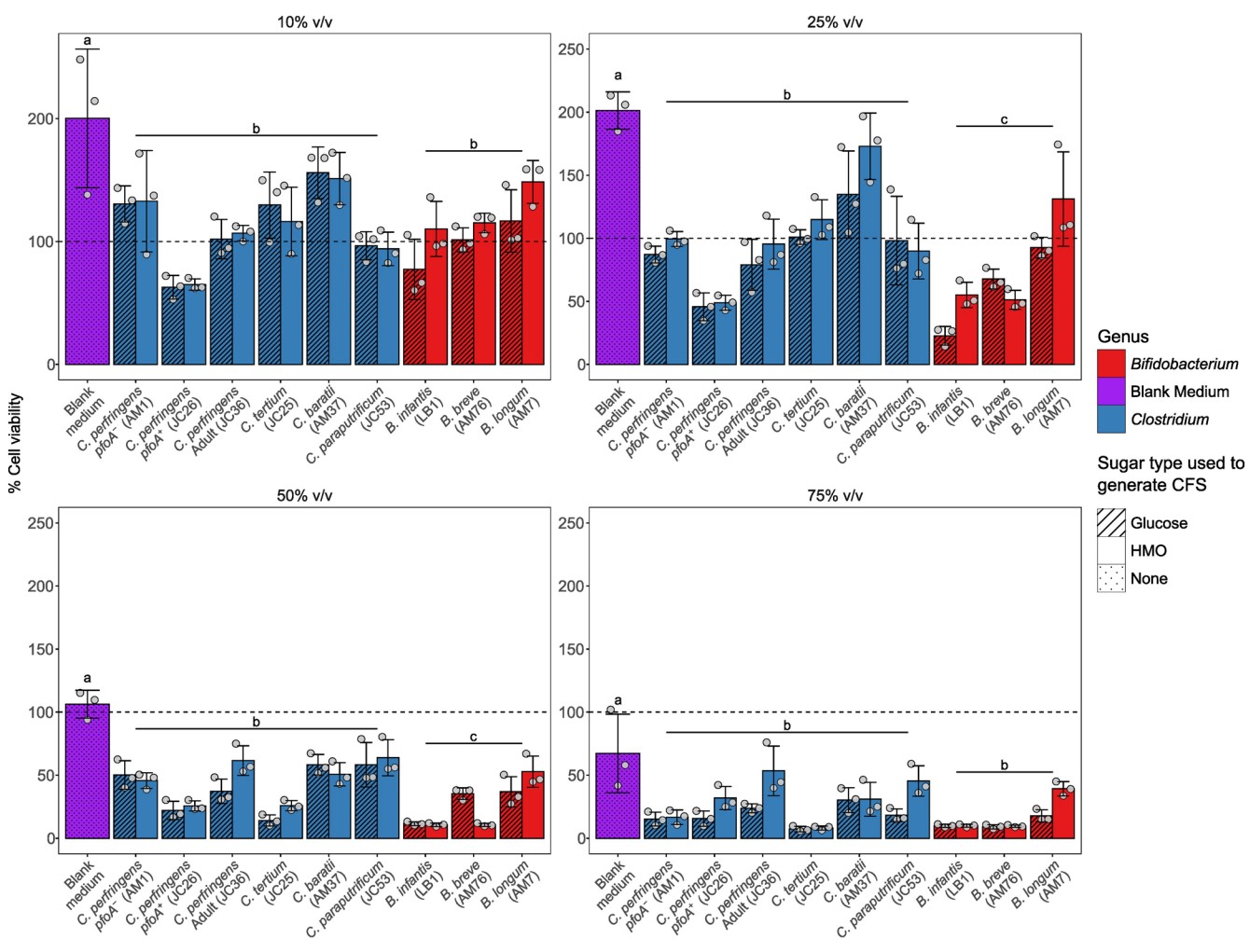

**Extended Data Fig. 5 | MTS tetrazolium dye cell viability assay data showing toxicity of cell free supernatants (CFSs) to Caco-2 cells (n = 3 per treatment).** Statistical comparisons were performed using average genus values, with an ANOVA followed by adjustment for multiple comparisons using two-tailed Tukey's HSD method. Groups with the same letters are not significantly different. Values are shown as % of the viability measured for Caco-2 cells incubated in regular DMEM. Data are presented as mean values +/- standard deviation. CFS, cell free supernatant; HMO, human milk oligosaccharide.

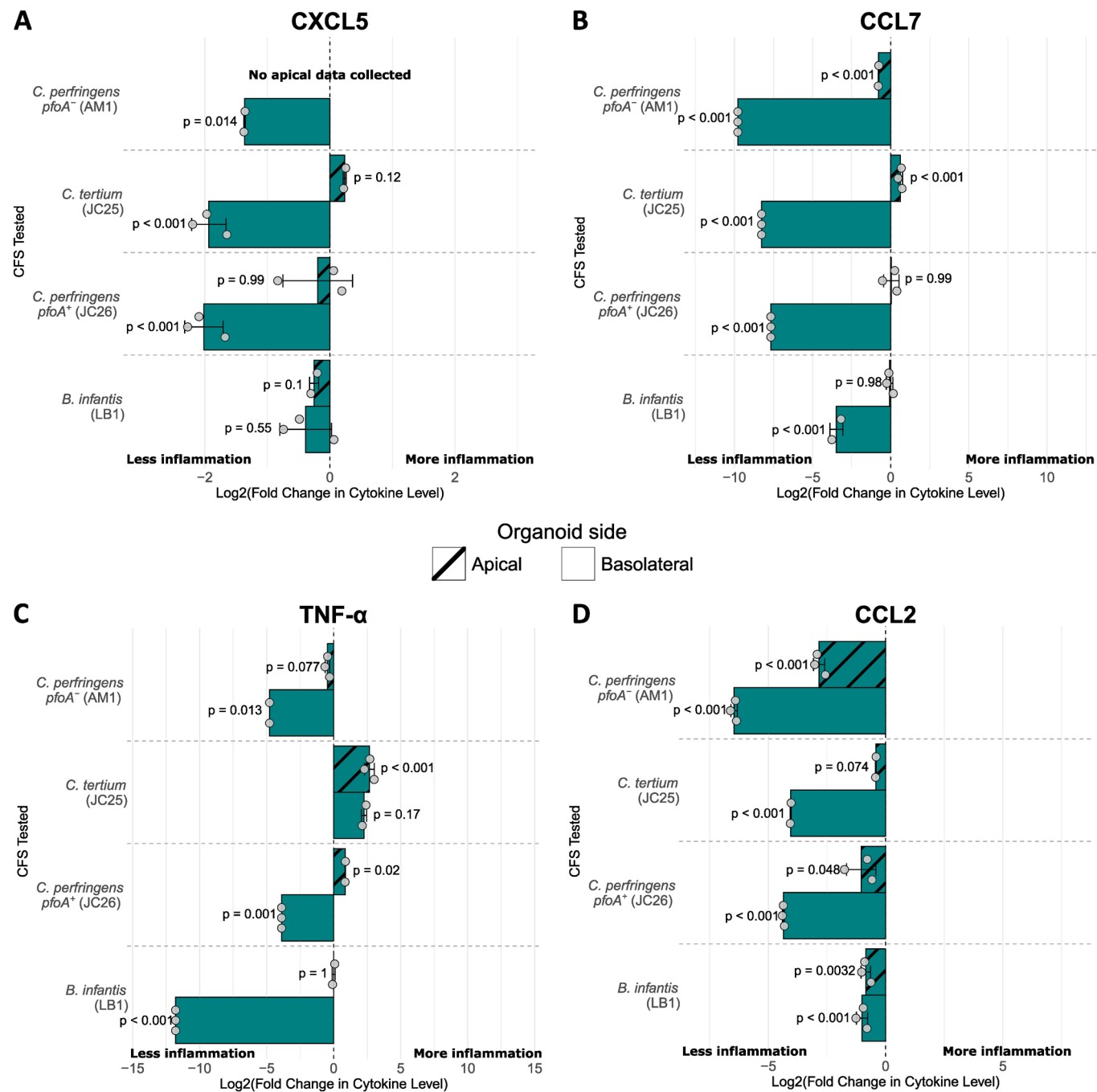

**Extended Data Fig. 6 | Changes in organoid monolayer apical and basolateral cytokine secretion during combined treatment with both 25% cell free supernatant (CFS) and inflammatory stimuli.** Data has been converted to fold change vs "Stimuli" condition and log2 transformed for plotting (n = 4-6 to calculate fold changes, per CFS tested). P values are for the differences between the unprocessed detected cytokine levels. Data are presented as mean values +/- standard deviation. Statistical comparisons were performed using an ANOVA, followed by Dunnett's Test to adjust for multiple comparisons, whereby cytokine secretion from "Stimuli only" was used as the control. (**A**) CXCL5 (**B**) CCL7 (**C**) TNF-α (**D**) CCL2 CFS, cell free supernatant.

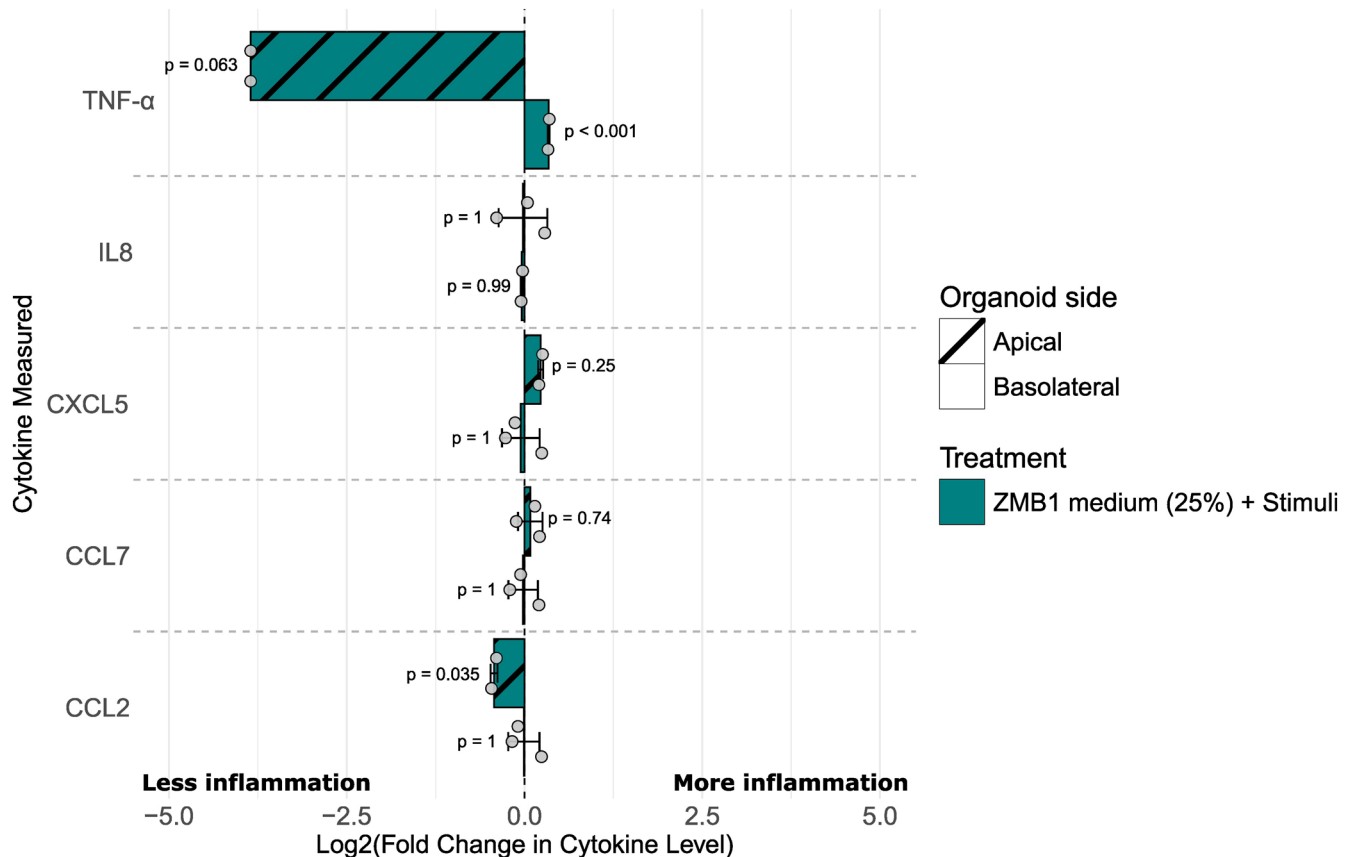

**Extended Data Fig. 7 | Changes in organoid monolayer apical and basolateral cytokine secretion during treatment with ZMB1 bacterial growth medium only, at 25% v/v concentration.** Data has been converted to fold change vs "Stimuli" condition and log2 transformed for plotting (n = 4-6 to calculate fold changes). P values are for the differences between the unprocessed detected cytokine levels. Data are presented as mean values +/- standard deviation. Statistical comparisons were performed using an ANOVA, followed by Dunnett's Test to adjust for multiple comparisons, whereby cytokine secretion from "Stimuli only" was used as the control.

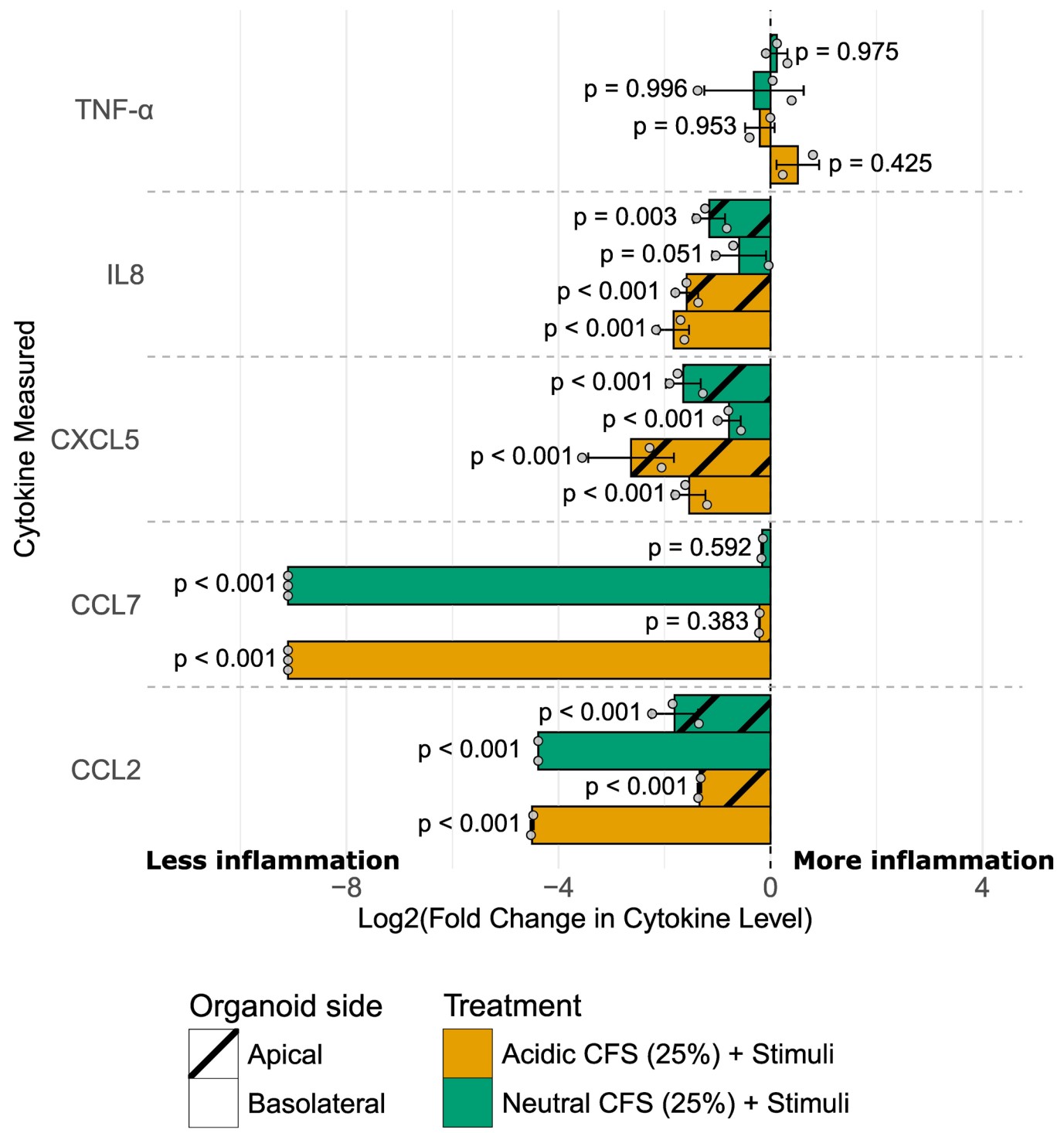

**Extended Data Fig. 8 | Changes in organoid monolayer apical and basolateral cytokine secretion during treatment with acidic and neutral pH *C. perfringens* AM1 cell free supernatant, at 25% v/v concentration.** Data has been converted to fold change vs "Stimuli" condition and log2 transformed for plotting (n = 4-6 to calculate fold changes, per CFS tested). P values are for the differences between the unprocessed detected cytokine levels. Data are presented as mean values +/- standard deviation. Statistical comparisons were performed using an ANOVA, followed by Dunnett's Test to adjust for multiple comparisons, whereby cytokine secretion from "Stimuli only" was used as the control. CFS, cell free supernatant.

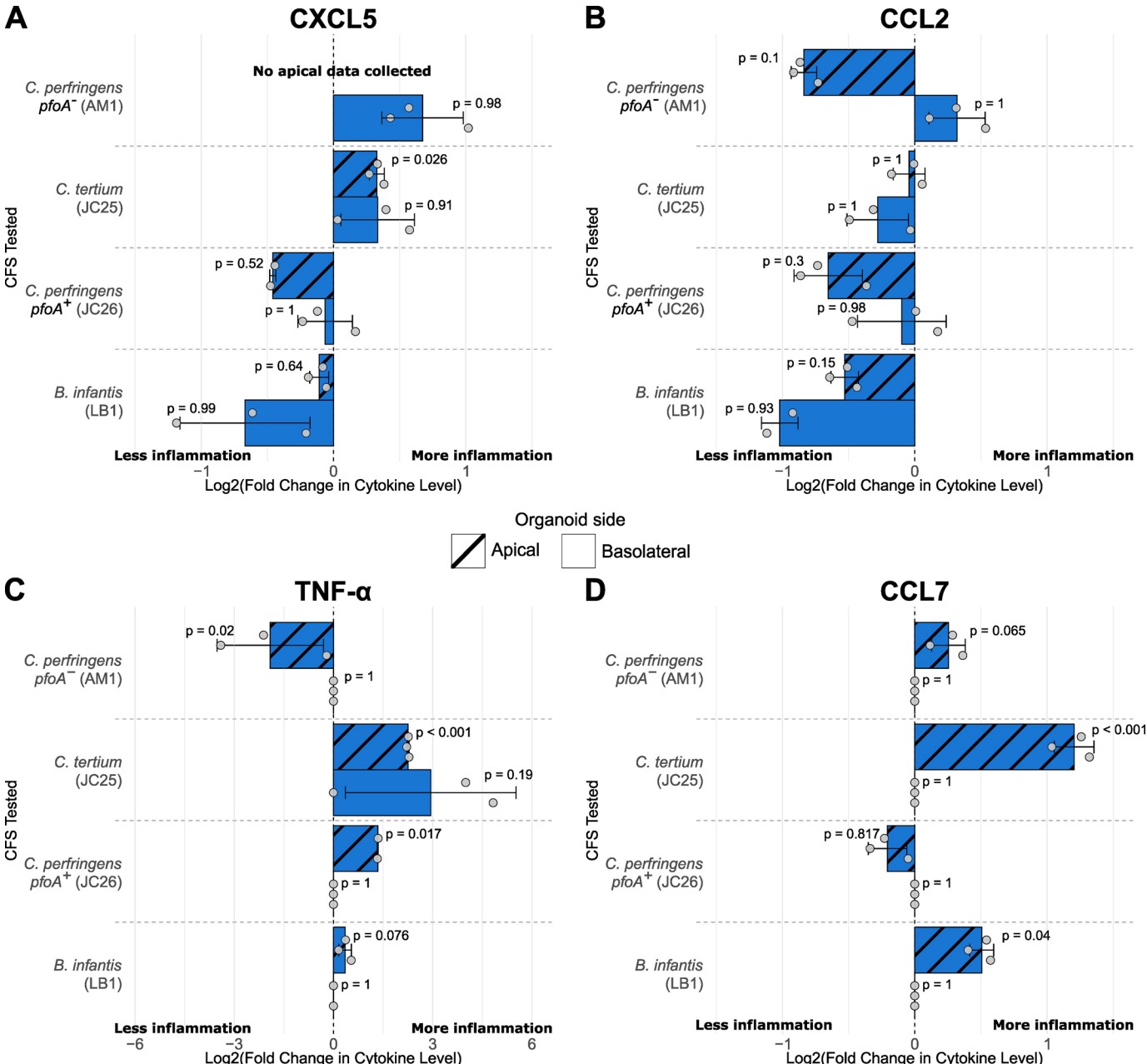

**Extended Data Fig. 9 | Changes in organoid monolayer apical and basolateral cytokine secretion during treatment with cell free supernatant only.** Data has been converted to fold change vs the 'no treatment' control and log2 transformed for plotting (n = 4-6 to calculate fold changes, per CFS tested). P values are for the differences between the unprocessed detected cytokine levels. Data are presented as mean values +/- standard deviation. Statistical comparisons were performed using an ANOVA, followed by Dunnett's Test to adjust for multiple comparisons, whereby cytokine secretion from "No treatment" was used as the control. (**A**) CXCL5 (**B**) CCL2 (**C**) TNF-α (**D**) CCL7. CFS, cell free supernatant.

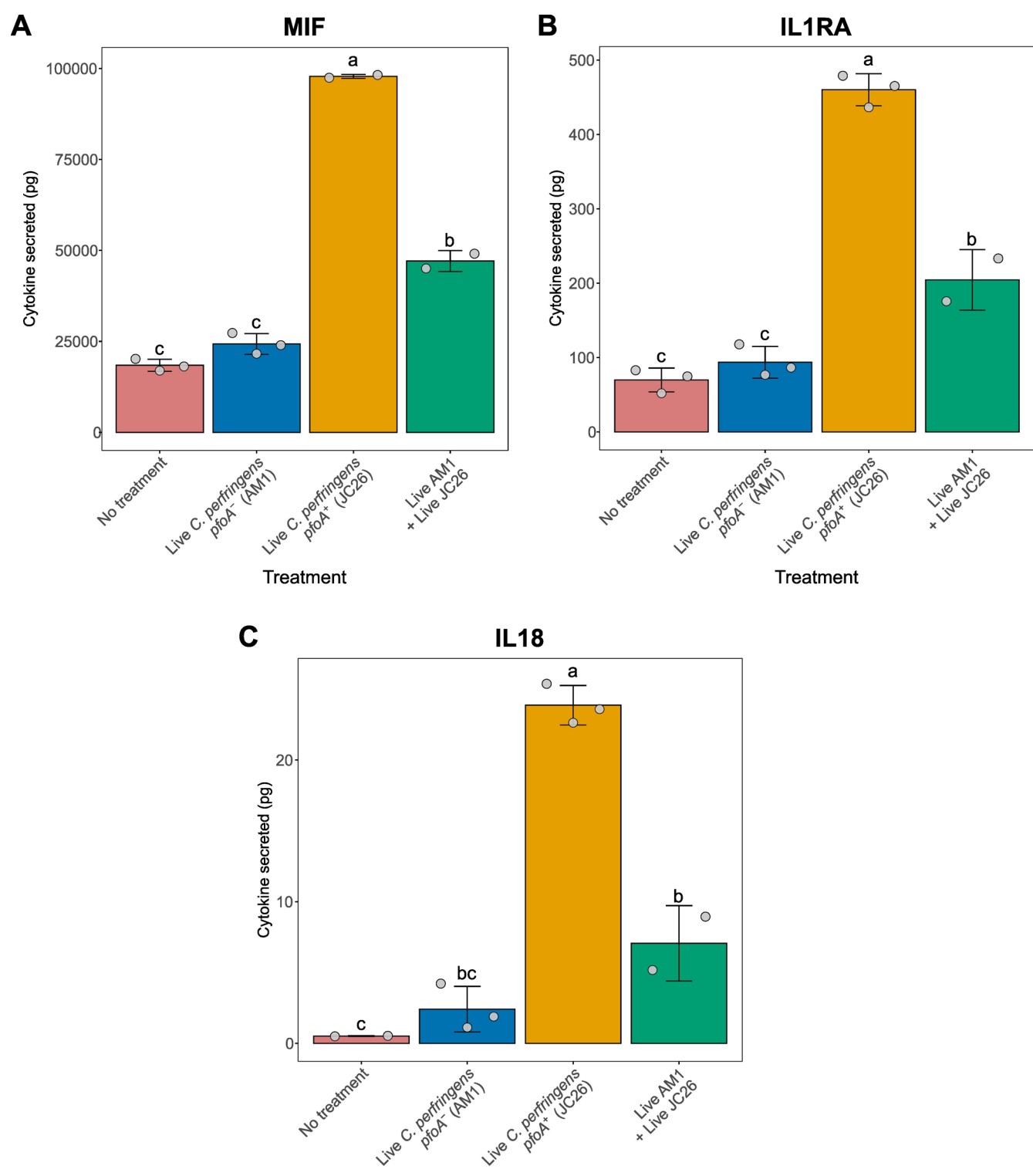

**Extended Data Fig. 10 | Apical secretion of cytokines from organoid monolayers treated with *pfoA⁻* and *pfoA⁺* *Clostridium perfringens* isolates (n = 2-3 per treatment).** Conditions with the same letters are not significantly different. Data are presented as mean values +/- standard deviation. Statistical comparisons were performed using an ANOVA, followed by adjustment for multiple comparisons using two-tailed Tukey's HSD method. (**A**) Apical MIF secretion from organoid monolayers treated with live *C. perfringens*. (**B**) Apical IL1RA secretion from organoid monolayers treated with live *C. perfringens*. (**C**) Apical IL18 secretion from organoid monolayers treated with live *C. perfringens*.

# Reporting Summary

## Statistics

For all statistical analyses, confirm that the following items are present in the figure legend, table legend, main text, or Methods section.

| n/a | Confirmed | |
|---|---|---|
| ☐ | ☒ | The exact sample size (*n*) for each experimental group/condition, given as a discrete number and unit of measurement |
| ☐ | ☒ | A statement on whether measurements were taken from distinct samples or whether the same sample was measured repeatedly |
| ☐ | ☒ | The statistical test(s) used AND whether they are one- or two-sided *Only common tests should be described solely by name; describe more complex techniques in the Methods section.* |
| ☒ | ☐ | A description of all covariates tested |
| ☐ | ☒ | A description of any assumptions or corrections, such as tests of normality and adjustment for multiple comparisons |
| ☐ | ☒ | A full description of the statistical parameters including central tendency (e.g. means) or other basic estimates (e.g. regression coefficient) AND variation (e.g. standard deviation) or associated estimates of uncertainty (e.g. confidence intervals) |
| ☐ | ☒ | For null hypothesis testing, the test statistic (e.g. *F*, *t*, *r*) with confidence intervals, effect sizes, degrees of freedom and *P* value noted *Give P values as exact values whenever suitable.* |
| ☒ | ☐ | For Bayesian analysis, information on the choice of priors and Markov chain Monte Carlo settings |
| ☒ | ☐ | For hierarchical and complex designs, identification of the appropriate level for tests and full reporting of outcomes |
| ☒ | ☐ | Estimates of effect sizes (e.g. Cohen's *d*, Pearson's *r*), indicating how they were calculated |

*Our web collection on statistics for biologists contains articles on many of the points above.*

## Software and code

Policy information about availability of computer code

| Data collection | No software were used for data collection. |
|---|---|
| Data analysis | For analyses using R, v4.4.0 |
| | Heatmaps were all visualised with pheatmap v1.0.12 |
| | Other plots were visualised with ggplot2 v3.5.0 |
| | ANOVA: aov() function in R |
| | Tukey's test: TukeyHSD() function in R, and HSD.test() from agricolae v1.3-7 |
| | Dunnett's test: glht() from multcomp v1.4-29 |
| | Unpaired T-test: t.test() function in R |
| | |
| | Whole genome sequencing: |
| | Assembly: SPAdes |
| | Quality: checkm v1.1.3 and GUNC v1.0.5 |
| | Taxonomic assignment: gtdb-tk v2.3.2 |
| | Annotation: prokka v1.14 |
| | Core gene alignment construction: panaroo v1.2.8 |
| | Phylogeny: IQ-TREE v2.0.5 |
| | Phylogenetic tree visualisation: iTOL v6.0 |
| | Clostridium toxin and colonisation factor gene screen: ABRicate v1.0.1 |
| | Genome sizes: sequence-stats v1.0 |
| | Mash distance sequence tree: Mashtree v1.2.0 |

Distance tree visualisation: iTOL v6.0

Screening Clostridium perfringens MAGs for pfoA: BlastN (v2.16.0)

RNAseq of C. perfringens:
Read alignment to genome: Bowtie v2.4.5
Gene level count data: HTSeq v2.0.8
Differential expression analysis: DESeq2 v1.44.0

Proteomics:
Acquired data processing: DIA-NN v1.8
Differential abundance analyses: Limma v3.6.4

Metabolomics data analysis:
PERMANOVA: Vegan v2.6.8
Differential abundance analysis: Limma v3.56.2
Venn diagram of shared metabolites: VennDiagram v1.7.3

CFS microbial activity assay
Area under the curve calculation: caTools v1.18.2

Seahorse mitochondrial stress test assay:
Raw data processing and export: Wave v2.6.3 (Agilent Technologies)

For manuscripts utilizing custom algorithms or software that are central to the research but not yet described in published literature, software must be made available to editors and reviewers. We strongly encourage code deposition in a community repository (e.g. GitHub). See the Nature Portfolio guidelines for submitting code & software for further information.

# Data

Policy information about availability of data

All manuscripts must include a data availability statement. This statement should provide the following information, where applicable:

- Accession codes, unique identifiers, or web links for publicly available datasets
- A description of any restrictions on data availability
- For clinical datasets or third party data, please ensure that the statement adheres to our policy

Source data are provided with this paper. All cytokine data from ELISA and multiplex MSD assays are provided in Supplementary tables 5-11. The RNA-seq data have been deposited in the Sequencing Read Archive (SRA) under study accession number PRJNA1214204. The proteomics datasets are deposited in MassIVE under submission ID MSV000096907. Sequencing reads for de novo genomes have been deposited in the European Nucleotide Archive (ENA) under accession number ERP187615.

# Research involving human participants, their data, or biological material

Policy information about studies with human participants or human data. See also policy information about sex, gender (identity/presentation), and sexual orientation and race, ethnicity and racism.

| | |
|---|---|
| Reporting on sex and gender | We do not report on sex and gender. |
| Reporting on race, ethnicity, or other socially relevant groupings | We do not report on these variables. |
| Population characteristics | We use intestinal epithelial organoids derived from the resected tissue of a preterm infant. |
| Recruitment | Preterm infants (born at <32 weeks gestation) were born or transferred to a single tertiary level Neonatal Intensive Care Unit in Newcastle upon Tyne, UK, and participated in the Supporting Enhanced Research in Vulnerable Infants (SERVIS) study (REC10/H0908/39) after written informed parental consent. Parents are approached in the first week of life when the study is explained by a member of the research team. Parents were approached in the first week of life when the study was explained by a member of the research team. Parents were given the option to opt in or out of each specific aspect on a single consent form. Parents of infants who were initially extremely unwell were only approached when they were considered stable by the bedside nurse and medical team. Approaches were by experienced neonatal staff familiar with the studies being described, sample collection, and parental communication. Written signed consent was obtained after the parents have had time to consider the information. Stool samples were regularly collected from nappies/diapers of preterm infants into sterile collection pots by nursing staff. Breast milk samples were collected from residuals from an infant's feeding systems. Samples were initially stored at -20°C before being transferred to -80°C for long term storage. Intestinal tissue samples used to generate organoid cell lines were salvaged following surgical resection. Participants are not compensated for donation. |
| Ethics oversight | The study protocol was approved by Newcastle Hospitals NHS Foundation Trust (NUTH), NRES Committee North East and N Tyneside 2 10/H0908/39, and the research complies with all relevant ethical regulations. |

Note that full information on the approval of the study protocol must also be provided in the manuscript.

# Field-specific reporting

Please select the one below that is the best fit for your research. If you are not sure, read the appropriate sections before making your selection.

☒ Life sciences      ☐ Behavioural & social sciences      ☐ Ecological, evolutionary & environmental sciences

For a reference copy of the document with all sections, see nature.com/documents/nr-reporting-summary-flat.pdf

# Life sciences study design

All studies must disclose on these points even when the disclosure is negative.

| | |
|---|---|
| Sample size | We screened the abilities of 29 bacterial isolates, mostly from preterm infant stool, to grow on six different HMOs, and glucose and lactose. These species were obtained by untargeted cultivation and represent a median of 80% (interquartile range 61%-91%) of all relative microbial abundance observed in preterm infants. Bacterial strains identified as being of interest, based on HMO utilisation profile, were taken forward for systematic experimentation. One organoid cell line was used, with all experiments performed in triplicate, which we deemed important for maintaining a consistent host genetic background for comparing different treatment conditions. For all laboratory experiments, no sample size calculations were performed. Experiments were conducted with n = 3 per condition, based on standard laboratory practice for precision, enabling the identification of and potential correction for outliers. |
| Data exclusions | For Seahorse mitochondrial stress test assays, organoid trans-epithelial electrical resistances and cytokine assays, outliers were removed from datasets. Outliers were identified by calculating robust/modified z scores for each data point. This method uses the median to calculate median absolute deviation. Any data point with a robust z score greater than +3.5 or less than -3.5 was removed from the dataset. At least two replicates were always retained per group. For this work, we wanted an unbiased, numerical measure for outlier assessment. We ultimately selected robust z scores, as they are more resistant to outlier distortion, due to reliance on median absolute deviation, making this method more appropriate these small datasets. |
| Replication | All experiments were performed with a minimum of 3 replicates per group, with outliers then removed following the methodology described above. |
| Randomization | Randomization was not required in this study, as the experimental design involved controlled manipulations of treatment conditions. Responses were measured in a systematic manner, with each treatment applied to a consistent bacterial strain and organoid cell line. This approach ensured reliable comparisons between conditions without the need for random assignment. |
| Blinding | Blinding was not used in this study, as experimental design involved controlled manipulations of treatment conditions, with each treatment applied to a consistent bacterial strain or organoid cell line. |

# Reporting for specific materials, systems and methods

We require information from authors about some types of materials, experimental systems and methods used in many studies. Here, indicate whether each material, system or method listed is relevant to your study. If you are not sure if a list item applies to your research, read the appropriate section before selecting a response.

## Materials & experimental systems

| n/a | Involved in the study |
|---|---|
| ☒ | ☐ Antibodies |
| ☐ | ☒ Eukaryotic cell lines |
| ☒ | ☐ Palaeontology and archaeology |
| ☒ | ☐ Animals and other organisms |
| ☒ | ☐ Clinical data |
| ☒ | ☐ Dual use research of concern |
| ☒ | ☐ Plants |

## Methods

| n/a | Involved in the study |
|---|---|
| ☒ | ☐ ChIP-seq |
| ☒ | ☐ Flow cytometry |
| ☒ | ☐ MRI-based neuroimaging |

# Eukaryotic cell lines

Policy information about cell lines and Sex and Gender in Research

| | |
|---|---|
| Cell line source(s) | This study used a preterm intestinal-derived organoid model which was established in the Stewart Lab, Newcastle University using resected surgical ileum tissue that was obtained from the neonatal intensive care unit of the Royal Victoria Infirmary, Newcastle. Informed consent was obtained by parents. The laboratory ID of this organoid cell line is NCL 27 and it was established from a male patient born at 24 weeks gestation, using ileum tissue salvaged from surgery performed on day of life 10 due to the development of necrotising enterocolitis. |
| Authentication | The cell line was not authenticated. |
| Mycoplasma contamination | Cell lines were not tested for mycoplasma contamination. |

| Commonly misidentified lines<br>(See ICLAC register) | No commonly misidentified cell lines were used. |

## Plants

| Seed stocks | *Report on the source of all seed stocks or other plant material used. If applicable, state the seed stock centre and catalogue number. If plant specimens were collected from the field, describe the collection location, date and sampling procedures.* |
| Novel plant genotypes | *Describe the methods by which all novel plant genotypes were produced. This includes those generated by transgenic approaches, gene editing, chemical/radiation-based mutagenesis and hybridization. For transgenic lines, describe the transformation method, the number of independent lines analyzed and the generation upon which experiments were performed. For gene-edited lines, describe the editor used, the endogenous sequence targeted for editing, the targeting guide RNA sequence (if applicable) and how the editor was applied.* |
| Authentication | *Describe any authentication procedures for each seed stock used or novel genotype generated. Describe any experiments used to assess the effect of a mutation and, where applicable, how potential secondary effects (e.g. second site T-DNA insertions, mosiacism, off-target gene editing) were examined.* |

