## [Peer Review File · Nature Microbiology]

Clostridia from preterm infants metabolise human milk oligosaccharides to suppress pathobionts and modulate intestinal function in organoids

Corresponding Author: Professor Christopher Stewart

Version 0:

Reviewer comments:

Reviewer #1

(Remarks to the Author)

This is a thorough attempt to define the range of putatively protective effector functions of toxin perfringolysin O-deficient *C. perfringens* isolated from preterm infants. The strengths of the work include the broad range of phenotypic assays used to define the beneficial function of toxin perfringolysin O-deficient *C. perfringens*, the highly likely relevance of the assays chosen (e.g., activity against pathobionts, effects on cells in vitro), the inter-play between pfoA- *C. perfringens* and disialyllacto-N-tetraose, which has been considered to protect against a devastating inflammatory disorder of the gut in infants born preterm. The limitation of this work is that this particular species and pfoA- *C. perfringens* strain have not been as critically protective in many or most cases of gut inflammation in infancy, or in the maintenance of gut functionality.

1. Fig 1A – how extensive and systematic was the survey? The comment, “We screened the abilities of 29 bacterial isolates, mostly from preterm infant stool, to grow on six different HMOs, and glucose and lactose (Fig. 1A)” provides little guidance about how prevalent colonization with the organisms of interest is, and the systematic nature of this screening. The premise of this paper, and the justification for the experiments, are based on the organism of interest, pfoA- *C. perfringens*, being truly associated with protection from NEC, and being associated with utilization of an HMO associated with protection NEC (but not clearly proven – see

J Perinatol. 2024 Sep 19. doi: 10.1038/s41372-024-02125-9).

2. Line 93: *B. infantis* is often used, but it is not considered to be therapeutic in the context of preventing or treating any disease. Given current knowledge, its use does seem to support the growth of microbes that are not considered pathogenic, but it is not yet a therapy.

3. Line 99: The authors state “found *Clostridium* was prevalent in preterm infants, but at significantly lower rates in those diagnosed with NEC compared with time-matched 101 healthy controls ($p < 0.001$; fig S2A)” as a finding that supports their contention that *C. perfringens* could be associated with protection against NEC. This figure, however, does not lend strong support to this contention. It seems like all children born after < 32 weeks’ gestation were enrolled in this study, but that encompasses ~9 weeks’ span of gestation among the study group. The more premature a child is born, the longer it takes to develop an anaerobic predominant microbial community, and the *Clostridium*-associated protection could merely reflect the degree of prematurity. Much more data are needed to use these data in support of this contention. The pre-NEC samples in FigS2A are not well portrayed. For example, the x-axis has overlapping days of life of sampling, and the binning of patients into 1 week categories requires more elaboration. The caption states “Figure S2. Prevalences and relative abundances of *Clostridium* and *Bifidobacterium* in NEC (n= 34) and control (n= 306) infants, who did not receive probiotics.” which would be a fairly large pre-NEC cohort. However, the insert state that these are samples, presumably from repeated samplings of a smaller number of children. Finally, labels of the X – axis add up to 32, not 34.

This concern might seem petty, except that the major appeal of this work is that *C. perfringens* protects the human infant host from inflammation of the gut that could lead to NEC but the data underlying this assertion are not strong.

Reviewer #2

(Remarks to the Author)

The authors describe novel HMO-utilizing bacteria in preterm infants beyond *Bifidobacteria*. *C. perfringens* produced metabolites prevented pathobiont growth and reduced inflammatory responses in preterm human organoid cultures even in absence of live bacteria. In addition, *C. perfringens* did not inhibit the growth of *Bifidobacteria*. The authors postulate this could provide new insights into postbiotics that could be used to support intestinal health in preterm infants.

Major points

I understand the authors resistance to an animal model to further elucidate the protective effects of the postbiotics given the complexity of the mouse or rat NEC models. However, their claims would be strengthened if they had either investigated other metabolites identified in their screen beyond butyrate, which as they mention in the discussion has already been shown to reduce NEC incidence in a rat model or provided experiments with metabolites only in preterm organoids to further promote their claim of the benefits of synthetic postbiotics, rather than CFS.

Minor points:

Two major cytokines involved in intestinal injury are missing from their panel (IL1b, IL-6), which have been implicated in NEC. Other cytokines cannot be modeled since their source is from the immune compartment.

Fig 2 write the HMO used for the growth conditions to measure the SCFA in the figure legend.

Fig.3 since the authors claim they want to use postbiotics rather than CFS from a bacteria or live bacteria the Acetate:Butyrate cocktail at pH:6 should be tested for growth inhibition of Bifido.

Fig4. Put the name of the cytokine and treatment condition in the figure for D-E so that it easy to grasp the point

S10 CCL2 data missing

S11 TNFa data does not match S9

S9, S12, S14 please place the name of the cytokine either on the y axis or the title of the graph for easier interpretation of the data

Genomes have not been deposited. Is this due to pending patent?

Growth curves should be analyzed with a more appropriate statistical method like growthcurver (PMID: 27094401) to estimate growth rate and percent inhibition quotients can be calculated to compare growth inhibition between the different treatment options not just for the pathobionts, but also for the bifido.

Reviewer #3

(Remarks to the Author)

The manuscript by Chapman et al describes the characterization of a *C. perfringens* strain lacking toxin genes and its ability to utilize human milk oligosaccharides (HMOs) in the context of preterm infant health. The manuscript is well written and clearly articulated.

There are some issues with the manuscript, both in framing and in the structural arguments presented, that should be addressed. Some minor points are also noted below.

Major issues:

1. The manuscript frames the rationale for the interest in this pfoA- *C. perfringens* as a potential probiotic in the context of preterm infants. This is provocative and intriguing, but requires more support. The authors cite some studies linking changes to the neonatal gut microbiome that are favorable for the addition of *Clostridium* to this community, but there is also evidence that clostridia (including *C. perfringens*) may exacerbate necrotising enterocolitis (NEC), with some evidence suggesting this may come about by the production of butyrate (see Ferraris et al 2023; Cassir et al 2019; de la Cochetière et al 2004). Others have suggested that formic acid (another fermentative product of clostridia) is responsible for NEC (see Casaburi et al 2022). Both potential causes include human and animal model data that is more extensive and more congruent than the results presented in this manuscript. I advise that the authors reframe this as an understanding of one infant gut microbe rather than drift toward the description or application of this organism as a therapeutic.
2. The description of the use of HMOs by clostridia suggests – in the summary and introduction – that this is a specific adaptation of these species or strains to the infant gut ecosystem. The alternative hypothesis may then be that these clostridia utilize mucin glycans, which bear structural similarity to HMOs, and they can repurpose these for the infant gut (e.g., see Van den Abbelle et al. 2013; Raimondi et al 2021; MacMillan et al. 2019). If this is the case, are these really HMO utilization genes? The authors could (a) demonstrate an inability to utilize mucin and/or (b) show a phylogenomic analysis where these infant-derived strains cluster specifically among *C. perfringens* or other human-associated clostridia to help support the assertions that these functions are clear adaptations to the infant gut – which would facilitate the claim for colonization and the other beneficial features described by the authors.
3. There is a significant focus on DSLNT as a key HMO that drives metabolic output and gene expression (e.g., Figure 1D-F) but Figure 1A shows there is essentially no growth by these organisms (or no data presented) on this sugar. DSLNT is a trace HMO, and the rationale for it as 'health promoting' (as the authors write) or being a key colonization determinant for the infant gut is not convincing. While I acknowledge that there are many studies on this HMO in the literature, the context as it is described here – as a key source of *C. perfringens* metabolites or as a colonization determinant for the strain – is not logically supported by the literature and empirical measurements of DSLNT in milk or the infant gut. The inconsistent use of DSLNT as a substrate (e.g., parts of Figure 1 but not Figure 2 or Figure S1) is also not clearly articulated. Why not show growth of the pfoA+ strain on DSLNT as well, for example? The data centered on DSLNT utilization needs to be better rationalized.

4. The authors have a clear interest in the pfoA- strain because of their hypothesis that it has the potential for competitive exclusion of other taxa (e.g., the pathobionts tested) but there is also allusions to the production of protective metabolites. However, there are comparisons between Bifidobacterium and Clostridium in terms of their production of metabolites like butyrate – which bifidobacterium are not expected to produce at all – but there is no measurements for lactate (which although not a SCFA, is relevant for the gut ecosystem, especially intestinal pH; see Brinck et al 2025). The addition of other relevant metabolites, including lactate and formate, should be included in Figure 2 and the results discussed therein.

There is also a condensed discussion of *C. perfringens* metabolites as beneficial, but this includes some compounds that are almost certainly not beneficial (e.g., cadaverine, putrescine, etc). These are enriched by *C. perfringens* (Figure 2) but they are also known to be cytotoxic compounds at physiologically relevant concentrations (e.g., see Del Rio et al 2019). The discussion of these metabolites and their impacts should be more carefully (and objectively) added to the manuscript.

Minor issues:

The description of *C. perfringens* as 'colonising' the infant gut should be supported by strain-level profiling over time – the 16S data in Figure S2 is insufficient to describe the pfoA- strain's ability to colonise (i.e., remain stably associated at consistent levels in an individual over time). Even qPCR for the pfoA- strain would be sufficient to help demonstrate that this organism is prevalent and consistent among the infant in the study (e.g., Figure S2).

The rationale for which carbon sources used throughout the paper is unclear. In some places, DSLNT is clearly of interest, in others glucose is used as a substrate (though it is rare in the gut), sometimes 'HMOs' are used without description, and then single HMOs are interrogated. The authors should (a) rationalize which carbon sources are used and (b) note which HMOs are used in which experiments clearly throughout the manuscript and figure legends.

There's a reference to a probiotic *B. infantis* strain (LM1) but no origin is presented in the methods. It stood out in Figure 1A as being the only 'probiotic' strain but it is unclear where it originated or how it may be related to other strains of the subspecies.

Decision Letter:

31st July 2025

Dear Chris,

Thank you for your patience while your manuscript "Human milk oligosaccharide metabolism by *Clostridium* species suppresses inflammation and pathogen growth" was under peer-review at Nature Microbiology. It has now been seen by 3 referees, whose comments you will find at the of this email. You will see from their comments below that while they find your work of interest, some important points are raised. We had a discussion taking your revision plan in consideration and we feel that we are very interested in the possibility of publishing your study in Nature Microbiology, but would like you to modify and resubmit a revised manuscript before we make a final decision on publication.

In particular, we recommend rewrite the manuscript to tone down all the claims around or that could be interpreted as proposing *C. perfringens* as a probiotic, instead focusing on the interesting biology that is uncovered. We also recommend adding more cohort data and improve figure S2a and the association between *C. perfringens* pfo- and protection from NEC. Besides this, please clarify rational for using DSLNT and clearly label when it is used in an experiment, add data on lactate data (which you already have), and make a broader discussion of the metabolites detected. We also suggest using public metagenome data to determine pfoA- strain prevalence to present the relevance of this work even more clearly.

If you have not done so already please begin to revise your manuscript so that it conforms to our Article format instructions at <http://www.nature.com/nmicrobiol/info/final-submission/>

The usual length limit for a Nature Microbiology Article is six display items (figures or tables) and 3,000 words. We have some flexibility, and can allow a revised manuscript at 3,500 words, but please consider this a firm upper limit. There is a trade-off of ~250 words per display item, so if you need more space, you could move a Figure or Table to Supplementary Information.

Some reduction could be achieved by focusing any introductory material and moving it to the start of your opening 'bold' paragraph, whose function is to outline the background to your work, describe in a sentence your new observations, and explain your main conclusions. The discussion should also be limited. Methods should be described in a separate section following the discussion, we do not place a word limit on Methods.

Nature Microbiology titles should give a sense of the main new findings of a manuscript, and should not contain punctuation. Please keep in mind that we strongly discourage active verbs in titles, and that they should ideally fit within 90 characters each (including spaces).

We strongly support public availability of data. Please place the data used in your paper into a public data repository, if one exists, or alternatively, present the data as Source Data or Supplementary Information. If data can only be shared on request,

please explain why in your Data Availability Statement, and also in the correspondence with your editor. For some data types, deposition in a public repository is mandatory - more information on our data deposition policies and available repositories can be found at <https://www.nature.com/nature-research/editorial-policies/reporting-standards#availability-of-data>.

Please include a data availability statement as a separate section after Methods but before references, under the heading "Data Availability". This section should inform readers about the availability of the data used to support the conclusions of your study. This information includes accession codes to public repositories (data banks for protein, DNA or RNA sequences, microarray, proteomics data etc...), references to source data published alongside the paper, unique identifiers such as URLs to data repository entries, or data set DOIs, and any other statement about data availability. At a minimum, you should include the following statement: "The data that support the findings of this study are available from the corresponding author upon request", mentioning any restrictions on availability. If DOIs are provided, we also strongly encourage including these in the Reference list (authors, title, publisher (repository name), identifier, year). For more guidance on how to write this section please see: <http://www.nature.com/authors/policies/data/data-availability-statements-data-citations.pdf>

To improve the accessibility of your paper to readers from other research areas, please pay particular attention to the wording of the paper's opening bold paragraph, which serves both as an introduction and as a brief, non-technical summary in about 150 words. If, however, you require one or two extra sentences to explain your work clearly, please include them even if the paragraph is over-length as a result. The opening paragraph should not contain references. Because scientists from other sub-disciplines will be interested in your results and their implications, it is important to explain essential but specialised terms concisely. We suggest you show your summary paragraph to colleagues in other fields to uncover any problematic concepts.

If your paper is accepted for publication, we will edit your display items electronically so they conform to our house style and will reproduce clearly in print. If necessary, we will re-size figures to fit single or double column width. If your figures contain several parts, the parts should form a neat rectangle when assembled. Choosing the right electronic format at this stage will speed up the processing of your paper and give the best possible results in print. We would like the figures to be supplied as vector files - EPS, PDF, AI or postscript (PS) file formats (not raster or bitmap files), preferably generated with vector-graphics software (Adobe Illustrator for example). Please try to ensure that all figures are non-flattened and fully editable. All images should be at least 300 dpi resolution (when figures are scaled to approximately the size that they are to be printed at) and in RGB colour format. Please do not submit Jpeg or flattened TIFF files. Please see also 'Guidelines for Electronic Submission of Figures' at the end of this letter for further detail.

Figure legends must provide a brief description of the figure and the symbols used, within 350 words, including definitions of any error bars employed in the figures.

When submitting the revised version of your manuscript, please pay close attention to our [href="https://www.nature.com/nature-research/editorial-policies/image-integrity">Digital Image Integrity Guidelines. and to the following points below:](https://www.nature.com/nature-research/editorial-policies/image-integrity)

EXTENDED DATA FIGURES

Please include a statement before the acknowledgements naming the author to whom correspondence and requests for materials should be addressed.

Finally, we require authors to include a statement of their individual contributions to the paper -- such as experimental work, project planning, data analysis, etc. -- immediately after the acknowledgements. The statement should be short, and refer to authors by their initials. For details please see the Authorship section of our joint Editorial policies at http://www.nature.com/authors/editorial_policies/authorship.html

* include a point-by-point response to any editorial suggestions and to our referees. Please include your response to the editorial suggestions in your cover letter, and please upload your response to the referees as a separate document.

* ensure it complies with our format requirements for Letters as set out in our guide to authors at www.nature.com/nmicrobiol/info/gta/

* state in a cover note the length of the text, methods and legends; the number of references; number and estimated final size of

figures and tables

* resubmit electronically if possible using the link below to access your home page:

Link Redacted

*This url links to your confidential homepage and associated information about manuscripts you may have submitted or be reviewing for us. If you wish to forward this e-mail to co-authors, please delete this link to your homepage first.

Please ensure that all correspondence is marked with your Nature Microbiology reference number in the subject line.

Nature Microbiology is committed to improving transparency in authorship. As part of our efforts in this direction, we are now requesting that all authors identified as 'corresponding author' on published papers create and link their Open Researcher and Contributor Identifier (ORCID) with their account on the Manuscript Tracking System (MTS), prior to acceptance. This applies to primary research papers only. ORCID helps the scientific community achieve unambiguous attribution of all scholarly contributions. You can create and link your ORCID from the home page of the MTS by clicking on 'Modify my Springer Nature account'. For more information please visit www.springernature.com/orcid.

We hope to receive your revised paper within three weeks. If you cannot send it within this time, please let us know.

Yours sincerely,

Reviewer Expertise:

Referee #1: Early life GI diseases

Referee #2: Early-life immunity, necrotising diseases

Referee #3: Microbiota and metabolism

Reviewers Comments:

Reviewer #1 (Remarks to the Author):

This is a thorough attempt to define the range of putatively protective effector functions of toxin perfringolysin O-deficient *C. perfringens* isolated from preterm infants. The strengths of the work include the broad range of phenotypic assays used to define the beneficial function of toxin perfringolysin O-deficient *C. perfringens*, the highly likely relevance of the assays chosen (e.g., activity against pathobionts, effects on cells in vitro), the inter-play between pfoA- *C. perfringens* and disialyllacto-N-tetraose, which has been considered to protect against a devastating inflammatory disorder of the gut in infants born preterm. The limitation of this work is that this particular species and pfoA- *C. perfringens* strain have not been as critically protective in many or most cases of gut inflammation in infancy, or in the maintenance of gut functionality.

1. Fig 1A – how extensive and systematic was the survey? The comment, “We screened the abilities of 29 bacterial isolates, mostly from preterm infant stool, to grow on six different HMOs, and glucose and lactose (Fig. 1A)” provides little guidance about how prevalent colonization with the organisms of interest is, and the systematic nature of this screening. The premise of this paper, and the justification for the experiments, are based on the organism of interest, pfoA- *C. perfringens*, being truly associated with protection from NEC, and being associated with utilization of an HMO associated with protection NEC (but not clearly proven – see

J Perinatol. 2024 Sep 19. doi: 10.1038/s41372-024-02125-9).

2. Line 93: *B. infantis* is often used, but it is not considered to be therapeutic in the context of preventing or treating any disease. Given current knowledge, its use does seem to support the growth of microbes that are not considered pathogenic, but it is not yet a therapy.

3. Line 99: The authors state “found *Clostridium* was prevalent in preterm infants, but at significantly lower rates in those diagnosed with NEC compared with time-matched 101 healthy controls ($p < 0.001$; fig S2A)” as a finding that supports their contention that *C. perfringens* could be associated with protection against NEC. This figure, however, does not lend strong support to this contention. It seems like all children born after < 32 weeks’ gestation were enrolled in this study, but that encompasses ~9 weeks’ span of gestation among the study group. The more premature a child is born, the longer it takes to develop an anaerobic predominant microbial community, and the *Clostridium*-associated protection could merely reflect the degree of prematurity. Much more data are needed to use these data in support of this contention. The pre-NEC samples in FigS2A are not well portrayed. For example, the x-axis has overlapping days of life of sampling, and the binning of patients into 1 week categories requires more elaboration. The caption states “Figure S2. Prevalences and relative abundances of *Clostridium* and *Bifidobacterium* in NEC (n= 34) and control (n= 306) infants, who did not receive probiotics. “ which would be a fairly large pre-NEC cohort. However, the insert state that these are samples, presumably from repeated samplings of a smaller number of children. Finally, labels of the X – axis add up to 32, not 34.

This concern might seem petty, except that the major appeal of this work is that *C. perfringens* protects the human infant host from inflammation of the gut that could lead to NEC but the data underlying this assertion are not strong.

Reviewer #2 (Remarks to the Author):

The authors describe novel HMO-utilizing bacteria in preterm infants beyond Bifidobacteria. *C. perfringens* produced metabolites prevented pathobiont growth and reduced inflammatory responses in preterm human organoid cultures even in absence of live bacteria. In addition, *C. perfringens* did not inhibit the growth of Bifidobacteria. The authors postulate this could provide new insights into postbiotics that could be used to support intestinal health in preterm infants.

Major points

I understand the authors resistance to an animal model to further elucidate the protective effects of the postbiotics given the complexity of the mouse or rat NEC models. However, their claims would be strengthened if they had either investigated other metabolites identified in their screen beyond butyrate, which as they mention in the discussion has already been shown to reduce NEC incidence in a rat model or provided experiments with metabolites only in preterm organoids to further promote their claim of the benefits of synthetic postbiotics, rather than CFS.

Minor points:

Two major cytokines involved in intestinal injury are missing from their panel (IL1b, IL-6), which have been implicated in NEC. Other cytokines cannot be modeled since their source is from the immune compartment.

Fig 2 write the HMO used for the growth conditions to measure the SCFA in the figure legend.

Fig.3 since the authors claim they want to use postbiotics rather than CFS from a bacteria or live bacteria the Acetate:Butyrate cocktail at pH:6 should be tested for growth inhibition of Bifido.

Fig4. Put the name of the cytokine and treatment condition in the figure for D-E so that it easy to grasp the point

S10 CCL2 data missing

S11 TNFa data does not match S9

S9, S12, S14 please place the name of the cytokine either on the y axis or the title of the graph for easier interpretation of the data

Genomes have not been deposited. Is this due to pending patent?

Growth curves should be analyzed with a more appropriate statistical method like growthcurver (PMID: 27094401) to estimate growth rate and percent inhibition quotients can be calculated to compare growth inhibition between the different treatment options not just for the pathobionts, but also for the bifido.

Reviewer #3 (Remarks to the Author):

The manuscript by Chapman et al describes the characterization of a *C. perfringens* strain lacking toxin genes and its ability to utilize human milk oligosaccharides (HMOs) in the context of preterm infant health. The manuscript is well written and clearly articulated.

There are some issues with the manuscript, both in framing and in the structural arguments presented, that should be addressed. Some minor points are also noted below.

Major issues:

1. The manuscript frames the rationale for the interest in this pfoA- *C. perfringens* as a potential probiotic in the context of preterm infants. This is provocative and intriguing, but requires more support. The authors cite some studies linking changes to the neonatal gut microbiome that are favorable for the addition of Clostridium to this community, but there is also evidence that clostridia (including *C. perfringens*) may exacerbate necrotising enterocolitis (NEC), with some evidence suggesting this may come about by the production of butyrate (see Ferraris et al 2023; Cassir et al 2019; de la Cochetière et al 2004). Others have suggested that formic acid (another fermentative product of clostridia) is responsible for NEC (see Casaburi et al 2022). Both potential causes include human and animal model data that is more extensive and more congruent than the results presented in this manuscript. I advise that the authors reframe this as an understanding of one infant gut microbe rather than drift toward the description or application of this organism as a therapeutic.

2. The description of the use of HMOs by clostridia suggests – in the summary and introduction – that this is a specific adaptation of these species or strains to the infant gut ecosystem. The alternative hypothesis may then be that these clostridia utilize mucin glycans, which bear structural similarity to HMOs, and they can repurpose these for the infant gut (e.g., see Van den Abbelle et al. 2013; Raimondi et al 2021; MacMillan et al. 2019). If this is the case, are these really HMO utilization genes? The authors could (a) demonstrate an inability to utilize mucin and/or (b) show a phylogenomic analysis where these infant-derived strains cluster specifically among *C. perfringens* or other human-associated clostridia to help support the assertions that these functions

are clear adaptations to the infant gut – which would facilitate the claim for colonization and the other beneficial features described by the authors.

3. There is a significant focus on DSLNT as a key HMO that drives metabolic output and gene expression (e.g., Figure 1D-F) but Figure 1A shows there is essentially no growth by these organisms (or no data presented) on this sugar. DSLNT is a trace HMO, and the rationale for it as 'health promoting' (as the authors write) or being a key colonization determinant for the infant gut is not convincing. While I acknowledge that there are many studies on this HMO in the literature, the context as it is described here – as a key source of *C. perfringens* metabolites or as a colonization determinant for the strain – is not logically supported by the literature and empirical measurements of DSLNT in milk or the infant gut. The inconsistent use of DSLNT as a substrate (e.g., parts of Figure 1 but not Figure 2 or Figure S1) is also not clearly articulated. Why not show growth of the pfoA+ strain on DSLNT as well, for example? The data centered on DSLNT utilization needs to be better rationalized.

4. The authors have a clear interest in the pfoA- strain because of their hypothesis that it has the potential for competitive exclusion of other taxa (e.g., the pathobionts tested) but there is also allusions to the production of protective metabolites. However, there are comparisons between *Bifidobacterium* and *Clostridium* in terms of their production of metabolites like butyrate – which *bifidobacterium* are not expected to produce at all – but there is no measurements for lactate (which although not a SCFA, is relevant for the gut ecosystem, especially intestinal pH; see Brinck et al 2025). The addition of other relevant metabolites, including lactate and formate, should be included in Figure 2 and the results discussed therein.

There is also a condensed discussion of *C. perfringens* metabolites as beneficial, but this includes some compounds that are almost certainly not beneficial (e.g., cadaverine, putrescine, etc). These are enriched by *C. perfringens* (Figure 2) but they are also known to be cytotoxic compounds at physiologically relevant concentrations (e.g., see Del Rio et al 2019). The discussion of these metabolites and their impacts should be more carefully (and objectively) added to the manuscript.

Minor issues:

The description of *C. perfringens* as 'colonising' the infant gut should be supported by strain-level profiling over time – the 16S data in Figure S2 is insufficient to describe the pfoA- strain's ability to colonise (i.e., remain stably associated at consistent levels in an individual over time). Even qPCR for the pfoA- strain would be sufficient to help demonstrate that this organism is prevalent and consistent among the infant in the study (e.g., Figure S2).

The rationale for which carbon sources used throughout the paper is unclear. In some places, DSLNT is clearly of interest, in others glucose is used as a substrate (though it is rare in the gut), sometimes 'HMOs' are used without description, and then single HMOs are interrogated. The authors should (a) rationalize which carbon sources are used and (b) note which HMOs are used in which experiments clearly throughout the manuscript and figure legends.

There's a reference to a probiotic *B. infantis* strain (LM1) but no origin is presented in the methods. It stood out in Figure 1A as being the only 'probiotic' strain but it is unclear where it originated or how it may be related to other strains of the subspecies.

Version 1:

Reviewer comments:

Reviewer #1

(Remarks to the Author)

I limit my response to the authors' rebuttal to my comments.

Response to my general comment. The authors took issue with my mention of weaknesses in the first manuscript because I did not provide specifics about my concerns that pfoA-*C. perfringens* strains have not been critically shown to be protective against disease, or improvers of gut functionality. Elaboration was provided in comment 1 in the original review. Therefore, their statement "The limitation raised by the reviewer does not seem to refer to any specific work undertaken as part of the manuscript" does not prompt me to reconsider this concern.

Specific concern 1 (2 parts in their rebuttal). The authors respond to my request to provide more information about how they landed on 29 isolates for this study.

They state that they

"... focused on the most abundant colonizers of preterm infants. This is stated in the main text (first Results paragraph) that these represent a median of 80% (interquartile range 61%-91%) of all relative microbial abundance observed in preterm infants. Thus, we feel this to be an extensive survey of bacteria that colonize preterm infants."

The first paragraph Results section in the revision states:

"We screened the abilities of 29 bacterial isolates, mostly from preterm infant stool, to grow on six different HMOs, and glucose and lactose (Fig. 1A). These species were obtained by untargeted cultivation and represent a median of 80% (interquartile range 61%-91%) of all relative microbial abundance observed in 85 preterm infants. (REF 21)"

Reference 21 is an analysis of 1,431 samples collected over time from 123 very preterm infants born <32 weeks' in a NICU in the UK, published in Nature Microbiology. The infants received 2 combination probiotics or maternal milk, and in that paper, data were presented showing that the probiotics influenced the kinetics of acquisition of members of the gut microbiota. In reviewing this paper, I do not find information in support of this statistic about relative abundance. Though I have no reason to doubt its validity, it would be nice to see substantiation of its use to select the bacteria described in the submitted manuscript.

Assuming that these taxa are appropriate to select, I still do not see how they end up with the 29 isolates. In re-reading the manuscript, I cannot find how many infants were screened, and on what days of life the stools of origin of these bacteria were collected. Perhaps I missed this information, and such data do not directly affect their in vitro findings, but the presence of cohort and specimen information would strengthen the value of this paper.

The authors also missed the point of my concern about the data underlying the justification for studying pfoA-C. perfringens (second section of itemized response 1). My point did not relate to data on mechanistic support, instead, the authors should qualify that HMOs and DSLNT are not the complete story in NEC susceptibility, based on the paper cited.

The authors also state "... we do not claim that the strain AM1 is associated with protection against NEC, rather, based on our multi-faceted data, we state its broad potential "as a live biotherapy or postbiotic for preterm infants at risk of NEC and likely other related pathologies". I address this below.

Specific concern 2. – concern resolved

Specific concern 3. I still believe that this paper infers that *C. perfringens* in general, and pfoA-C. perfringens in particular, offer protection against gut pathology and functionality in preterm infants and against NEC. Such postulated protection is important to the relevance of this work. The authors respond to my concern that the original manuscript made inferences that *C. perfringens* in general, and pfoA-C. perfringens in particular, offer protective against gut pathology and functionality in preterm infants and against NEC:

"We would like to reiterate that we were deliberate throughout the manuscript to avoid claims that *C. perfringens* is protective against NEC." They also state (in response to concern 1): "... we do not claim that the strain AM1 is associated with protection against NEC, rather, based on our multi-faceted data, we state its broad potential "as a live biotherapy or postbiotic for preterm infants at risk of NEC and likely other related pathologies.""

However, in the current manuscript, they continue to imply the disease relevance of their findings:

Line 96: "The HMO disialyllacto-N-tetraose (DSLNT) has been consistently associated with protection from NEC, (REFS 5,7), and *C. perfringens* strain AM1 reached the highest optical density on this HMO (Fig. 1A)."

Line 320-321: "... but only a pfoA- strain could use DSLNT, a HMO that is linked to reduced NEC risk. (REFS 5-7)."

Furthermore, in the pending patent application, the authors refer to what are data in the paper and state:

"The inventors have also surprisingly shown that *Clostridium perfringens* is an abundant and prevalent species present in the gut in early life and is present at higher levels in healthy infants compared to infants diagnosed with necrotising enterocolitis (NEC) during the critical window of NEC risk (the first 10 to 40 days of life). Furthermore, the inventors have demonstrated that *Clostridium perfringens* is more abundant in the gut of healthy infants compared to infants before they were diagnosed with NEC, but not after recovery from NEC.

This suggests that *Clostridium perfringens* plays an important role in protecting against NEC and suggests that increased levels of *Clostridium perfringens* in the gut would be beneficial and protective against disease."

Reviewer #2

(Remarks to the Author)

The authors addressed the majority of the concerns raised in the first round of reviews. I still remain confused about figure 2A. Since the strains can grow on more than one HMO was data obtained from growth on different individual HMOs combined in the same plot for a given strain? Are these triplicates of strains grown in the same HMO? Please clarify whether you chose one specific HMO when the strains were able to utilize more than one. In all other experiments a mix of HMOs specific to the strain was used, which is easier to interpret.

The other are just minor formatting or spelling issues.

Figure 1 legend (D) RNA-seq instead of RNA-srq (F) DSNLT instead of DSNT. Only include abbreviations used in the figure legend at the end.

Given the overlap between the upregulated metabolites identified across species a venn diagram would help with interpretation of the data.

Extended figure 2 swap C D panels around so that the proportion and presence of the same bacteria are under each other

Extended figure 3 there are abbreviations included that are not used in the figure legend.

Extended Figure 4 is this data from a single outgrowth or averaged over multiple ones? If average please provide more detail.

Extended Figure 8: "Statistical comparisons were performed using average genus values." Does that mean that all the clostridia were grouped together when comparing? If yes it is misleading as the legend states that "Conditions with the same letters are not significantly different." When those conditions were never tested but grouped together

Data:

Since the authors mention they will deposit the genome sequences please provide identifiers for SRA.

Text spelling:

Results

CAZyme instead of cazyme

Extended Data Fig. 3D and S3E. There is no figure S3E.

Methods

and relative abundance

Reviewer #4

(Remarks to the Author)

Overall, the authors have sufficiently addressed or rebutted the concerns of the original Reviewer 3, and the manuscript is improved as a result. A few further minor points of clarification are remain:

R3, Comment 3: I agree with the authors that growth of *C. perfringens* pfoA- on DSLNT is interesting and worthy of highlighting. However, I agree with the original reviewer that it was strange to not have included DSLNT in Figure S1. If the rationale for excluding it from Fig S1 was that none of those strains grew on DSLNT, then this should be described in the text.

R3, Comment 5: The authors have added the caveat that these polyamines are cytotoxic at "higher concentrations". How do the concentrations identified as toxic in the DelRio paper compare to those in the CFS? Would they consider these to be biologically relevant?

R3, Comment 7: Further clarification is required. "We therefore quantified SCFAs within culture supernatants of strains grown on the individual HMOs we found they could utilise (Fig. 1A)..."  This wording suggests one HMO per culture, whereas the following suggests a mix of different HMOs in each culture  "Isolates were grown on strain-specific mixtures of the HMOs we found they can utilise (Fig. 1A)." Given that the results section occurs before the methods section in Nature publications, the description of these experiments in the results must be clear to readers. Please also clarify in the methods whether the HMOs were mixed at equimolar ratios.

Decision Letter:

2nd December 2025

Dear Chris,

Thank you for your patience while your manuscript "Human milk oligosaccharide metabolism by *Clostridium* species suppresses inflammation and pathogen growth" was under peer-review at Nature Microbiology. It has now been seen by 3 referees, whose comments you will find at the of this email.

It took us a while to discuss it internally considering the comments by R1 with respect to the patent. However, after talking to other editorial teams, we think we can make an informed decision now. We are very interested in the possibility of publishing your study in Nature Microbiology, but would like to consider your response to these concerns in the form of a revised manuscript before we make a final decision on publication.

The only changes required to the manuscript are now textual - acknowledgement of limitations, some clarifications, addition of some details, and some improvement of figures and discussion. Once it is revised, it is unlikely that we will send it back to the referees.

If you have not done so already please begin to revise your manuscript so that it conforms to our Article format instructions at <http://www.nature.com/nmicrobiol/info/final-submission/>

The usual length limit for a Nature Microbiology Article is six display items (figures or tables) and 3,000 words. We have some flexibility, and can allow a revised manuscript at 3,500 words, but please consider this a firm upper limit. There is a trade-off of ~250 words per display item, so if you need more space, you could move a Figure or Table to Supplementary Information.

Some reduction could be achieved by focusing any introductory material and moving it to the start of your opening 'bold' paragraph, whose function is to outline the background to your work, describe in a sentence your new observations, and explain your main conclusions. The discussion should also be limited. Methods should be described in a separate section following the discussion, we do not place a word limit on Methods.

Nature Microbiology titles should give a sense of the main new findings of a manuscript, and should not contain punctuation. Please keep in mind that we strongly discourage active verbs in titles, and that they should ideally fit within 90 characters each (including spaces).

Please include a data availability statement as a separate section after Methods but before references, under the heading "Data Availability". This section should inform readers about the availability of the data used to support the conclusions of your study. This information includes accession codes to public repositories (data banks for protein, DNA or RNA sequences, microarray, proteomics data etc...), references to source data published alongside the paper, unique identifiers such as URLs to data repository entries, or data set DOIs, and any other statement about data availability. At a minimum, you should include the following statement: "The data that support the findings of this study are available from the corresponding author upon request", mentioning any restrictions on availability. If DOIs are provided, we also strongly encourage including these in the Reference list (authors, title, publisher (repository name), identifier, year). For more guidance on how to write this section please see: <http://www.nature.com/authors/policies/data/data-availability-statements-data-citations.pdf>

To improve the accessibility of your paper to readers from other research areas, please pay particular attention to the wording of the paper's opening bold paragraph, which serves both as an introduction and as a brief, non-technical summary in about 150 words. If, however, you require one or two extra sentences to explain your work clearly, please include them even if the paragraph is over-length as a result. The opening paragraph should not contain references. Because scientists from other sub-disciplines will be interested in your results and their implications, it is important to explain essential but specialised terms concisely. We suggest you show your summary paragraph to colleagues in other fields to uncover any problematic concepts.

If your paper is accepted for publication, we will edit your display items electronically so they conform to our house style and will reproduce clearly in print. If necessary, we will re-size figures to fit single or double column width. If your figures contain several parts, the parts should form a neat rectangle when assembled. Choosing the right electronic format at this stage will speed up the processing of your paper and give the best possible results in print. We would like the figures to be supplied as vector files - EPS, PDF, AI or postscript (PS) file formats (not raster or bitmap files), preferably generated with vector-graphics software (Adobe Illustrator for example). Please try to ensure that all figures are non-flattened and fully editable. All images should be at least 300 dpi resolution (when figures are scaled to approximately the size that they are to be printed at) and in RGB colour format. Please do not submit Jpeg or flattened TIFF files. Please see also 'Guidelines for Electronic Submission of Figures' at the end of this letter for further detail.

Figure legends must provide a brief description of the figure and the symbols used, within 350 words, including definitions of any error bars employed in the figures.

When submitting the revised version of your manuscript, please pay close attention to our [href="https://www.nature.com/nature-research/editorial-policies/image-integrity">Digital Image Integrity Guidelines. and to the following points below:](https://www.nature.com/nature-research/editorial-policies/image-integrity)

EXTENDED DATA FIGURES

Please include a statement before the acknowledgements naming the author to whom correspondence and requests for materials should be addressed.

Finally, we require authors to include a statement of their individual contributions to the paper -- such as experimental work, project planning, data analysis, etc. -- immediately after the acknowledgements. The statement should be short, and refer to authors by their initials. For details please see the Authorship section of our joint Editorial policies at http://www.nature.com/authors/editorial_policies/authorship.html

* include a point-by-point response to any editorial suggestions and to our referees. Please include your response to the editorial suggestions in your cover letter, and please upload your response to the referees as a separate document.

* ensure it complies with our format requirements for Letters as set out in our guide to authors at www.nature.com/nmicrobiol/info/gta/

* state in a cover note the length of the text, methods and legends; the number of references; number and estimated final size of figures and tables

* resubmit electronically if possible using the link below to access your home page:

Link Redacted

*This url links to your confidential homepage and associated information about manuscripts you may have submitted or be reviewing for us. If you wish to forward this e-mail to co-authors, please delete this link to your homepage first.

Please ensure that all correspondence is marked with your Nature Microbiology reference number in the subject line.

Nature Microbiology is committed to improving transparency in authorship. As part of our efforts in this direction, we are now requesting that all authors identified as 'corresponding author' on published papers create and link their Open Researcher and Contributor Identifier (ORCID) with their account on the Manuscript Tracking System (MTS), prior to acceptance. This applies to primary research papers only. ORCID helps the scientific community achieve unambiguous attribution of all scholarly contributions. You can create and link your ORCID from the home page of the MTS by clicking on 'Modify my Springer Nature account'. For more information please visit www.springernature.com/orcid.

We hope to receive your revised paper within three weeks. If you cannot send it within this time, please let us know.

Yours sincerely,

Reviewers Comments:

Reviewer #1 (Remarks to the Author):

I limit my response to the authors' rebuttal to my comments.

Response to my general comment. The authors took issue with my mention of weaknesses in the first manuscript because I did not provide specifics about my concerns that pfoA-C. perfringens strains have not been critically shown to be protective against disease, or improvers of gut functionality. Elaboration was provided in comment 1 in the original review. Therefore, their statement "The limitation raised by the reviewer does not seem to refer to any specific work undertaken as part of the manuscript" does not prompt me to reconsider this concern.

Specific concern 1 (2 parts in their rebuttal). The authors respond to my request to provide more information about how they landed on 29 isolates for this study.

They state that they

"... focused on the most abundant colonizers of preterm infants. This is stated in the main text (first Results paragraph) that these represent a median of 80% (interquartile range 61%-91%) of all relative microbial abundance observed in preterm infants. Thus, we feel this to be an extensive survey of bacteria that colonize preterm infants."

The first paragraph Results section in the revision states:

"We screened the abilities of 29 bacterial isolates, mostly from preterm infant stool, to grow on six different HMOs, and glucose and lactose (Fig. 1A). These species were obtained by untargeted cultivation and represent a median of 80% (interquartile range 61%-91%) of all relative microbial abundance observed in 85 preterm infants. (REF 21)"

Reference 21 is an analysis of 1,431 samples collected over time from 123 very preterm infants born <32 weeks' in a NICU in the UK, published in Nature Microbiology. The infants received 2 combination probiotics or maternal milk, and in that paper, data were presented showing that the probiotics influenced the kinetics of acquisition of members of the gut microbiota. In reviewing this paper, I do not find information in support of this statistic about relative abundance. Though I have no reason to doubt its validity, it would be nice to see substantiation of its use to select the bacteria described in the submitted manuscript.

Assuming that these taxa are appropriate to select, I still do not see how they end up with the 29 isolates. In re-reading the manuscript, I cannot find how many infants were screened, and on what days of life the stools of origin of these bacteria were collected. Perhaps I missed this information, and such data do not directly affect their in vitro findings, but the presence of cohort

and specimen information would strengthen the value of this paper.

The authors also missed the point of my concern about the data underlying the justification for studying pfoA-C. perfringens (second section of itemized response 1). My point did not relate to data on mechanistic support, instead, the authors should qualify that HMOs and DSLNT are not the complete story in NEC susceptibility, based on the paper cited.

The authors also state "... we do not claim that the strain AM1 is associated with protection against NEC, rather, based on our multi-faceted data, we state its broad potential "as a live biotherapy or postbiotic for preterm infants at risk of NEC and likely other related pathologies". I address this below.

Specific concern 2. – concern resolved

Specific concern 3. I still believe that this paper infers that *C. perfringens* in general, and pfoA-C. perfringens in particular, offer protection against gut pathology and functionality in preterm infants and against NEC. Such postulated protection is important to the relevance of this work. The authors respond to my concern that the original manuscript made inferences that *C. perfringens* in general, and pfoA-C. perfringens in particular, offer protective against gut pathology and functionality in preterm infants and against NEC:

"We would like to reiterate that we were deliberate throughout the manuscript to avoid claims that *C. perfringens* is protective against NEC." They also state (in response to concern 1): "... we do not claim that the strain AM1 is associated with protection against NEC, rather, based on our multi-faceted data, we state its broad potential "as a live biotherapy or postbiotic for preterm infants at risk of NEC and likely other related pathologies.""

However, in the current manuscript, they continue to imply the disease relevance of their findings:

Line 96: "The HMO disialyllacto-N-tetraose (DSLNT) has been consistently associated with protection from NEC, (REFS 5,7), and *C. perfringens* strain AM1 reached the highest optical density on this HMO (Fig. 1A)."

Line 320-321: "... but only a pfoA- strain could use DSLNT, a HMO that is linked to reduced NEC risk. (REFS 5-7)."

Furthermore, in the pending patent application, the authors refer to what are data in the paper and state:

"The inventors have also surprisingly shown that *Clostridium perfringens* is an abundant and prevalent species present in the gut in early life and is present at higher levels in healthy infants compared to infants diagnosed with necrotising enterocolitis (NEC) during the critical window of NEC risk (the first 10 to 40 days of life). Furthermore, the inventors have demonstrated that *Clostridium perfringens* is more abundant in the gut of healthy infants compared to infants before they were diagnosed with NEC, but not after recovery from NEC.

This suggests that *Clostridium perfringens* plays an important role in protecting against NEC and suggests that increased levels of *Clostridium perfringens* in the gut would be beneficial and protective against disease."

Reviewer #2 (Remarks to the Author):

The authors addressed the majority of the concerns raised in the first round of reviews. I still remain confused about figure 2A. Since the strains can grow on more than one HMO was data obtained from growth on different individual HMOs combined in the same plot for a given strain? Are these triplicates of strains grown in the same HMO? Please clarify whether you chose one specific HMO when the strains were able to utilize more than one. In all other experiments a mix of HMOs specific to the strain was used, which is easier to interpret.

The other are just minor formatting or spelling issues.

Figure 1 legend (D) RNA-seq instead of RNA-srq (F) DSLNT instead of DSNT. Only include abbreviations used in the figure legend at the end.

Given the overlap between the upregulated metabolites identified across species a venn diagram would help with interpretation of the data.

Extended figure 2 swap C D panels around so that the proportion and presence of the same bacteria are under each other

Extended figure 3 there are abbreviations included that are not used in the figure legend.

Extended Figure 4 is this data from a single outgrowth or averaged over multiple ones? If average please provide more detail.

Extended Figure 8: "Statistical comparisons were performed using average genus values." Does that mean that all the clostridia were grouped together when comparing? If yes it is misleading as the legend states that "Conditions with the same letters are not significantly different." When those conditions were never tested but grouped together

Data:

Since the authors mention they will deposit the genome sequences please provide identifiers for SRA.

Text spelling:

Results

CAZyme instead of cazyme

Extended Data Fig. 3D and S3E. There is no figure S3E.

Methods

and relative abundance

Reviewer #4 (Remarks to the Author):

Overall, the authors have sufficiently addressed or rebutted the concerns of the original Reviewer 3, and the manuscript is improved as a result. A few further minor points of clarification are remain:

R3, Comment 3: I agree with the authors that growth of *C. perfringens* pfoA- on DSLNT is interesting and worthy of highlighting. However, I agree with the original reviewer that it was strange to not have included DSLNT in Figure S1. If the rationale for excluding it from Fig S1 was that none of those strains grew on DSLNT, then this should be described in the text.

R3, Comment 5: The authors have added the caveat that these polyamines are cytotoxic at "higher concentrations". How do the concentrations identified as toxic in the DelRio paper compare to those in the CFS? Would they consider these to be biologically relevant?

R3, Comment 7: Further clarification is required. "We therefore quantified SCFAs within culture supernatants of strains grown on the individual HMOs we found they could utilise (Fig. 1A)..."  This wording suggests one HMO per culture, whereas the following suggests a mix of different HMOs in each culture  "Isolates were grown on strain-specific mixtures of the HMOs we found they can utilise (Fig. 1A)." Given that the results section occurs before the methods section in Nature publications, the description of these experiments in the results must be clear to readers. Please also clarify in the methods whether the HMOs were mixed at equimolar ratios.

Version 2:

Decision Letter:

Our ref: NMICROBIOL-25051794B

22nd December 2025

Dear Chris,

Thank you for your patience while your manuscript "Human milk oligosaccharide metabolism by *Clostridium* species suppresses inflammation and pathogen growth" was under peer review at Nature Microbiology. Please accept my apologies for the time it took, but the end of the year is a really busy time and we wanted to make sure that we could assess the latest version of the manuscript without referee support.

We feel that the concerns raised in the previous rounds of revision have been addressed appropriately and, therefore, we are happy to offer to publish it in principle in Nature Microbiology, pending minor revisions to comply with our editorial and formatting guidelines.

Thank you again for your interest in Nature Microbiology Please do not hesitate to contact me if you have any questions.

I hope you have a relaxing holiday.

Season's Greetings,

Version 3:

Decision Letter:

17th February 2026

Dear Chris,

I am pleased to accept your Article "Clostridia from preterm infants metabolise human milk oligosaccharides to suppress pathobionts and modulate intestinal function in organoids" for publication in Nature Microbiology. Thank you for having chosen to submit your work to us and many congratulations.

Authors may need to take specific actions to achieve compliance with funder and institutional open access mandates. If your research is supported by a funder that requires immediate open access (e.g. according to <https://www.springernature.com/gp/open-science/plan-s-compliance> Plan S principles or the <https://www.springernature.com/gp/open-science/us-federal-agency-compliance> NIH public access policy) then you should select the gold OA route, and we will direct you to the compliant route where possible. Because authors warrant under our subscription licensing terms that they haven't committed to licensing any version of their article under a licence inconsistent with the terms of our agreement – including the applicable embargo period – publication under the subscription model isn't suitable for authors whose funders require no embargo.

With kind regards,

P.S. Click on the following link if you would like to recommend Nature Microbiology to your librarian
<http://www.nature.com/subscriptions/recommend.html#forms>

** Visit the Springer Nature Editorial and Publishing website at http://editorial-jobs.springernature.com?utm_source=ejP_NMicro_email&utm_medium=ejP_NMicro_email&utm_campaign=ejp_NMicro for more information about our career opportunities. If you have any questions please click [here](mailto:editorial.publishing.jobs@springernature.com).**

We wish to thank the editors for the constructive correspondence over this manuscript, as well as the reviewers for their positive feedback and helpful suggestions to improve our manuscript. We provide point-by-point responses to the editorial and reviewer comments below.

R1: Early life microbiota, GI infections

This is a thorough attempt to define the range of putatively protective effector functions of toxin perfringolysin O-deficient *C. perfringens* isolated from preterm infants. The strengths of the work include the broad range of phenotypic assays used to define the beneficial function of toxin perfringolysin O-deficient *C. perfringens*, the highly likely relevance of the assays chosen (e.g., activity against pathobionts, effects on cells *in vitro*), the inter-play between pfoA- *C. perfringens* and disialyllacto-N-tetraose, which has been considered to protect against a devastating inflammatory disorder of the gut in infants born preterm. The limitation of this work is that this particular species and pfoA- *C. perfringens* strain have not been as critically protective in many or most cases of gut inflammation in infancy, or in the maintenance of gut functionality.

We thank the reviewer for noting the several strengths of the study. The limitation raised by the reviewer does not seem to refer to any specific work undertaken as part of the manuscript. If they are referring to the novelty of our data, this reflects the strain-level approaches we used, which are critical for determining pfoA status and would be missed in almost all previous studies.

1. Fig 1A – how extensive and systematic was the survey? The comment, “We screened the abilities of 29 bacterial isolates, mostly from preterm infant stool, to grow on six different HMOs, and glucose and lactose (Fig. 1A)” provides little guidance about how prevalent colonization with the organisms of interest is, and the systematic nature of this screening.

We focused on the most abundant colonisers of preterm infants. This is stated in the main text (first Results paragraph) that these represent a median of 80% (interquartile range 61%-91%) of all relative microbial abundance observed in preterm infants. Thus, we feel this to be an extensive survey of bacteria that colonise preterm infants.

The premise of this paper, and the justification for the experiments, are based on the organism of interest, pfoA- *C. perfringens*, being truly associated with protection from NEC, and being associated with utilization of an HMO associated with protection NEC (but not clearly proven – see *J Perinatol.* 2024 Sep 19. doi: 10.1038/s41372-024-02125-9).

The reviewer is correct that the ability of strain AM1 to utilise the HMO DSLNT was used as *one* criterion (of many) to select this specific strain for further experiments, owing to the widespread research interest in DSLNT and the result that so few of the bacteria we tested could utilise it (see Fig. 1A). We gather from the above statement that the reviewer is suggesting that the HMO DSLNT has not been mechanistically proven to provide protection against NEC, for which we agree. For this reason, we do not focus our claims only on DSLNT in this manuscript, but rather on the general proposition that bacteria capable of metabolising any HMOs may provide a beneficial effect to the host. Furthermore, we do not claim that the strain AM1 is associated with protection against NEC, rather, based on our multi-faceted data, we state its broad potential “as a live biotherapy or postbiotic for preterm infants at risk of NEC and likely other related pathologies”.

2. Line 93: *B. infantis* is often used, but it is not considered to be therapeutic in the context of preventing or treating any disease. Given current knowledge, its use does seem to support the growth of microbes that are not considered pathogenic, but it is not yet a therapy.

We agree and have reworded “therapeutic strain” to “this clinically used strain”.

3. Line 99: The authors state “found *Clostridium* was prevalent in preterm infants, but at significantly lower rates in those diagnosed with NEC compared with time-matched 101 healthy controls ($p < 0.001$; fig S2A)” as a finding that supports their contention that *C. perfringens* could be associated with protection against NEC. This figure, however, does not lend strong support to this contention. It seems like all children born after < 32 weeks’

gestation were enrolled in this study, but that encompasses ~9 weeks' span of gestation among the study group. The more premature a child is born, the longer it takes to develop an anaerobic predominant microbial community, and the *Clostridium*-associated protection could merely reflect the degree of prematurity. Much more data are needed to use these data in support of this contention. The pre-NEC samples in FigS2A are not well portrayed. For example, the x-axis has overlapping days of life of sampling, and the binning of patients into 1 week categories requires more elaboration. The caption states "Figure S2. Prevalences and relative abundances of *Clostridium* and *Bifidobacterium* in NEC (n= 34) and control (n= 306) infants, who did not receive probiotics." which would be a fairly large pre-NEC cohort. However, the insert state that these are samples, presumably from repeated samplings of a smaller number of children. Finally, labels of the X – axis add up to 32, not 34. This concern might seem petty, except that the major appeal of this work is that *C. perfringens* protects the human infant host from inflammation of the gut that could lead to NEC but the data underlying this assertion are not strong.

We would like to reiterate that we were deliberate throughout the manuscript to avoid claims that *C. perfringens* is protective against NEC. This figure shows an association from published data, representing an additional layer of evidence that *Clostridium* prevalence is higher in healthy controls. This was explored only after discovering almost all tested *Clostridium* spp. could metabolise HMOs and so does not "underlie" the assertions made in this study.

We could not evidence of more prematurity slowing development of an anaerobic predominant community. Indeed, as microbial colonisation begins at birth and gut microbiome development is generally comparable across gestational ages of 23-32 weeks (i.e., those hospitalised in neonatal intensive care), we do not expect the earlier gestations to have any meaningful impacts on the anaerobic predominant microbial community. Importantly, the gestational ages of NEC and controls in this analysis was comparable (**Review Table 1**; P = 0.10) and we have now included this information in Figure S2 legend. Finally on this point, as *Bifidobacterium* are also obligate anaerobes like *Clostridium*, you would then expect the difference in their prevalence in NEC vs control to mirror the trends seen with *Clostridium* if this effect was purely due to the NEC infants being more premature and 'having lower anaerobe colonisation', but this was not the case in this analysis (Fig. S2B).

Review Table 1. Gestational Age of NEC and control infants (P = 0.10)

	Median	Q1	Q3
Ctrl	28	26	29
NEC	27	26	28

We apologise for the confusion over whether "n=" refers to patients or infants. We have now added the patient numbers to the figure legend and removed all information about sample numbers from the plots themselves, instead listing this information within the figure legend. We appreciate the reviewer spotting a couple of minor typos that we have corrected (e.g., overlapping days and the number of NEC samples not adding up to 34 on the X-axis).

R2: Necrotising diseases, early life immunity

The authors describe novel HMO-utilizing bacteria in preterm infants beyond *Bifidobacteria*. *C. perfringens* produced metabolites prevented pathobiont growth and reduced inflammatory responses in preterm human organoid cultures even in absence of live bacteria. In addition, *C. perfringens* did not inhibit the growth of *Bifidobacteria*. The authors postulate this could provide new insights into postbiotics that could be used to support intestinal health in preterm infants.

Major points

I understand the authors resistance to an animal model to further elucidate the protective effects of the postbiotics given the complexity of the mouse or rat NEC models. However, their claims would be strengthened if they had either investigated other metabolites identified in their screen beyond butyrate, which as they mention in the discussion has already been shown to reduce NEC incidence in a rat model or provided

experiments with metabolites only in preterm organoids to further promote their claim of the benefits of synthetic postbiotics, rather than CFS.

Regarding the suggestion to perform further experiments to disentangle the “... benefits of synthetic postbiotics, rather than CFS”, we wish to clarify that ‘postbiotics’ and ‘CFS’ (cell-free supernatant) are the same, but because “postbiotics” have a specific definition that involves improving health, we deliberately avoided the term and opted for CFS throughout when referring to our data. So to confirm, we do not make any claims about “synthetic postbiotics” in the text and our implication is that the CFS would be a postbiotic, not that we would formulate one synthetically.

We recognise the value in screening specific beneficial metabolites identified in the metabolomics data, but note this would be a vast undertaking requiring extensive fractionation and purification processes that are time and cost intensive, and thus outside the scope of the current project. For example, we found 44 metabolites of interest for strain AM1 (pfoA negative *C. perfringens*) that would require testing alone and in combinations. This also ignores that other metabolites might have combinatorial effects that would be missed. While we are discussing with industry partners and funders about performing this next step, this is several years away. Ultimately, we are careful in the manuscript to never attribute all of the observed effects of CFSs and *Clostridium* to any single metabolite, but instead postulate that “...*Clostridium* species use HMOs in the preterm gut, producing a broad range of SCFAs, tryptophan catabolites, and other potentially beneficial immunomodulatory metabolites that positively influence host physiology”. Thus, having avoided any claim to a specific metabolite or small subset of metabolites, we feel such systematic work represents an entirely new study and does not diminish our robust results as presented.

Minor points:

Two major cytokines involved in intestinal injury are missing from their panel (IL1b, IL-6), which have been implicated in NEC. Other cytokines cannot be modeled since their source is from the immune compartment. In optimisation trials within the lab, we have attempted to measure IL1b and IL6, but have found that, under the conditions we use, our organoids do not produce detectable quantities of these cytokines. We thus removed them from the custom panel in favour of more reliably detectable cytokines.

Fig 2 write the HMO used for the growth conditions to measure the SCFA in the figure legend.

We have stated in the methods that SCFA profiling was conducted on supernatants collected following the HMO growth assay shown in Fig 1A. So the HMOs used were those on which the named bacteria showed growth. We have updated the legend of Fig 2 to clarify this:

“Box plots showing individual short chain fatty acid (SCFA) production by species grown on individual HMOs. SCFA profiling was performed on culture supernatants harvested following the HMO growth assay shown in Fig 1A. Per species, the HMOs used were those on which growth was seen (as indicated in Fig 1A).”

Fig.3 since the authors claim they want to use postbiotics rather than CFS from a bacteria or live bacteria the Acetate:Butyrate cocktail at pH:6 should be tested for growth inhibition of Bifido.

To clarify, as noted above, our assertion is that the CFS would be the postbiotic and a specific acetate:butyrate mixture would not be a postbiotic. Thus, we do not suggest that the acetate:butyrate mixture would be a therapeutic formulation. This mixture was used only as a positive control for the known antibacterial activities of SCFAs, as we have shown the *Clostridium* to produce high quantities of these metabolites. Furthermore, the data in Fig. 3D & 3E shows the *Clostridium* CFS does not inhibit two of the *Bifidobacterium* strains and only inhibits one by 48 hrs. Thus, we do not observe strong antibacterial effects by the CFS for *Bifidobacterium* that warrant further investigation in the context of the current study.

Fig4. Put the name of the cytokine and treatment condition in the figure for D-E so that it easy to grasp the point Figure 4D and E have been amended accordingly. The cytokine is now named at the top of each, colour keys have been added to make clear which treatments have been plotted and the x axes labels have been amended to state against which other treatment the fold change comparisons have been made.

S10 CCL2 data missing

This was a formatting oversight on our part, and we have now added the correct image showing the CCL2 data.

S11 TNFa data does not match S9

S11 shows data from a different organoid experiment, wherein the AM1 CFS was added at different pHs to the organoids.

S9, S12, S14 please place the name of the cytokine either on the y axis or the title of the graph for easier interpretation of the data

We have added the cytokine names to the plots as requested.

Genomes have not been deposited. Is this due to pending patent?

This was originally due to exploring the patent but we would be glad to deposit these genomes and have them released publicly at the point of publication.

Growth curves should be analyzed with a more appropriate statistical method like growthcurver (PMID: 27094401) to estimate growth rate and percent inhibition quotients can be calculated to compare growth inhibition between the different treatment options not just for the pathobionts, but also for the bifido.

We thank the reviewer for suggesting this tool. Notably, this package is the same approach to that used in the current work, specifically it “computes an empirical AUC by summing the areas of the trapezoids made up by connecting consecutive data points”, and this AUC trapezoid method is what we used when calculating AUCs using the trapz package, demonstrating our current analysis is appropriate.

R3: Microbiota and metabolism

The manuscript by Chapman et al describes the characterization of a *C. perfringens* strain lacking toxin genes and its ability to utilize human milk oligosaccharides (HMOs) in the context of preterm infant health. The manuscript is well written and clearly articulated.

There are some issues with the manuscript, both in framing and in the structural arguments presented, that should be addressed. Some minor points are also noted below.

Major issues:

1. The manuscript frames the rationale for the interest in this pfoA- *C. perfringens* as a potential probiotic in the context of preterm infants. This is provocative and intriguing, but requires more support. The authors cite some studies linking changes to the neonatal gut microbiome that are favorable for the addition of *Clostridium* to this community, but there is also evidence that clostridia (including *C. perfringens*) may exacerbate necrotising enterocolitis (NEC), with some evidence suggesting this may come about by the production of butyrate (see Ferraris et al 2023; Cassir et al 2019; de la Cochetière et al 2004). Others have suggested that formic acid (another fermentative product of clostridia) is responsible for NEC (see Casaburi et al 2022). Both potential causes include human and animal model data that is more extensive and more congruent than the results presented in this manuscript. I advise that the authors reframe this as an understanding of one infant gut microbe rather than drift toward the description or application of this organism as a therapeutic.

We respectively disagree that “[previous work on *C. perfringens*] potential causes [in NEC] include human and animal model data that is more extensive and more congruent than the results presented in this manuscript”. Of the three papers listed, one used a ‘quail model’ with disputed relevance to preterm infants, one focused only on *Clostridium butyricum* which our work found does not use HMOs (Fig. 1A) and so does not impact our current analysis or conclusion, and the last study included only 12 infants, 3 with NEC, and used low-resolution gel-based methods that do not provide information on the pfoA status of the *C. perfringens* (the annotation to species to level was based on sequencing a band with an amplicon of ~400bps, which is error-prone and putative). Thus, our current analysis that combines specific preterm strains and preterm (human) intestinal organoids represents a significance advance on this previous work.

Regarding the comment that “the authors reframe this as an understanding of one infant gut microbe”, we wish to highlight we initially screened unique strains from at least 15 infants for growth on HMOs, performed SCFA

on isolates from 9 infants, performed metabolomics on isolates from 4 infants, performed growth/inhibition assays using isolates from 7 infants, and finally performed organoid experiments with representative pfoA⁺ and pfoA⁻ strains from different infants. Despite the eventual focus on *C. perfringens*, we hope clarifies that the work truly represents more than an understanding of one strain.

2. The description of the use of HMOs by clostridia suggests – in the summary and introduction – that this is a specific adaptation of these species or strains to the infant gut ecosystem. The alternative hypothesis may then be that these clostridia utilize mucin glycans, which bear structural similarity to HMOs, and they can repurpose these for the infant gut (e.g., see Van den Abbelle et al. 2013; Raimondi et al 2021; MacMillan et al. 2019). If this is the case, are these really HMO utilization genes? The authors could (a) demonstrate an inability to utilize mucin and/or (b) show a phylogenomic analysis where these infant-derived strains cluster specifically among *C. perfringens* or other human-associated clostridia to help support the assertions that these functions are clear adaptations to the infant gut – which would facilitate the claim for colonization and the other beneficial features described by the authors.

We appreciate the opportunity to clarify this. We do not suggest that these *Clostridium* are specifically adapted to the infant gut, rather we highlight on line 136 that one of the genes upregulated during growth on HMOs has “been shown to act on mucin O-glycans and may also target HMOs owing to structural similarity”, which is in agreement with the reviewer. We also state on line 120 that “Whole genome sequencing confirmed that the *Clostridium* strains did not contain the same HMO utilisation gene cluster observed in *Bifidobacterium* strains”. We do not speculate on this area any further, and do not base any of our assertions on this. We only show which genes and proteins were upregulated during growth on HMOs. Assessment of mucin degradation would not impact this outcome. Lastly, our “claim for colonization” is well founded as most of these *Clostridium* strains were isolated from preterm infant stool samples and our additional analysis of publicly available data (see response to Reviewer 1 and 3) further shows stable temporal colonisation by pfoA⁻ strains.

Ultimately, our main point is that these strains can use HMOs, which our assays unambiguously prove, and they therefore possess enzymatic machinery that enables this.

3. There is a significant focus on DSLNT as a key HMO that drives metabolic output and gene expression (e.g., Figure 1D-F) but Figure 1A shows there is essentially no growth by these organisms (or no data presented) on this sugar. DSLNT is a trace HMO, and the rationale for it as ‘health promoting’ (as the authors write) or being a key colonization determinant for the infant gut is not convincing. While I acknowledge that there are many studies on this HMO in the literature, the context as it is described here – as a key source of *C. perfringens* metabolites or as a colonization determinant for the strain – is not logically supported by the literature and empirical measurements of DSLNT in milk or the infant gut. The inconsistent use of DSLNT as a substrate (e.g., parts of Figure 1 but not Figure 2 or Figure S1) is also not clearly articulated. Why not show growth of the pfoA⁺ strain on DSLNT as well, for example? The data centered on DSLNT utilization needs to be better rationalized.

The reviewer is correct that most of the organisms could not use DSLNT, including pfoA⁺ *C. perfringens* (JC26), which is presented in Figure 1. To also clarify, DSLNT constitutes a minor aspect of this work and was one of many factors justifying our decision to focus on the AM1 isolate, but we felt it was worth mentioning given the consistent associations of this HMO with health (<https://gut.bmj.com/content/61/10/1417.long>; <https://gut.bmj.com/content/67/6/1064.long>; <https://gut.bmj.com/content/70/12/2273.long>). And we feel these health associations justified the limited focus on DSLNT utilisation, versus any of the other HMOs, in the form of RNAseq and proteomics data presented in Fig. 1.

4. The authors have a clear interest in the pfoA⁻ strain because of their hypothesis that it has the potential for competitive exclusion of other taxa (e.g., the pathobionts tested) but there is also allusions to the production of protective metabolites. However, there are comparisons between *Bifidobacterium* and *Clostridium* in terms of their production of metabolites like butyrate – which *bifidobacterium* are not expected to produce at all – but there is no measurements for lactate (which although not a SCFA, is relevant for the gut ecosystem, especially intestinal pH; see Brinck et al 2025). The addition of other relevant metabolites, including lactate and formate, should be included in Figure 2 and the results discussed therein.

Formate was not profiled within the SCFA profiling by Creative Proteomics and was also not detected by metabolomics, so we are unable to add data on formate. Lactate was detected in metabolomics and we have now added it to Fig 2D, Fig S5B-J and Fig S6B. We have also added a statement on lactate to the Results section, incorporating the suggested Brinck et al. 2025 citation, as follows:

“Finally, both phenyllactate (broad spectrum antimicrobial)³⁶ and lactate (major metabolite of the infant gut, acidifies the gut lumen and an intermediate for SCFAs)³⁷ were significantly higher in all *Bifidobacterium* and *Clostridium* CFSs compared to media only, except for *C. baratii* (fig. S6B).”

There is also a condensed discussion of *C. perfringens* metabolites as beneficial, but this includes some compounds that are almost certainly not beneficial (e.g., cadaverine, putrescine, etc). These are enriched by *C. perfringens* (Figure 2) but they are also known to be cytotoxic compounds at physiologically relevant concentrations (e.g., see Del Rio et al 2019). The discussion of these metabolites and their impacts should be more carefully (and objectively) added to the manuscript.

Cadaverine and putrescine are polyamines that have had some beneficial effects attributed to them, as can be read in references 31 and 32 of our manuscript. Indeed, human cells can produce these compounds as part of normal amino acid metabolism. Notwithstanding, we acknowledge there is also evidence that polyamines can have negative effects, as the reviewer states, and we have now added the following to raise this caveat regarding polyamines and ensure we convey a more careful and nuanced viewpoint:

“associated with increased tight junction expression and inflammation reduction, although accumulation to higher concentrations has been shown to induce cytotoxicity”

We have also added the suggested citation of Del Rio et al. 2019.

Minor issues:

The description of *C. perfringens* as ‘colonising’ the infant gut should be supported by strain-level profiling over time – the 16S data in Figure S2 is insufficient to describe the *pfoA*- strain’s ability to colonise (i.e., remain stably associated at consistent levels in an individual over time). Even qPCR for the *pfoA*- strain would be sufficient to help demonstrate that this organism is prevalent and consistent among the infant in the study (e.g., Figure S2).

We would like to clarify that Figure S2 uses publicly available data and as such, we do not have the eDNA from the study to conduct qPCR based assays. To address the issue raised by the reviewer regarding strain-level profiling, we have accessed publicly available metagenome data from another cohort (Olm et al., 2019) to screen for colonisation by *pfoA*- *C. perfringens*. Following analysis of these data, prevalence and persistence plots for *pfoA*- strains have been added as fig. S2E & F respectively, with the results discussed in the main text as follows:

“Further strain-level analysis using publicly available metagenome-assembled genomes showed *pfoA* negative *C. perfringens* strains were relatively common in preterm infants (detected in 32/158 infants), with a median abundance of 0.6% (IQR = 0.01 - 2.06). Greater proportional prevalence (fig. S2E; $p = 0.14$) and relative abundance (Control: median 0.64%, IQR = 0.01– 2.05%; NEC: median 0.26%, IQR = 0.12 – 21.0%; $p = 0.07$) of *pfoA* strains was observed in Control compared to NEC infants, but these differences did not reach the significance threshold. Temporal colonisation by *pfoA* strains was significantly more stable in Controls (median = 100%, IQR = 100 – 100%) compared to NEC cases (median = 53%, IQR = 43 – 77%) (fig. S2F; $p = 0.03$).”

Details on the analysis undertaken have been added to the Methods.

The rationale for which carbon sources used throughout the paper is unclear. In some places, DSLNT is clearly of interest, in others glucose is used as a substrate (though it is rare in the gut), sometimes ‘HMOs’ are used without description, and then single HMOs are interrogated. The authors should (a) rationalize which carbon

sources are used and (b) note which HMOs are used in which experiments clearly throughout the manuscript and figure legends.

We apologise for any confusion and have improved the clarity on this point throughout by consistently referring the reader back to Fig. 1A, as a reference for which HMOs each strain could use. We have made the following updates to the text:

“We therefore quantified SCFAs within culture supernatants of strains grown on the individual HMOs we found they could utilise (Fig. 1A)...”

“...using either a strain-specific mixture of the HMOs we found they can utilise...”

And to the legend of Figure 2:

“Box plots showing individual short chain fatty acid (SCFA) production by species grown on individual HMOs. SCFA profiling was performed on culture supernatants harvested following the HMO growth assay shown in Fig 1A. Per species, the HMOs used were those on which growth was seen (as indicated in Fig 1A).”

“Box plots showing the levels of tryptophan catabolites produced by *Bifidobacterium* and *Clostridium* spp. during growth on on strain-specific mixtures of the HMOs we found they can utilise (Fig. 1A)”

To the legend of Figure S5:

“...of isolates grown on strain-specific mixtures of the HMOs we found they can utilise (Fig. 1A) compared with blank ZMB1 medium.”

To the legend of Figure S6:

“Isolates were grown on strain-specific mixtures of the HMOs we found they can utilise (Fig. 1A).”

There’s a reference to a probiotic *B. infantis* strain (LM1) but no origin is presented in the methods. It stood out in Figure 1A as being the only ‘probiotic’ strain but it is unclear where it originated or how it may be related to other strains of the subspecies.

This is an oversight on our part. We have now added “(LB1; Labinic™, Biofloratech Ltd)” after the first mention of *B. infantis* in the Results section (second paragraph) and have added the following to the methods:

“The *B. infantis* strain LB1 was isolated and identified from a sample of the probiotic product Labinic™ (Biofloratech Ltd) using the same methodology described above.”

Reviewers Comments:

Reviewer #1 (Remarks to the Author):

I limit my response to the authors' rebuttal to my comments.

Response to my general comment. The authors took issue with my mention of weaknesses in the first manuscript because I did not provide specifics about my concerns that *pfoA*-*C. perfringens* strains have not been critically shown to be protective against disease, or improvers of gut functionality. Elaboration was provided in comment 1 in the original review. Therefore, their statement "The limitation raised by the reviewer does not seem to refer to any specific work undertaken as part of the manuscript" does not prompt me to reconsider this concern.

We agree with the reviewer that previously published studies (that lack strain-level analysis) have not yet shown *C. perfringens* strains lacking *pfoA* to be protective against disease or to improve gut functionality. Combining molecular biology, microbiology, and infant organoid co-culture, we demonstrate *pfoA*-*C. perfringens* inhibits the growth of common preterm infant gut pathogens, promotes the growth of beneficial *Bifidobacterium* species, produces beneficial metabolites which are known to enhance healthy gut development and function, and shows promising anti-inflammatory activity. This is new biology for *C. perfringens* and we hope will lead to further studies investigating potential health-associated beneficial effects. Importantly, our novel discovery likely reflects our multi-faceted interdisciplinary approach and thus we maintain that while no previous work has shown this, that is not a "weakness" of the results we present.

Specific concern 1 (2 parts in their rebuttal). The authors respond to my request to provide more information about how they landed on 29 isolates for this study.

They state that they

"... focused on the most abundant colonizers of preterm infants. This is stated in the main text (first Results paragraph) that these represent a median of 80% (interquartile range 61%-91%) of all relative microbial abundance observed in preterm infants. Thus, we feel this to be an extensive survey of bacteria that colonize preterm infants."

The first paragraph Results section in the revision states:

"We screened the abilities of 29 bacterial isolates, mostly from preterm infant stool, to grow on six different HMOs, and glucose and lactose (Fig. 1A). These species were obtained by untargeted cultivation and represent a median of 80% (interquartile range 61%-91%) of all relative microbial abundance observed in 85 preterm infants. (REF 21)"

Reference 21 is an analysis of 1,431 samples collected over time from 123 very preterm infants born <32 weeks' in a NICU in the UK, published in Nature Microbiology. The infants received 2 combination probiotics or maternal milk, and in that paper, data were presented showing that the probiotics influenced the kinetics of acquisition of members of the gut microbiota. In reviewing this paper, I do not find information in support of this statistic about relative abundance. Though I have no reason to doubt its validity, it would be nice to see substantiation of its use to select the bacteria described in the submitted manuscript.

The reviewer is correct that this information is not specifically presented in the referenced paper, rather it was cited because we used the data from this study to provide the statistics of the combined abundance of the isolates we tested. We have amended the text to clarify this was a re-analysis of the data from that study – "Post hoc we re-analysed metagenome data from Beck et al. ²¹, revealing that these 29 isolates represent a median of 80% (interquartile range 61%-91%) of all relative microbial abundance observed in 123 preterm infants."

Assuming that these taxa are appropriate to select, I still do not see how they end up with the 29 isolates. In re-reading the manuscript, I cannot find how many infants were screened, and on what days of life the stools of origin of these bacteria were collected. Perhaps I missed this information, and such data do not directly affect their in vitro findings, but the presence of cohort and specimen information would strengthen the value of this paper.

We appreciate the reviewer's understanding that this requested information would not impact our *in vitro* findings and thank them for the opportunity to explain this further. As stated in the results "These species were obtained by untargeted cultivation", thus, this was not a systematic culturomics approach and instead focused on strains available in the groups bacterial culture collection. We stopped testing isolates for growth once the main genera/species had been included (see above response where we reached 80% of all abundance), and once we had different strains from some novel HMO users (e.g., *C. perfringens*, hence why these are overrepresented), which is how we happened to end up including 29 isolates. We did not capture specific information on the day of life of the stool sample, but all samples were collected from preterm infants while in neonatal intensive care. Given the persistence of strains from birth until

discharge, the precise age of the infant when an isolate was cultured is unlikely to add additional value. As suggested, we have added information on how many infants the preterm isolates were derived – “We screened the abilities of 29 bacterial isolates, mostly from preterm infant stool (n >15 infants)...”. To further clarify the matter, we have added to Figure 1 legend that “*Clostridium* and *Bifidobacterium* are overrepresented owing to their ability to use HMOs and subsequent testing of species and strain variability in HMO utilisation”.

The authors also missed the point of my concern about the data underlying the justification for studying pfoA-C. *perfringens* (second section of itemized response 1). My point did not relate to data on mechanistic support, instead, the authors should qualify that HMOs and DSLNT are not the complete story in NEC susceptibility, based on the paper cited. The authors also state “... we do not claim that the strain AM1 is associated with protection against NEC, rather, based on our multi-faceted data, we state its broad potential “as a live biotherapy or postbiotic for preterm infants at risk of NEC and likely other related pathologies”. I address this below.

We regret the reviewer feels we missed the point of their comment. Notably, that the AM1 *C. perfringens* isolate could metabolise DSLNT was one of many reasons this strain was chosen to be the focus for subsequent investigation. We have responded to this in more detail below.

Specific concern 2. – concern resolved

We thank the reviewer for the suggestion to improve our work.

Specific concern 3. I still believe that this paper infers that *C. perfringens* in general, and pfoA-C. *perfringens* in particular, offer protection against gut pathology and functionality in preterm infants and against NEC. Such postulated protection is important to the relevance of this work. The authors respond to my concern that the original manuscript made inferences that *C. perfringens* in general, and pfoA-C. *perfringens* in particular, offer protective against gut pathology and functionality in preterm infants and against NEC:

“We would like to reiterate that we were deliberate throughout the manuscript to avoid claims that *C. perfringens* is protective against NEC.” They also state (in response to concern 1): “... we do not claim that the strain AM1 is associated with protection against NEC, rather, based on our multi-faceted data, we state its broad potential “as a live biotherapy or postbiotic for preterm infants at risk of NEC and likely other related pathologies.””

However, in the current manuscript, they continue to imply the disease relevance of their findings:

Line 96: “The HMO disialyllacto-N-tetraose (DSLNT) has been consistently associated with protection from NEC, (REFS 5,7), and *C. perfringens* strain AM1 reached the highest optical density on this HMO (Fig. 1A).”

Only two studies (those cited) have thoroughly investigated HMOs in NEC and both found a lack of DSLNT to be strongly associated with disease. It is also true that *C. perfringens* strain AM1 reached the highest optical density on this HMO. We mention this to rationalise why AM1 became the focus later in the study, specifically it was chosen because it lacked the pfoA gene, it was sensitive to antibiotics, and it could use a range of HMOs including DSLNT. Notwithstanding, we have revised this sentence to remove “consistently” and add that these are “observational studies” - “The HMO disialyllacto-N-tetraose (DSLNT) has previously been associated with protection from NEC in observational studies”

Line 320-321: “... but only a pfoA- strain could use DSLNT, a HMO that is linked to reduced NEC risk. (REFS 5-7).”

To avoid a notion that a link to treating NEC could be inferred from this, we have removed specific mention to reduced NEC risk - “... but only a pfoA- strain could use DSLNT, a health-associated HMO.

Furthermore, in the pending patent application, the authors refer to what are data in the paper and state:

“The inventors have also surprisingly shown that *Clostridium perfringens* is an abundant and prevalent species present in the gut in early life and is present at higher levels in healthy infants compared to infants diagnosed with necrotising enterocolitis (NEC) during the critical window of NEC risk (the first 10 to 40 days of life). Furthermore, the inventors have demonstrated that *Clostridium perfringens* is more abundant in the gut of healthy infants compared to infants before they were diagnosed with NEC, but not after recovery from NEC.

This suggests that *Clostridium perfringens* plays an important role in protecting against NEC and suggests that increased levels of *Clostridium perfringens* in the gut would be beneficial and protective against disease.”

The patent is not a peer-reviewed scientific document but a legal one, where many of its passages were drafted or redrafted by legal professionals. As the reviewer will appreciate, patents aim to cover a broad range of ‘potential applications’ of a novel technology to maximise intellectual property protection. This includes, in this case, possible

relevance to NEC, but the patent also discusses preventing or treating a gastrointestinal disorder, an allergy, nappy rash, colic, or a symptom thereof. We emphasise that these claims do not appear in our manuscript, where we have been careful to restrict our conclusions to the data presented. We have also been transparent in declaring the existence of the patent in the conflicts of interest section, so that readers are fully aware of its presence while recognising that the scientific claims in this paper stand independently of it.

Reviewer #2 (Remarks to the Author):

The authors addressed the majority of the concerns raised in the first round of reviews. I still remain confused about figure 2A. Since the strains can grow on more than one HMO was data obtained from growth on different individual HMOs combined in the same plot for a given strain? Are these triplicates of strains grown in the same HMO? Please clarify whether you chose one specific HMO when the strains were able to utilize more than one. In all other experiments a mix of HMOs specific to the strain was used, which is easier to interpret.

We thank the reviewer for taking the time to improve our work and we apologise for ongoing the confusion over this. SCFA profiling was performed on culture supernatants harvested following the HMO growth assay shown in Fig 1A. Therefore, the plots in Fig 2A contain data obtained from growth of each strain on different individual HMOs, combined into the same box plot for a given species. We have now updated Fig 2A to show each HMO used as a different shaped point. Additionally, we have added the following to the legend of Fig 2A: “These adjusted concentrations for each strain were then combined into a single box plot per species.”

The other are just minor formatting or spelling issues.

Figure 1 legend (D) RNA-seq instead of RNA-srq (F) DSNLT instead of DSNT. Only include abbreviations used in the figure legend at the end.

We have corrected these as suggested.

Given the overlap between the upregulated metabolites identified across species a venn diagram would help with interpretation of the data.

We thank the reviewer for their suggestion to aid reader interpretation of the data and have now added a venn diagram showing the numbers of upregulated metabolites shared between each of the five preterm infant-derived *Clostridium*. This has been added as Extended Data Fig. 5B, with the figure legend updated accordingly. The following has also been added to the Results text:

“However, similarity across all five preterm infant-derived *Clostridium* strains was low, with only 13 upregulated metabolites shared by all (Extended Data Fig. 5B).”

Extended figure 2 swap C D panels around so that the proportion and presence of the same bacteria are under each other

We have swapped these panels over and corrected the legend to reflect this.

Extended figure 3 there are abbreviations included that are not used in the figure legend.

These have been removed.

Extended Figure 4 is this data from a single outgrowth or averaged over multiple ones? If average please provide more detail.

We have now updated the legend to the following to clarify that these values are strain level concentrations which have been averaged by species: “**Extended Data Figure 4. Total unadjusted concentration of short chain fatty acids in culture supernatants of *Bifidobacterium* and *Clostridium* spp. grown on glucose (A) and lactose (B).** SCFA profiling was performed on culture supernatants harvested following the HMO growth assay shown in Fig 1A. The raw SCFA concentrations for each strain were averaged per species.”

We have also added “unadjusted” to the caption for Figure 2B for added clarity: “Stacked bar plots showing the total unadjusted concentration of SCFAs in culture supernatants of *Bifidobacterium* and *Clostridium* spp. grown on individual HMOs. The raw SCFA concentrations for each strain were averaged per species.”

Extended Figure 8: “Statistical comparisons were performed using average genus values.” Does that mean that all the

clostridia were grouped together when comparing? If yes it is misleading as the legend states that “Conditions with the same letters are not significantly different.” When those conditions were never tested but grouped together

To confirm, all statistics are presented at the genus level, but individual species are shown to highlight the species-to-species variation. To avoid potential confusion since “conditions” are used to describe species elsewhere in the manuscript, we have updated the legend to refer to “groups” instead of “conditions”: “Groups with the same letters are not significantly different.”

Data:

Since the authors mention they will deposit the genome sequences please provide identifiers for SRA.

We are in the process of submitting the sequence data for the 11 *Clostridium perfringens* isolates to SRA (currently under submission ID: SUB15810063) and will ensure the finalised accession numbers are available at the time of publication.

Text spelling:

Results

CAZyme instead of cazyme

Extended Data Fig. 3D and S3E. There is no figure S3E.

Methods

and relative abundance

The above spelling corrections have been made.

Reviewer #4 (Remarks to the Author):

Overall, the authors have sufficiently addressed or rebutted the concerns of the original Reviewer 3, and the manuscript is improved as a result.

We thank the reviewer for their feedback and positive assessment of our changes to the manuscript.

A few further minor points of clarification are remain:

R3, Comment 3: I agree with the authors that growth of *C. perfringens* pfoA- on DSLNT is interesting and worthy of highlighting. However, I agree with the original reviewer that it was strange to not have included DSLNT in Figure S1. If the rationale for excluding it from Fig S1 was that none of those strains grew on DSLNT, then this should be described in the text.

Growth on DSLNT was not tested during the assays shown in Fig S1/Extended Data Fig 1 due to the exceptionally limited availability of this particular HMO (owing to current challenges to synthesise it). We had added the following to the figure legend to clarify this point - “Growth of these isolates on DSLNT was not tested, due to the limited availability of this HMO.”

R3, Comment 5: The authors have added the caveat that these polyamines are cytotoxic at “higher concentrations”. How do the concentrations identified as toxic in the DelRio paper compare to those in the CFS? Would they consider these to be biologically relevant?

Our metabolomics dataset did not give absolute quantities of each metabolite, only relative intensity values. As such we cannot make a comment on how the concentrations detected in the CFS compare with those published from other studies. However, we agree with the reviewer that such a comparison would be useful and accurate quantification of metabolites should be considered in future work. We have added this as a limitation to the Discussion of the manuscript:

“A limitation of the untargeted metabolomics data shown here, however, is that only relative intensities were obtained for each metabolite, which prevents direct comparison with published quantitative datasets.”

R3, Comment 7: Further clarification is required. “We therefore quantified SCFAs within culture supernatants of strains grown on the individual HMOs we found they could utilise (Fig. 1A)...”  This wording suggests one HMO per culture, whereas the following suggests a mix of different HMOs in each culture  “Isolates were grown on strain-specific mixtures of the HMOs we found they can utilise (Fig. 1A).” Given that the results section occurs before

the methods section in Nature publications, the description of these experiments in the results must be clear to readers. Please also clarify in the methods whether the HMOs were mixed at equimolar ratios.

We have now updated the first quoted sentence to the below for greater clarity: “SCFA profiling was therefore performed on culture supernatants harvested following the HMO growth assay shown in Fig 1A, allowing assessment of production during growth on the individual HMOs, glucose or lactose.”

The second quoted sentence is from the legend of Fig 2E. This refers to metabolite data generated from cell free supernatants (CFSs), so are different samples than were used for the SCFA analysis. This is why the isolates were grown on a mixture of HMOs rather than individual HMOs. We have now updated the legend of Fig 2E to clarify this: “(E) Box plots showing the levels of tryptophan catabolites for CFSs generated from *Bifidobacterium* and *Clostridium* spp. during growth on strain-specific growing on cocktails of HMOs we found they can utilise (see Fig. 1A for per strain HMOs).”

The HMOs were not mixed at equimolar ratios, but instead at equal masses. The following sentence has been updated within the “Preparation of bacterial cell free supernatants” section of the methods to reflect this: “the HMOs known to sustain growth of the isolate (Fig. 1A) were added to the second aliquot in equal masses, to a total concentration of 10 mg/ml.”